# Sentinel-3 radar altimetry for river monitoring - a catchment-scale evaluation of satellite water surface elevation from Sentinel-3A and Sentinel-3B

Cecile M. M. Kittel[1], Liguang Jiang[1], Christian Tøttrup[2], and Peter Bauer-Gottwein[1]

[1]Department of Environmental Engineering, Technical University of Denmark, Technical University of Denmark, Kgs. Lyngby, 2800, Denmark
[2]DHI-GRAS, Hørsholm, 2970, Denmark

**Correspondence:** Cecile M. M. Kittel (ceki@env.dtu.dk)

**Abstract.** Sentinel-3 is the first satellite altimetry mission to operate both in Synthetic Aperture Radar (SAR) mode and in open-loop tracking mode nearly globally. Both features are expected to improve the ability of the altimeters to observe inland water bodies. Additionally, the two-satellite constellation offers a unique compromise between spatial and temporal resolution with over 65,000 potential water targets sensed globally. In this study, we evaluate the possibility to extract river water surface elevation (WSE) at catchment level from Sentinel-3A and Sentinel-3B radar altimetry, using Level-1b and Level-2 data from two public platforms: the Copernicus Open Access Hub, (i.e. SciHub), and GPOD (Grid Processing on Demand). The objectives of the study are to demonstrate that by using publicly available processing platforms, such databases can be created for any catchment globally, to suit specific study areas and with a wide range of applications in hydrology. We select the Zambezi River as a study area. In the Zambezi basin, 156 virtual stations (VS) contain useful WSE information in both datasets. The Root Mean Square Deviation (RMSD) is between 2.9 cm and 31.3 cm at six VS where in-situ data are available, and all VS reflect the observed WSE climatology throughout the basin. Some VS are exclusive to either the SciHub or GPOD datasets, highlighting the value of considering multiple processing options beyond global altimetry-based WSE databases. In particular, we show that the processing options available on GPOD affect the number of useful VS; specifically, extending the size of the receiving window, considerably improved data at 13 Sentinel-3 VS. This was largely related to the implementation of GPOD parameters: while correct on board elevation information is crucial, the post-processing options must be adapted to handle the steep changes in the receiving window position. Finally, we extract Sentinel-3 observations over key wetlands in the Zambezi basin. We show that clear seasonal patterns are captured in the Sentinel-3 WSE, reflecting flooding events in the floodplains. These results highlight the benefit of the high spatio-temporal resolution of the dual-satellite constellation.

## 1  Introduction

Monitoring river water levels is an important step in hydrological studies, including characterization of the river dynamics, flood monitoring and forecasting, and the planning and designing of water resources infrastructure. The last decades have seen a steady decline in available water monitoring information, particularly in Africa (Hannah et al., 2011; Vörösmarty et al.,

2001). Furthermore, it is often impractical to measure water levels in floodplains in-situ. In the last 25 years, satellite radar altimetry has therefore provided an important, alternative source of water surface elevation (WSE) observations at so-called virtual stations (VS), or crossings between the satellite tracks and river center line.

Advancements in instrument design and processing tools have steadily improved the accuracy of data products to the order of decimeters (e.g. Vu et al. (2018); and Villadsen et al. (2016) for a summary of mission performance evaluations across the literature). Satellite radar altimetry has been widely used in hydrological studies, for instance to monitor and quantify storage variations at regional scale (e.g. Arsen et al., 2015; Jiang et al., 2017a; Boergens et al., 2017; Kleinherenbrink et al., 2015; Villadsen et al., 2015), to assess river dynamics and estimate river discharge (e.g. Domeneghetti et al., 2014; Kittel et al., 2018; Michailovsky et al., 2013; Schneider et al., 2017; Bogning et al., 2018) and to constrain hydrologic/hydrodynamic model parameters (e.g. Getirana and Peters-Lidard, 2013; Liu et al., 2015; Jiang et al., 2019b). Altimetry has proven extremely valuable in poorly gauged regions for hydrologic modelling. For example, in Kittel et al. (2018), WSE from Envisat and Jason-2 was used to calibrate a rainfall-runoff model of the Ogooué River. The observations supplemented historical discharge records by providing contemporary observations of river levels, and were shown to help constrain routing model parameters in the poorly gauged catchment.

Wetlands and floodplains provide important economic and ecological services and are intrinsically linked to river dynamics. Several studies have used altimetry WSE to characterize river-floodplain interactions (e.g. Park, 2020; Zakharova et al., 2014; Ovando et al., 2018; da Silva et al., 2012; Dettmering et al., 2016). Park (2020) recently showed the potential in using satellite altimetry for this purpose using Jason-2 WSE in the Amazon and Zakharova et al. (2014) assessed the seasonal variability of boreal wetlands in Western Siberia using Envisat altimetry. Due to the temporal resolution of Envisat (35 days), an interannual characterization of the wetland processes was not possible. By definition, the satellite orbit is a compromise between spatial and temporal sampling. Dettmering et al. (2016) used Envisat altimetry to characterize water levels in the Pantanal Wetlands but their methods were constrained by the accuracy of the method compared to the level variations in large regions of the Pantanal. They cited SAR technology as a potential solution to overcome these limitations.

The Sentinel-3 mission was developed by the European Space Agency (ESA) mission for the Copernicus program. The mission currently operates in a two-satellite constellation: Sentinel-3A and Sentinel-3B launched in February 2016 and April 2018 respectively. The satellites both carry dual-frequency (Ku- and C-band) Synthetic Aperture Radar Altimeters (SRAL) on board, building on the heritage of the CryoSat-2 and Jason missions (Drinkwater and Rebhan, 2007). In Synthetic Aperture Radar (SAR) mode, the altimeter has a higher along-track resolution of 300 m compared to 1.64 km in Low Resolution Mode (LRM). The instruments operate 100% in SAR mode between 60°N and 60°S, making Sentinel-3 the first satellite altimetry mission to provide near global coverage in SAR mode. SAR altimeters have improved data quality and accuracy in coastal areas and over inland water thanks to the smaller along-track footprint, which is less affected by land contamination (Dinardo et al., 2018; Jiang et al., 2017b; Nielsen et al., 2017; Wingham et al., 2006). Thus smaller water bodies, including narrower rivers can be sensed by the altimeter (Villadsen et al., 2016).

The on board tracking mode of Sentinel-3 is different from the previous SAR altimetry mission CryoSat-2. The tracking mode determines how the range window is re-positioned as the satellite proceeds along its orbit. The positioning of the range

window, which is typically 60 m wide, ensures that the echo reflected by expected surface targets is correctly recorded by the altimeter. CryoSat-2 and SARAL/AltiKa both operate in closed-loop, that is, the range window is positioned based on infor-

mation from previous measurements. However, if the satellite fails to correctly record the river echo, e.g. in steep river valleys where the satellite records the valley top instead of the valley bottom, the error will be transmitted to future measurements as the satellite locks on the wrong target. Studies have demonstrated this challenge for steep-river valleys, e.g. in France (Biancamaria et al., 2018) and in China (Jiang et al., 2017b). In open-loop mode, a priori information about the surface topography controls the range window position, in the form of an on board lookup table, i.e. the Open-Loop Tracking Command (OLTC) tables.

Previous studies have demonstrated the advantages of open-loop tracking and have indicated that Sentinel-3 is less affected by abrupt changes in topography, provided the on board elevation information is correct (Jiang et al., 2020, 2019a). The OLTC is based on DEM information, which must be accurate and up-to-date, as new dams for instance, can alter the surface elevation significantly (Zhang et al., 2020). Sentinel-3 is a marine and land mission, with the altimetric gauging of inland water being a secondary objective to the ocean and ice topographic mission objectives (Drinkwater and Rebhan, 2007). However, the OLTC

tables on-board Sentinel-3A and Sentinel-3B contain a database of over 65,000 virtual stations, or hydrological targets, defined using state-of-the-art water surface masks and high resolution Digital Elevation Models. The OLTC is expected to be a key factor in establishing global databases of water level and to be integrated on future altimetry missions (Le Gac et al., 2019). It is therefore important to understand the implications of the open-loop tracking mode and interactions between the OLTC and post-processing choices on the WSE datasets.

The Sentinel-3 tracks are spaced 52 km apart at the Equator, offering a high spatial density of potential virtual stations (VS) on rivers globally, with a return period of 27 days. This is interesting when compared to traditional short-repeat missions such as the Jason mission (10 days repeat period and 315 km inter-track interval) or Envisat (35 days and 80 km) and geodetic missions such as CryoSat-2 (369 days and 7.5 km). Sentinel-3 provides a denser VS network than Jason-2 and Envisat, with a shorter return period than Envisat. This creates interesting possibilities for monitoring rivers and wetlands at catchment scale.

Several databases provide global, ready-to-use and publicly available time series of WSE for inland water bodies derived from satellite altimetry observations, including from Sentinel-3 e.g. Hydroweb (http://hydroweb.theia-land.fr/), DAHITI (https://dahiti.dgfi.tum.de/en/) and HydroSat (http://hydrosat.gis.uni-stuttgart.de/php/index.php). However, they do not provide full catchment-scale coverage and there is a time-lag between data acquisition and the inclusion of the VS in the database. The Sentinel-3 dataset is available on public processing platforms with dedicated tools for WSE extraction over inland water. In

order to benefit from the high spatio-temporal coverage of Sentinel-3 and large number of hydrological targets, automatic processing workflows and evaluation tools are necessary. For instance, the mission has operated in dual-satellite constellation since November 2018, providing over a year of non-time critical data from Sentinel-3B not yet available on the aforementioned databases.

The aim of this study is to demonstrate the potential of the Sentinel-3 mission in hydrological applications (e.g. monitoring,

modelling and river-floodplain interactions) by extracting a catchment-scale WSE monitoring network of VS using the full Sentinel-3 records. We evaluate the satellite performance directly against in-situ data where these are available and investigate the impact of processing choices on the WSE time series at selected VS. Finally, we explore the potential of the dual-satellite

constellation for spatio-temporal monitoring of wetlands and floodplains. To address these objectives, we use two publicly
accessible databases and present an automatically extracted catchment-scale river WSE monitoring network based on Sentinel-
3 radar altimetry for the Zambezi. All processing steps are performed on publicly accessible databases or using open-access
code.

## 2 Study area and data

### 2.1 The Zambezi

The Zambezi is the largest river in Southern Africa and drains 1,390,000 km$^2$ stretching over eight countries (Fig. 1). Water
resources in the basin are crucial for human consumption, hydropower production, irrigation and ecosystem services (Beilfuss,
2012). There are three distinct seasons: the wet and warm season from November to April, the cool and dry season from May
to July and the hot and dry season between August and October. The river and its tributaries display a strong seasonal signal,
which should be reflected by the satellite altimetry dataset.

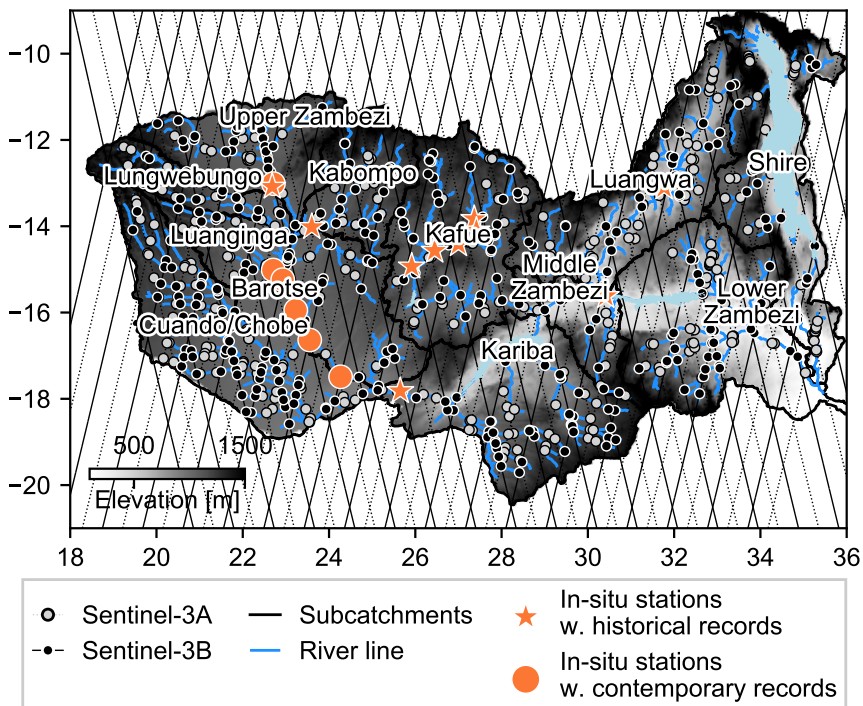

**Figure 1.** Base map of the study area with in-situ stations used for validation of the Sentinel-3 WSE time series. All Sentinel-3 tracks and
river-track-crossings, so-called virtual stations (VS), are shown for the entire Zambezi basin.

Previous studies have evaluated other altimetry missions over the Zambezi, providing a reference in terms of performance of new satellites (Michailovsky et al., 2012; Michailovsky and Bauer-Gottwein, 2014). As shown in Fig. 1, satellite tracks cross the river and its tributaries at multiple locations, and several ground-tracks cross important wetlands (e.g. the Barotse floodplain, Chobe floodplain and the Kafue flats).

Water resources in the Zambezi River basin are increasingly subject to stress, as several drought episodes have affected Southern Africa in the last 30 years (Abiodun et al., 2019). Monitoring is key to adaptation and mitigation efforts. Remote sensing observations of WSE can provide useful monitoring information and inform forecasting and planning tools in poorly instrumented areas. The collection of consistent water level remains a challenge for the member states, especially in the upper parts of the Zambezi, where system failure and vandalism are a constant disruption of the existing ground monitoring system. Thus, a WSE monitoring network based on altimetry observations could ensure steady information on water levels in the catchment even if the existing ground system is not in operation.

## 2.2 Auxiliary data

### 2.2.1 Virtual station localization

A virtual station (VS) is defined as the intersection between a river line and a satellite ground track. Each time the satellite returns on the given pass, new observations of the river can be added to the WSE time series at the virtual station. The river line is from the open data set of global river networks from Yan et al. (2019), which is based on two DEMs (Digital Elevation Model): the SRTM (Shuttle Radar Topography Mission) and ASTER GDEM v.2 (Advanced Spaceborne Thermal Emission and Reflection Radiometer Global Digital Elevation Model) datasets at 90 m and 30 m resolutions respectively. Yan et al. (2019) included a stream burning step prior to the application of the river delineation algorithm to improve the river localization compared to the DEM processing alone particularly over plain areas.

### 2.2.2 Water mask

To ensure that observations are over water, we use v.1.1 of the water occurrence maps from Pekel et al. (2016). The maps are based on 3 million satellite images from Landsat from 1984 to 2018 and indicate seasonal and annual changes in global surface water occurrence at 30-meter resolution. The occurrence map indicates the percentage occurrence of water. We use a threshold of 10% water occurrence frequency over the 34 years of record. A low threshold is chosen on purpose to ensure all valuable data, including seasonal water, is extracted at the cost of a higher outlier frequency. This ensures that data points are not masked out because of low water occurrence probability, which could be partly due to cloud cover.

### 2.2.3 Digital elevation model (DEM)

We use the MERIT DEM (Multi-Error-Removed Improved-Terrain DEM) as reference surface elevation (Yamakazi et al., 2017). MERIT is based on widely used DEMs, including SRTM, which have been corrected for several error sources (speckle

noise, tree height bias etc.). It is provided at 3 sec resolution and referenced to the EGM96 geoid. We reproject the DEM onto the EGM2008 geoid using the VDatum software (Myers et al., 2007), to consistently use the same geoid for all datasets.

### 2.2.4 Open-loop tracking command (OLTC) tables

The OLTC contains targets based on elevation information from either hydrology databases (e.g. Hydroweb), virtual stations networks and a global DEM (e.g. the Altimeter Corrected Elevations v.2 Digital Elevation Model, ACE-2). Details about the generation of the OLTC tables for inland water targets can be found in Le Gac et al. (2019). The on-board table is updated periodically for both satellites. Relevant for this study in particular, is the March 2019 update of the Sentinel-3A OLTC. The OLTC can be visualized on www.altimetry-hydro.eu, where contributions can be submitted for future updates. An overview of the OLTC updates on-board the two satellites is shown in Table 1. In this paper, the latest Sentinel-3B update in June 2020 is not considered due to the limited records available at time of writing. Since March 2019, over 65,000 virtual stations on inland water bodies are defined on-board Sentinel-3A and Sentinel-3B.

**Table 1.** OLTC versions considered in this study. The number of targets corresponds to the latest version and can be visualized and found on www.altimetry-hydro.eu.

|  | Sentinel-3A | Sentinel-3B |
|---|---|---|
| Initial version | 4.2 (24/05/2016) | 2.0 (27/11/2018) |
| Update | 5.0 (09/03/2019) | |
| Targets | 33,261 | 32,515 |

Sentinel-3A and Sentinel-3B operate in open-loop between 60°N and 60°S since March 9th 2019 and since the beginning of mission life respectively. Prior to March 9th 2019, Sentinel-3A followed a mode mask, switching between closed- and open-loop, and operated in open-loop over the Zambezi catchment, with the exception of a short transition phase in March 2019, when the OLTC was updated.

In total, there are 87 new hydrology targets over the Zambezi River from hydrology databases represented in v. 5 of the Sentinel-3A OLTC, compared to only two in v. 4.2, which mainly used ACE2 DEM data. In v. 4.2, 64 additional targets were defined at virtual stations. These targets have been updated with refined elevation information in v. 5 to improve spatial coverage. The Sentinel-3B OLTC v. 2 contains 115 hydrology targets.

### 2.2.5 In-situ water level stations

Level observations from 14 operational gauging stations were kindly provided by the Zambezi River Authority (ZRA), who maintain the dataset. Six of the in-situ gauging stations are in sufficient proximity to a Sentinel-3 virtual station (< 20 km) and located on the same stream, and therefore suitable for direct comparison. The catchment areas are sufficiently similar between the in-situ and virtual stations to justify comparison (e.g. no major tributaries between the two stations): the contributing areas

differ by less than 5.5% in all cases. At all selected stations the level records are labeled as "Very good quality" and provided at daily temporal resolution, with an accuracy of 1 mm.

Additionally, historical records from 2000-2010 are available at 12 additional gauging stations. At ten stations, the in-situ station and Sentinel-3 VS are located on the same stream and within close enough proximity to be representative of similar catchment areas. All stations considered are shown in Fig. 1.

### 2.3 Sentinel-3 Level-1b and Level-2 data

Table 2 summarizes mission specifications for the Sentinel-3 satellites. Level-1b and Level-2 data for the area of interest
are retrieved from 1) the ESA GPOD SARvatore (Grid Processing on Demand SAR Versatile Altimetric Toolkit for Ocean Research and Exploitation) service (available on https://gpod.eo.esa.int/) and 2) the Copernicus open access hub, SciHub (available on https://scihub.copernicus.eu/). Both services are freely available upon registration and use the exact same Level-1a data for processing. With the exception of in-situ observations, none of the processing steps or data are catchment specific.

**Table 2.** Sentinel-3 mission specifications.

|  | Sentinel-3A | Sentinel-3B |
|---|---|---|
| Launch | 16/02/2016 | 25/04/2018 |
| Data coverage | 01/06/2016 - present | 01/11/2018 - present |
| Planned Lifespan | 7 years | 7 years |
| Elapsed lifespan | 4 years | 2 years |
| Orbit | Polar, sun-synchronous | |
|  | 27 day repeat cycle | |
| Ground track separation | 104 km at the Equator | |
|  | (52 km in two-satellite constellation) | |
| Instrument | Synthetic Aperture Radar Altimeter (SRAL) | |
|  | Ku-band | |
|  | (300 m resolution after SAR processing) | |
| Operating mode | Open-loop | |
| Footprint | 300 m x 1.64 km (along-track x across-track) | |

Figure 2 illustrates the processing workflow used for the two datasets from download to the data later presented in the results
section. The Level-1b data and Level-2 data are specific to the two databases and both datasets contain the auxiliary data necessary to compute the water surface elevation. The datasets are evaluated at multiple stages. The following sections detail specific processing on each of the two platforms, and provide additional information on each local processing step.

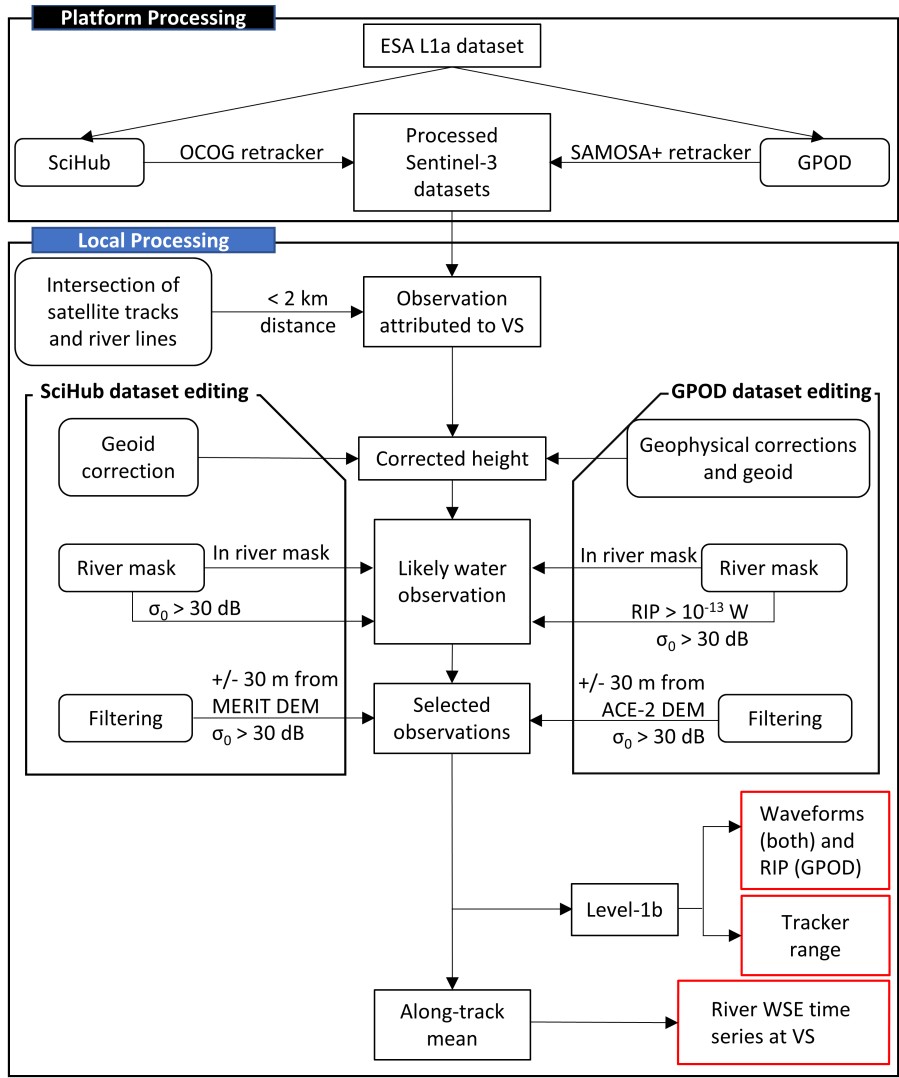

**Figure 2.** Outline of the processing workflow applied in this study to obtain water surface elevation (WSE) time series and waveform statistics. The differences between the two datasets are highlighted left (SciHub) and right (GPOD).

### 2.3.1 Grid Processing on Demand (GPOD) processing

A processing configuration tailored for inland water is available on GPOD. In particular, four specific options are applied during processing (Dinardo et al., 2018):

– A Hamming weighting window is applied on the burst data prior to the azimuth Fast Fourier Transform (FFT) to reduce the impact from off-nadir bright targets by reducing the side-lobes of the Delay-Doppler beam

- A factor two oversampling of the radar waveform prior to the range FFT to improve sampling efficiency of peaky echoes from bright targets

- An extension of the receiving window N times, to better accommodate the Level-1b echoes in the receiving window over rough topography.

The range window is the vertical window during which the altimeter records the return echo from the emitted pulse. During processing, the return echoes from several pulses are stored in a so-called "receiving window" (i.e. a temporal matrix) and combined to form an echogram. For satellites operating in closed-loop mode, there may be a transition phase before the range window is correctly positioned in regions with rapidly changing topography (Dinardo et al., 2018). If the topography is too steep, the standard fixed-size receiving window of 256 samples cannot store the elevation of all the samples in the echogram prior to Level-1b processing (e.g. Figure 5 in Dinardo et al. (2018)). By extending the receiving window, all the echoes can be stored in the same matrix without truncating the leading edge, which will be retracked to obtain the WSE. In open-loop mode, truncation might occur close to changes in the OLTC, where the receiving window may still be positioned according to the previous target. OLTC targets might be far apart due to space limitations of the OLTC (Le Gac et al., 2019), resulting in steep changes when a new target is introduced. Extending the receiving window can accommodates these sudden shifts in the position of the range window as well. We therefore process all tracks using a double and triple receiving window, to identify where the extension might be useful.

GPOD uses the Samosa+ retracker to retrieve the nadir range. Samosa+ is a physically-based retracker specifically dedicated to coastal regions and described in detail in Dinardo et al. (2018). The GPOD datasets are referred to as the "GPOD dataset" in the following sections, with 2x and 3x respectively indicating the double and triple receiving window extension.

### 2.3.2 Copernicus Open Access Hub (SciHub) processing

The Copernicus Open Access Hub (previously Sentinels Scientific Data Hub) provides Sentinel-3 SAR data at various processing levels, including Level-1b and Level-2. In the Level-1b dataset, the echo waveforms are provided, in "counts" and are therefore not directly comparable to the GPOD waveforms. In the Level-2 dataset, several retrackers are used. Over land, the empirical OCOG (Offset Center of Gravity) retracker is used (Wingham et al., 1986). The resulting dataset is referred to as the "SciHub dataset" in the following sections.

### 2.3.3 Data selection

First, we coarsely select observations less than 2 km from a virtual station. We then filter the observations over the water occurrence mask. The along-track resolution is 300 m. Therefore, a buffer of one observation around each water body is allowed. The water mask is based on Landsat observations and thus sensitive to cloud and tree cover. In order to avoid discarding valid observations over water due to gaps in the water mask, observations with high maximum Range Integrated Power (RIP) (> $10^{-13}$ W) or a high backscatter coefficient ($\sigma_0 > 30$ dB) are also classified as water at this stage. The backscatter coefficient

threshold is set based on trial and error for the basin and previous studies (e.g. Michailovsky et al. (2012)). This step also ensures that valid observations are not removed, in case smaller tributaries are not present in the water mask.

### 2.3.4 Correction of the retracked range

The retracked range must be corrected for instrumental and geophysical effects. In both datasets, instrumental corrections have already been applied to the 20 Hz retracked range, $R_{unc}$ (i.e. USO (Ultrastable Oscillator) drift correction, internal path correction, distance antenna-COG (Center of Gravity) and Doppler corrections). $R_{unc}$ must also be corrected for geophysical and propagation effects (i.e. pole tides, solid Earth tides, ionosphere, and dry and wet troposphere), here summed into $R_{geo}$ to obtain the corrected range, $R_c$ (Eq. 1).

$$R_c = R_{unc} - R_{geo} \tag{1}$$

In the GPOD dataset, the geophysical corrections are aggregated and provided as a single variable to be subtracted from the retracked range. In the SciHub dataset, the geophysical corrections have already been subtracted from the OCOG-retracked elevation. In both cases, all corrections are also available separately.

The water surface elevation is the satellite's altitude, h, relative to the reference WGS84 ellipsoid minus the corrected satellite range. The final WSE, $H_{WSE}$, is projected onto the EGM2008 geoid, by subtracting the geoid height, $H_{Geoid}$ (Eq. 2).

$$H_{WSE} = h - R_c - H_{geiod} \tag{2}$$

All variables are expressed in meters. The corrections and geoid data are provided along with the retracked data in each respective dataset. Although both use the EGM2008 geoid model, the geoid model parameters as well as the geophysical corrections can differ slightly. We observe a bias between the two datasets of varying magnitude throughout the basin. Therefore, only the relative change in water surface elevation is considered when comparing the datasets.

### 2.3.5 Outlier filtering

Outlier filtering is based on digital elevation values using the ACE-2 DEM included in the GPOD dataset, and the MERIT DEM for the SciHub dataset. Differences in height exceeding 30 m are considered as outliers. The expected uncertainty of the MERIT DEM is less than 2 m for 58% of land pixels globally (Yamakazi et al., 2017). Based on the project accuracy matrix, ACE-2 has an accuracy better than 10 m for over half of the virtual stations in the basin and better than 16 m throughout the catchment (Berry et al., 2019, 2010). Thus, we do not expect a significant number of false negative outliers due to DEM accuracy based on the allowed window of uncertainty. One exception may be new dams and reservoirs, altering the surface elevation by more than 30 m; however, this does not appear to be an issue in this catchment. We also do not expect the choice of DEM to impact the final results, especially as both DEMs are based on the SRTM DEM. A $\sigma_0$ threshold of 30 dB ensures that only observations of bright targets (such as water) are included in the final selection used to produce WSE time series at each VS.

### 2.3.6 Level-1b waveforms

To evaluate and summarize the Level 1b waveforms, we calculate the following parameters (Jiang et al., 2020):

- Stack Peakiness (SP): ratio between the maximum RIP and sum of RIP

- Maximum Power (MP): maximum value of a waveform

- Pulse Peakiness (PP): ratio between maximum power and the sum of the waveform

- Number of peaks (NP): number of peaks in a waveform – a peak is defined as exceeding 25% of the MP (Jiang et al., 2020)

MP and NP are indicators of the presence and number of bright targets respectively, while SP and PP provide information on the shape of the waveform. A river-like surface is typically smooth and highly reflective, resulting in quasi-specular reflections. This will typically translate into narrow, peaky waveforms and consequently high SP and PP values. We use NP to classify the VS at Level-1b, assuming stations with over 90% single-peak waveforms are likely to be good water targets with useful time series.

The tracker range is the on board positioning of the expected leading edge according to the OLTC. Plotting the along-track tracker range reveals how the range window position changes based on the OLTC targets and updates to the OLTC. The on-board surface elevation must be correct and the surface elevation must be within the range window to obtain useful observations of the water surface. The tracker range also provides insight into the Level-1b processing options of the two datasets, particularly where the range window is repositioned. If this occurs close to a virtual station, there may be impacts on the tracker range depending on how the transition is handled, e.g. by extending the receiving window.

On Sentinel-3, the tracker range is positioned at bin 43 (counting from 0, also called the nominal tracking position), one-third of the full window. The positioning of the window is done through the so-called window delay, or the delay between the pulse emission and the time of record of the tracker range. The epoch is the distance between the nominal tracking position and the retracking position after Level-2 processing. The GPOD dataset contains the epoch (in m), which can be converted to number of bins and used to extract the retracking position. Repositioning to the center of the original reception window requires taking into account 1) oversampling and 2) the receiving window extension (2x or 3x), as described in Section 2.3.1. The tracker range (in m and referenced to the nominal tracking position) is directly provided in the enhanced measurement file from SciHub.

### 2.3.7 Level-2 water surface elevation (WSE)

We calculate the along-track mean of all observations retained at a given virtual station to produce a WSE time series. At six VS, the time series was evaluated against ground observations of water level. In order to account for any vertical bias between the two ground and satellite observations, the mean level at overlapping sensing dates is subtracted from the in-situ and satellite WSE respectively. Performance is evaluated by calculating the RMSD (Root Mean Square Deviation), $D_{RMS}$, between the relative in-situ ($w_g$) and satellite ($w_s$) levels (Eq. 3), and the WRMSD (Weighted RMSD) by dividing with the residuals with

the in-situ standard deviation.

$$D_{RMS} = \sqrt{\frac{1}{N} \sum_{i=1}^{N} (w_{g,i} - w_{s,i})^2} \qquad (3)$$

Based on past mission performance as summarized in Villadsen et al. (2016), RMSD values below 30 cm are considered good, between 30 and 60 cm are considered moderate. We calculate Pearson's correlation coefficient to evaluate the linear correlation between the in-situ and remote sensing WSE. The correlation coefficient should be above 0.9.

Additionally, the annual water level variations recorded by Sentinel-3 are assessed against records from ten in-situ gauging stations using historical observations from 2000-2010. Although the stations could not be used directly due to the lack of temporal overlap, they can still support a visual assessment of the annual water level variations recorded by Sentinel-3.

## 3   Results

The Sentinel-3 VS in the Zambezi are shown in Fig. 3. In total, 364 Sentinel-3A and 367 Sentinel-3B virtual stations are
identified. At each VS, the percentage of missing data is calculated as the number of days with WSE observations divided by the number of days the satellite passed over the VS. In general, the VS with complete records are predominantly located on higher level branches and tributaries of the basin and close to or on floodplains (Fig. 3). These targets are generally wider, perennial and the topography flatter. Conversely, several VS with a high percentage of missing data are located in the headwater subcatchments on smaller tributaries.

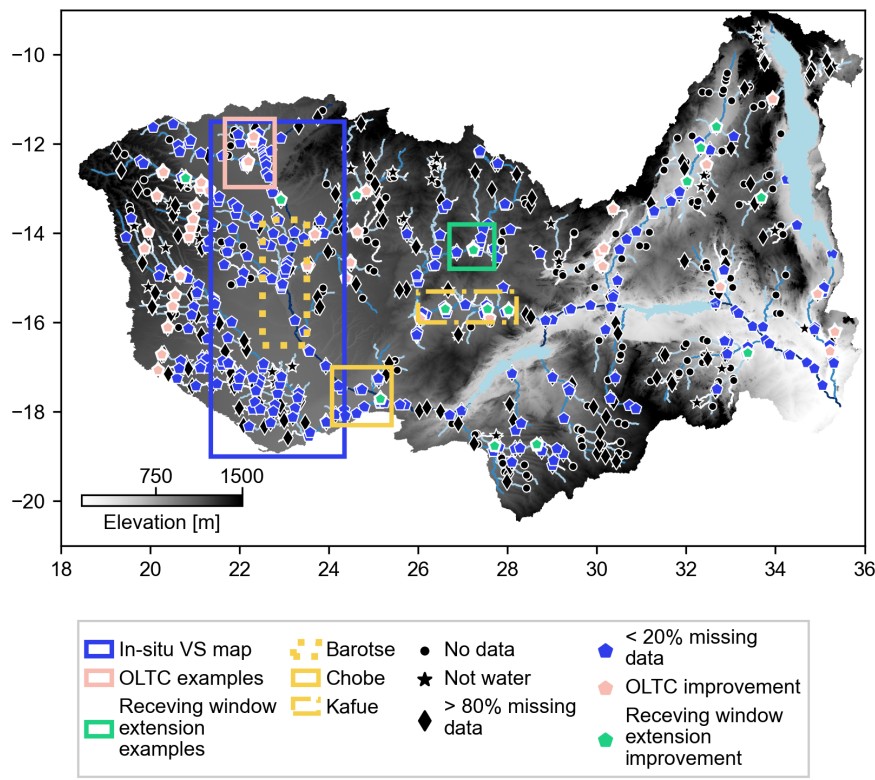

**Figure 3.** Zambezi Sentinel-3A and 3B VS after outlier filtering. Stations which improved by modifying processing steps (either on board through the Sentinel-3A OLTC update or on-ground by extending the receiving window on GPOD) are highlighted separately. The frames indicate examples highlighted in the next sections of this study. Additional maps are included in the supplementary material (Figs. A1, A2, and A3).

## 285  3.1  Evaluation of Sentinel-3 virtual stations (VS) in the Zambezi catchment

Table 3 summarizes the number of Sentinel-3A and Sentinel-3B in the Zambezi catchment with less than 20% missing data (L2 columns) in either dataset and in both . The rejection rate is slightly higher when using the GPOD dataset than when using the SciHub dataset (respectively 67% and 62% for Sentinel-3A and 66% and 63% for Sentinel-3B). This difference can be attributed to the higher proportion of no-data values and the generally lower $\sigma_0$ values in the GPOD dataset. The higher

percentage of no-data values is due to the nature of the Samosa+ retracker: it is a physically based model and more sensitive to erroneous waveforms than the empirical OCOG retracker. $\sigma_0$ is inherently related to the Level-1b processing how the waveform is derived and will be different in the two datasets. We note that, $\sigma_0$ is around 30% higher in the SciHub dataset. We use the same threshold because the intention is to remove obvious non-water targets. Increasing the threshold for the OCOG dataset does not improve outlier filtering as clearly defined non-water targets still have much lower $\sigma_0$ values, while some SciHub

outliers have very high backscatter (high standard deviation within the selected pass, no consistent seasonal pattern, poor L1b statistics etc.).

At 30 Sentinel-3A stations, no observations are available in the either dataset before the March 2019 OLTC update, suggesting the water surface elevation was outside the range window prior to the update causing the poor results prior to the update. Indeed, at over 90% of these stations, the Level-1b statistics are consistent with water targets. We also see that for 13 VS a triple extension of the receiving window improves the time series from the GPOD dataset, confirming the importance of considering this option at certain locations.

**Table 3.** Number of VS fulfilling criteria on Level-1b (L1b, % of VS retained from Level-2) and Level-2 (L2) in GPOD and SciHub datasets using the Samosa+ and OCOG retrackers respectively, and in both datasets. We consider S3A VS with data only after the OLTC update in March 2019 (line "OLTC v. 5") as well as the two processing settings on GPOD (line "3x window extension") separately. The total contains all stations present in both datasets.

| | GPOD | | SciHub | | Both | |
|---|---|---|---|---|---|---|
| | L2 | L1b | L2 | L1b | L2 | L1b |
| **Sentinel-3A - 364 VS** | | | | | | |
| OLTC v. 4.2 | 82 | 75 (91%) | 139 | 60 (43%) | 82 | 54 (66%) |
| 3x extension | 7 | 6 (86%) | - | - | - | - |
| OLTC v. 5 | 32 | 31 (97%) | 45 | 21 (47%) | 32 | 20 (63%) |
| Total | 121 | 112 (93%) | 184 | 81 (44%) | 121 | 78 (64%) |
| **Sentinel-3B - 367 VS** | | | | | | |
| OLTC v. 2 | 113 | 107 (94%) | 176 | 86 (49%) | 113 | 75 (66%) |
| 3x extension | 10 | 7 (70%) | - | - | - | - |
| Total | 123 | 114 (93%) | 176 | 86 (49%) | 123 | 78 (63%) |

We evaluate the Level-1b data to assess whether the observations at the VS are consistent with observations of water (L1b columns in Table 3). In the GPOD dataset, a high percentage of the VS have high PP and SP values (respectively above 0.1 and 0.2) combined with single peak waveforms (NP = 1) and high power (MP > $1e^{-15}$ W). High SP and PP values indicate a quasi-specular reflection, consistent with river surfaces, while a unique peak and high power indicate low contamination from surrounding bright targets.

The majority of stations with complete time series also have a high number of single-peak waveforms. A number of VS have high PP and SP values but multi-peak waveforms due to nearby bright targets (Fig. 4). Furthermore, as the SP and PP cannot be calculated based on the waveforms processed on SciHub, the VS are evaluated at Level-1b based on the NP. We select stations with predominantly single-peak waveforms (along-track median NP = 1 in over 90% of the observations associated to the VS). In total, 74 Sentinel-3A and 77 Sentinel-3B have complete records and promising waveform statistics in both datasets (112 and 81 Sentinel-3A VS and 114 and 85 Sentinel-3B from GPOD and SciHub respectively).

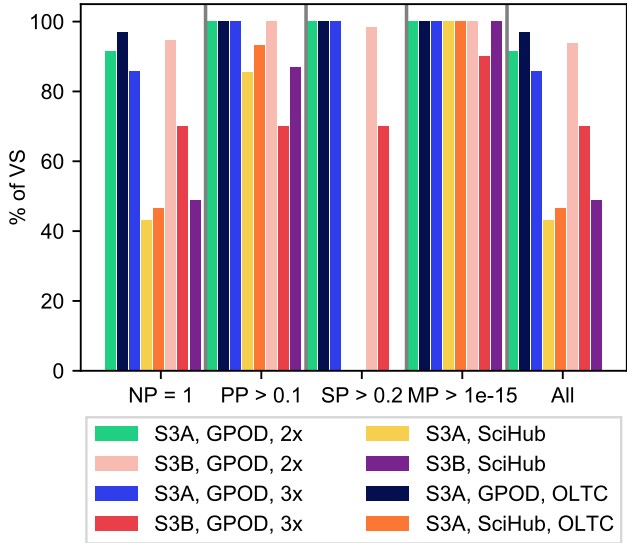

**Figure 4.** Stations fulfilling the Level-1b (L1b) selection criteria in percentage of virtual stations (VS) selected based on Level-2 data. Note that ESA SciHub does not provide the waveform in power; therefore, the stack peakiness (SP) and maximum power (MP) are not calculated and are not part of the "all criteria fulfilled" evaluation. "OLTC" indicates the waveform statistics after the Open-loop tracking command (OLTC) update on Sentinel-3A in March 2019.

The rejection rate is higher in the SciHub dataset, with rejected stations throughout the basin. This is mainly due to the lower percentage of missing data in the Level-2 data. OCOG is an empirical retracker, less likely to fail on non-water waveforms. Samosa+ is a physical retracker developed for coastal regions but suited to inland water targets. If the model misfit is too high, the retracker fails and the VS is rejected based on this missing data.

A closer look at the stations with a large fraction of missing observations or multi-peak waveforms in both datasets reveals that at some stations, the outliers that cause the rejection could be removed with dedicated, manual post-processing if the stations are located in areas of interest. In several cases, the rejected stations are located on narrow rivers crossing seasonal floodplains, with along-track standard deviations exceeding the seasonal variation. This is mostly the case when the station is rejected based on the single-peak criteria, justifying the rejection of the station. The proposed approach allows users to group the VS for further inspection, e.g. starting out with the VS most likely to hold useful river WSE observations.

It is interesting to note that 13 VS are improved by extending the receiving window in the GPOD dataset. Extending the receiving window ensures that the leading edge of the L1b echo is preserved. This is an advantage at VS where topography changes abruptly, as the full return echo can be contained in the receiving window from all beams used during multi-looking.

## 3.2 OLTC tables and geographical location of hydrological targets

Table 4 shows the geographical distribution of the VS selected in section 3.1 in the Zambezi basin and the corresponding number of expected VS based on the OLTC. The Sentinel-3A OLTC update introduced several new VS, which had no useful

information prior to the update. Most new VS are located in the Western part of the catchment (Upper Zambezi, Lungwebungo

and Cuando/Chobe), where there were fewer targets in v. 4.2. The number of VS consistently exceeds the number of targets, which is to be expected for plane areas, where even a single target may be sufficient to correctly track the WSE at multiple nearby crossing points.

**Table 4.** Open-loop tracking command (OLTC) targets before and after March 2019 for the Zambezi catchment and their source (ACE-2 – global DEM or hydrology databases, HDB) – the targets are obtained from https://www.altimetry-hydro.eu/. The Hydroweb Theia S3A virtual stations (VS) within each watershed are available on http://hydroweb.theia-land.fr/). The VS are grouped by major watersheds and according to the processing setting required. The versions indicate the OLTC version after which useful water surface elevation observations could be retrieved at the VS. The total sums correspond to the values in Table 3.

| | S3A | | | | | | | | | S3B | | | |
|---|---|---|---|---|---|---|---|---|---|---|---|---|---|
| OLTC Version | v. 4.2 | | v. 5 | Number of VS | | | | | | v. 2 | Number of VS | | |
| | ACE2 | HDB | HDB | Hydroweb | GPOD | | | SciHub | | HDB | GPOD | | SciHub |
| | | | | | v. 4.2 2x | v. 4.2 3x | v. 5 | v. 4.2 | v. 5 | | v. 2 2x | v. 2 3x | v. 2 |
| Upper Zambezi and Luena | 0 | 0 | 6 | 1 | 6 | 0 | 4 | 4 | 4 | 8 | 22 | 0 | 16 |
| Kabompo | 0 | 0 | 3 | 3 | 1 | 0 | 3 | 0 | 3 | 2 | 1 | 1 | 2 |
| Lungwebungo | 3 | 0 | 7 | 3 | 5 | 0 | 7 | 5 | 5 | 8 | 8 | 1 | 6 |
| Luanginga | 0 | 0 | 0 | 0 | 8 | 0 | 3 | 7 | 2 | 0 | 5 | 0 | 5 |
| Cuando/Chobe | 1 | 0 | 2 | 4 | 12 | 0 | 6 | 10 | 3 | 3 | 21 | 0 | 17 |
| Barotse | 5 | 0 | 5 | 5 | 3 | 0 | 3 | 4 | 2 | 18 | 11 | 1 | 10 |
| Middle Zambezi (Kariba) | 15 | 1 | 9 | 0 | 3 | 0 | 6 | 1 | 0 | 12 | 5 | 0 | 3 |
| Kafue | 7 | 0 | 17 | 14 | 17 | 4 | 3 | 14 | 0 | 14 | 14 | 1 | 10 |
| Mupato | 2 | 0 | 3 | 2 | 3 | 0 | 0 | 2 | 0 | 4 | 2 | 0 | 1 |
| Luangwa | 4 | 0 | 9 | 6 | 6 | 1 | 3 | 4 | 1 | 6 | 4 | 2 | 4 |
| Lower Zambezi (Tete) | 25 | 1 | 20 | 8 | 7 | 1 | 0 | 7 | 0 | 33 | 8 | 0 | 7 |
| Shire | 2 | 0 | 6 | 2 | 4 | 0 | 2 | 2 | 1 | 7 | 3 | 1 | 1 |
| Total | 64 | 2 | 87 | 48 | 75 | 6 | 31 | 60 | 21 | 115 | 107 | 7 | 86 |

## 3.3 Evaluation of the water surface elevation

### 3.3.1 Validation at in-situ stations

The retracked WSE data are compared to the in-situ gauge levels at six locations in the basin; where VS and in-situ stations are sufficiently close geographically (Fig. 5). In all six cases, the twice-extended receiving window is sufficient. The OLTC did not significantly change at the VS considered, meaning WSE observations are available for the entire Sentinel-3 sensing period.

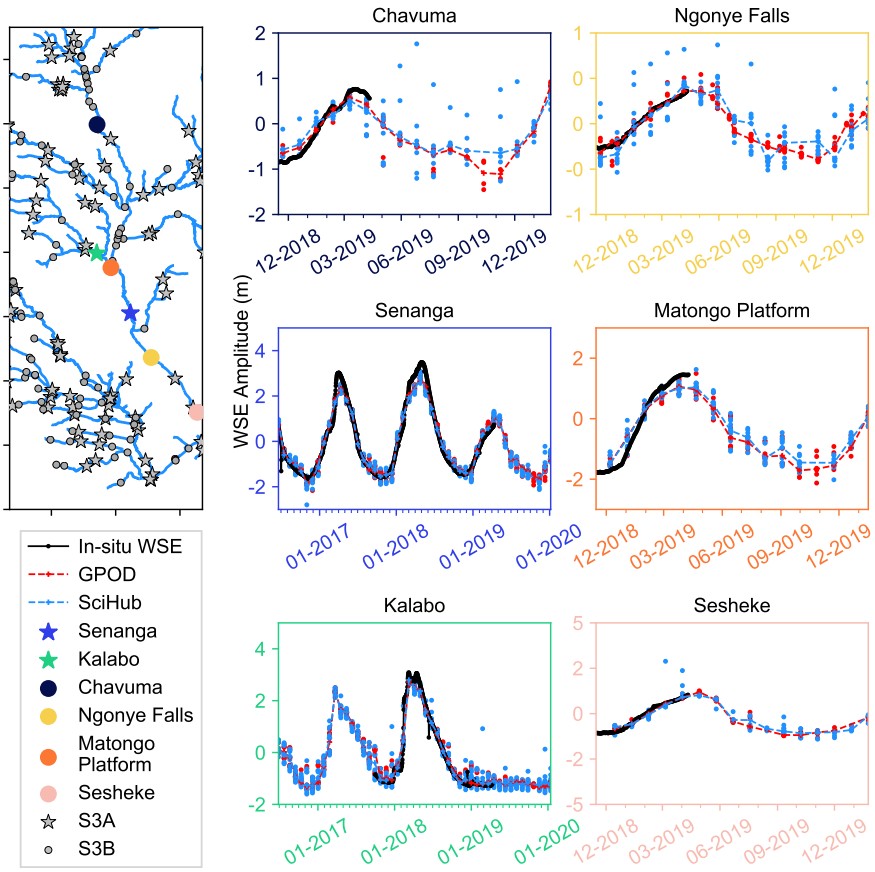

**Figure 5.** In-situ and satellite water surface elevation (WSE) at six locations in the Upper Zambezi basin (blue polygon on Fig. 3). The plot colors correspond to the marker colors for each station in the map. The lines indicate the along-track mean WSE.

Performance at all six stations is highly satisfactory, based on visual inspection, performance statistics and in comparison to performance reported in past studies (Villadsen et al., 2016). The RMSD between the in-situ and satellite relative river levels is below 30 cm at five out of six stations, only the Matongo Platform has a moderate RMSD of 32 cm (Table 5). The largest RMSD are seen for the two VS furthest away from the closest in-situ gauge (19.3 km for Senanga and 15.8 km for the Matongo platform). The systematic deviations between the in-situ and satellite WSE is to be expected given the long distance. For the remaining four stations, the RMSD is less than 15 cm with the GPOD dataset. Michailovsky et al. (2012) obtained RMSD between 24 and 106 cm at Envisat VS in the Zambezi catchment. The improvement in performance is consistent with the instrumental improvement between the two missions. The GPOD dataset performs better than the SciHub dataset at all stations, improving the RMSD with between 1.1 cm (7.5%, at Kalabo) and 10.2 cm (39.2% at Chavuma); except Matongo Platform, where the SciHub dataset improves the RMSD by 1.4 cm (4.5%).

The WRMSD from the GPOD dataset varies between 4.9% and 18.9% of the in-situ standard deviation (Table 5). Thus, the error represents less than 20% of the variation in water level expected at each given location. The error is equivalent to 1.5-6.3% of the mean annual amplitude. This confirms a low degree of uncertainty relative to the seasonal water level amplitudes. A closer look at the seasonal deviations provides additional insight into the uncertainty. As expected from Fig. 5, the underestimation of the peak water level is the main source of error at Senanga and Kalabo, whereas the error is similar across seasons at Ngonye Falls and Chavuma, and larger in the dry season at Sesheke and Matongo Platform.

**Table 5.** Performance statistics compared to neighboring in-situ gauge stations using the G. GPOD processing platform and S. SciHub, to obtain satellite water surface elevation (WSE). The relative root mean square deviation (RMSD) is given in percent of the in-situ standard deviation. At all six stations, observations are available until April 2019. All the stations have complete WSE records since the start of the Sentinel-3A time series (June 2016), with the exception of Kalabo (October 2017).

| In-situ station | VS platform and ID | Distance to VS [km] | River width [m] | RMSD [cm] (% of the mean annual amplitude) | Dry season RMSD [cm] | Wet season RMSD [cm] | WRMSD [%] | $r^2$ |
|---|---|---|---|---|---|---|---|---|
| Senanga | S3A | 19.5 | 260 | G. 25.8 (5.6) | 15.2 | 36.4 | 17.9 | 0.987 |
|  | A062 |  |  | S. 28.2 (6.1) | 16.0 | 39.6 | 19.6 | 0.985 |
| Kalabo | S3A | 4.8 | 35-600 | G. 13.6 (3.1) | 8.6 | 18.8 | 9.4 | 0.998 |
|  | A037 |  | (floodplain) | S. 14.7 (3.4) | 11.4 | 18.6 | 10.1 | 0.998 |
| Ngonye Falls | S3B | 1.7 | 1100 | G. 2.9 (1.5) | 3.0 | 3.7 | 4.9 | 0.997 |
|  | B077 |  |  | S. 7.2 (3.6) | 7.0 | 7.3 | 12.0 | 0.992 |
| Chavuma | S3B | 7.6 | 210 | G. 15.8 (3.3) | 15.9 | 15.7 | 11.9 | 0.997 |
|  | B021 |  |  | S. 26.0 (5.4) | 25.5 | 26.6 | 19.6 | 0.973 |
| Matongo Platform | S3B | 15.8 | 95 | G. 31.3 (6.3) | 35.3 | 28.2 | 18.9 | 0.990 |
|  | B068 |  |  | S. 29.9 (6.0) | 32.0 | 28.4 | 18.1 | 0.992 |
| Sesheke | S3B | 2.7 | 430 | G. 10.5 (1.7) | 13.6 | 7.8 | 5.4 | 0.991 |
|  | B078 |  |  | S. 15.4 (2.5) | 19.2 | 12.2 | 7.9 | 0.979 |

The in-situ stations are mainly located in the Upper Zambezi, therefore the validation is geographically constrained. However, the river morphology at the ground stations is diverse, ranging from 95 m wide rivers to 35-600 m on the Barotse floodplain. Therefore the validation is presumed to be an encouraging indication of the performance basin-wide.

### 3.3.2 Evaluation of hydrological pattern at catchment level

In-situ water level observations are available at ten other locations, where records end in the 2000s. As there is no overlap between the in-situ and VS time series, the stations cannot be used to quantitatively validate the nearest virtual stations. Instead, we visualize the annual water level variations to evaluate whether the time series appear coherent with the expected hydrologic patterns (Fig. 6).

In general, the patterns at several stations are coherent with the annual hydrological cycle observed in the corresponding region over the last two decades. The WSE observed by the satellite corresponds well with the amplitudes recorded at the gauging stations. The satellite time series appear smoother (e.g. at stations 3045, 4669 and 5099). This is logical as the 27-day
return period increases the risk of missing the peak or low flow compared to a daily gauging record. We do note some obvious outliers, e.g. at station 5650, in the Sentinel-3A time series.

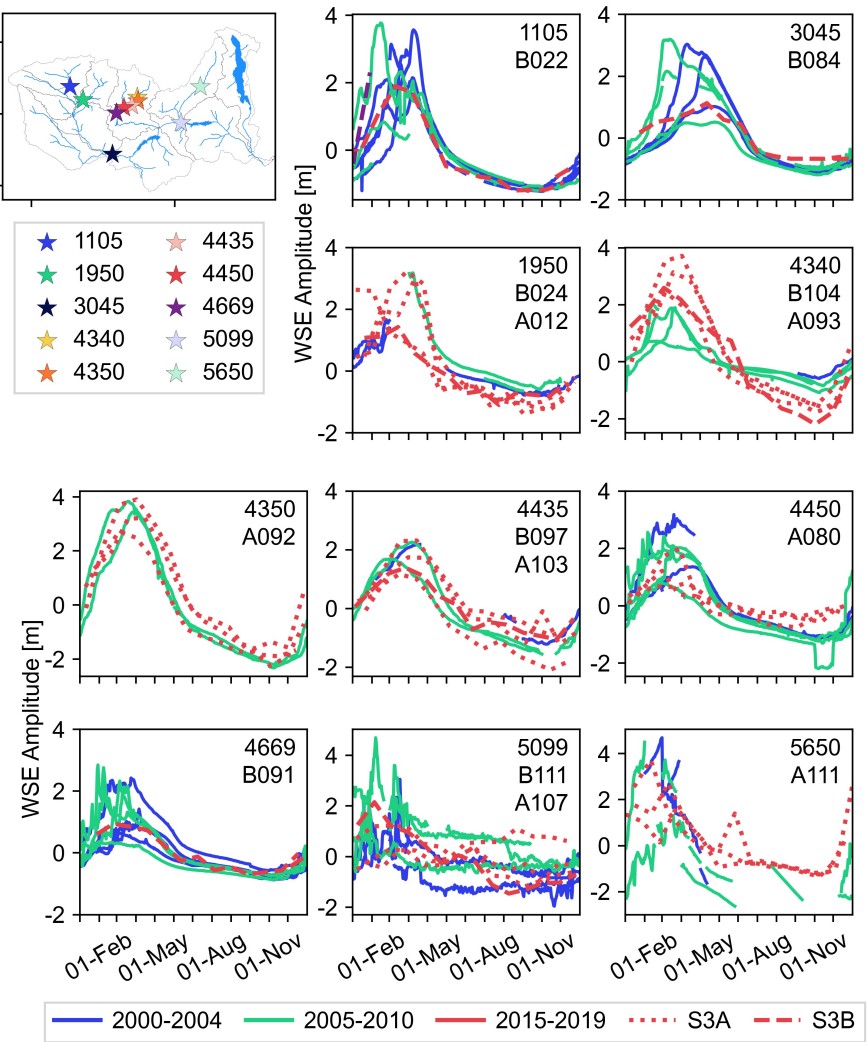

**Figure 6.** Comparison between in-situ annual water surface elevation (WSE) and satellite WSE at ten virtual stations in the Zambezi basin. The colors indicate the time of observation. All elevations are referenced to the long-term average WSE to avoid bias due to the vertical datum. Stations 4340-4669 are all located on the Kafue in close proximity.

### 3.3.3 Annual amplitude of water surface elevation

Fig. 7 shows boxplots of all selected VS based on the evaluation of the Level-1b and Level-2 data (< 20% missing data and along-track NP = 1 for 90% of the tracks). The boxes delimit the IQR (Inter-Quartile Range – or between the first and third quartiles, $Q_{75}$ and $Q_{25}$ respectively) and the whiskers extend from $Q_{25} - 1.5 \times IQR$ to $Q_{25} + 1.5 \times IQR$. The amplitude within the whiskers varies between 1 m and 8.3 m. We note that for Sentinel-3B, the amplitudes are smaller than for Sentinel-3A. This is due to the length of records, with indications of 2019 being a dryer year than 2016-2018, as seen in Fig. 5 at Senanga and Kalabo, and when comparing the Sentinel-3B records to in-situ records in Fig. 6.

A closer look at the WSE recorded at the selected VS reveals a large number of extreme values in the initial dataset (Fig. 7, a). Even after outlier removal, there are still stations with very large amplitudes (> 20 m), which, based on the overall basin statistics, is unlikely. Ground observations of WSE indicate annual amplitudes in the order of magnitude of 5 m and similar values are obtained from Sentinel-3 at directly comparable stations.

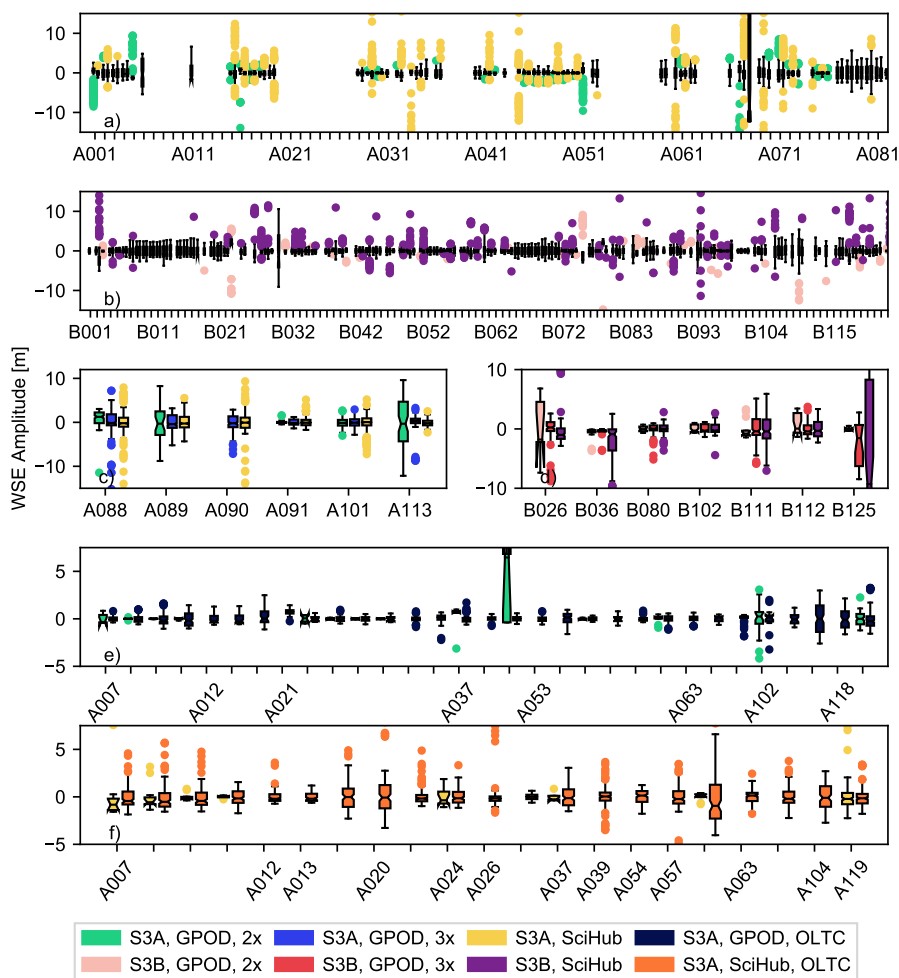

**Figure 7.** Boxplots of valid water surface elevation (WSE) for each virtual station, VS (along the x-axis): a) all S3A VS, b) all S3B VS, c) S3A VS improved with the 3x receiving window extension, d) S3B VS improved with the 3x receiving window extension, and e) and f) S3A VS with observations after OLTC update only in the GPOD dataset and SciHub datasets respectively. The gaps in a) and b) correspond to the VS shown in c), d), e) and f). The points are WSE outside of 1.5 times the InterQuartile Range (IQR), a common measure to identify outliers, to improve readability; the extreme outliers are cropped out of the plots. See Appendix for coordinates of the VS.

There are outlier removal approaches, which could be used to address this issue (e.g. IQR outlier removal, where points outside of the boxplot whiskers would be removed); however, in several cases the filtering also removes peak annual discharge. In some cases, the 3x window extension reduces this amplitude, as does the OLTC update. At other stations, the amplitude increases with the temporal coverage and data volume. If we consider the stations with less than 20% missing data and over 90% single-peak waveforms, there are 156 Sentinel-3 VS in the Zambezi, which contain potentially valuable information about WSE. Thus, automatically processing all Sentinel-3 observations within an area of interest can provide a highly valuable addition to global altimetric WSE databases, by increasing the spatial density of VS at catchment scale. The assessment

based on the degree of missing data and on single-peak waveforms constitutes a preliminary validation of the virtual stations, although dedicated outlier filtering and validation might be necessary at some stations to ensure consistency with the catchment dynamics.

## 4 Discussion

### 4.1 Catchment-scale processing

At four stations in the Upper Zambezi, there are no observations prior to the OLTC update (Fig. 8). The four VS are unavailable on global river WSE databases from radar altimetry (e.g. Hydroweb, DAHITI). These examples illustrate benefit from processing the Sentinel-3 records on GPOD or SciHub as Level-1a data is published. For global databases, it might be impractical to process short time series (in this case less than a year), although they might contain useful information for hydrological studies at catchment level. Furthermore, the OLTC update and ensuing increase in number of hydrology targets increase the number

of potential VS, and are key to the success of the open-loop tracking mode (Le Gac et al., 2019). Unfortunately, at 30 VS in the Zambezi two and a half years of Sentinel-3A observations are invalid due to the lack of OLTC targets.

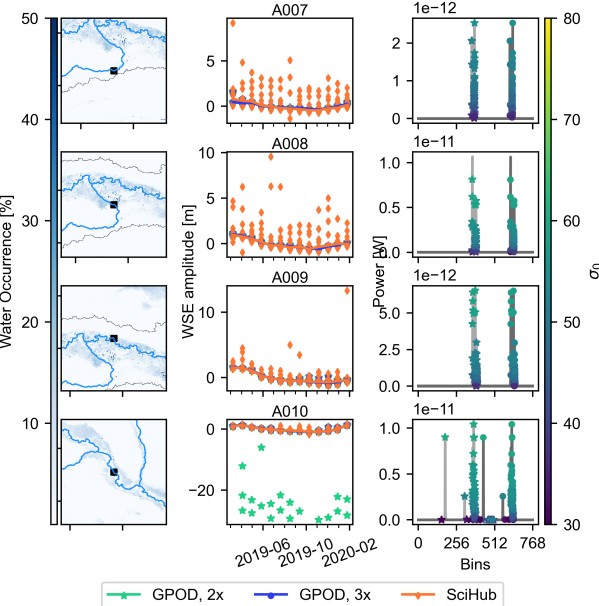

**Figure 8.** Water surface elevation (WSE) time series in the Upper Zambezi and GPOD waveforms after the OLTC update at four virtual stations (VS). The columns show the VS location (left), the WSE time series from March 2019 to January 2020 (middle) and the waveforms (right). For clarity, we only show the GPOD waveforms, as the conclusions are not affected by the Level-1b processing in any of these cases. Due to the window extension and padding of the Hamming window on GPOD, the waveforms are shifted by respectively 256 and 512 bins for the double and triple extensions respectively.

Fig. 9 illustrates the stack waveforms for VS A011 before and after the OLTC update and the positioning of the altimeter reception window. The bright water target is clearly visible in the stack waveform after the OLTC update, whereas the waveform prior to the update is clearly just noise. A closer look at the tracker range clearly indicates the discrepancy between the on board elevation information and the actual surface elevation before the update. The changes in the on board DEM after the OLTC update introduce sharp transitions in the reception window close to the VS, mimicking the effect of steep topographical changes. The short closed-loop transition during the OLTC update reveals that the topography at the target is in fact relatively flat. Fig. 8 reveals several outliers at the last VS (A011) when using the standard GPOD processing options for inland water (GPOD 2x). This is caused by incorrect retracking (points on the $y = 0$ axis in the waveform subplot) and erroneous heights (WSE 10 to 20 m below the mean WSE). At the three other VS, increasing the window extension factor has no effect. The time series at A011 indicates that the triple extension may be more robust even for plain areas. Therefore, processing decisions should not be based on the topography alone but instead take into account the on board information as well. Furthermore, we note that while the along-track spread of the SciHub WSE is wider than the GPOD 3x observations, the final time series are similar.

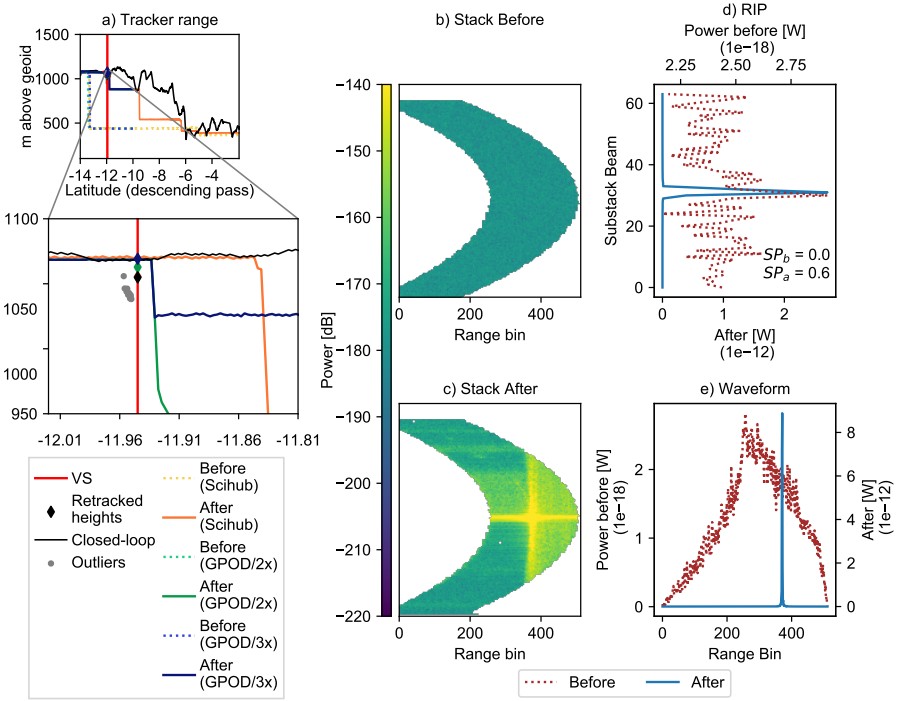

**Figure 9.** Level-1b data at virtual stations (VS) on ground track 135 (descending path). a) Tracker range before, during and after the OLTC update and outliers using GPOD dataset with double window extension, b)-e) waveform statistics – the waveforms have been processed on GPOD: b) and c) stack waveforms d) Range Integrated Power (RIP) and e) waveforms before and after the Sentinel-3A OLTC update.

## 4.2 Processing options on GPOD

The Level-1b processing steps to generate the waveforms are different on GPOD and SciHub, and at some VS, this has clear consequences. Fig. 10 shows the waveforms and WSE time series at VS A102 on the Kafue, located at 1116 m above the geoid. Using the OCOG retracker and the standard SciHub dataset successfully produces a WSE time series with a clear seasonal pattern. When using the GPOD dataset, a 3x extension of the receiving window is necessary to obtain data at this particular VS.

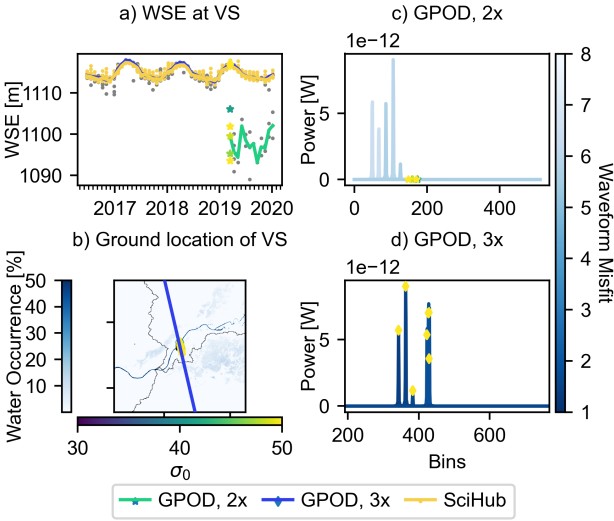

**Figure 10.** Comparison of water surface elevation (WSE) time series (a) and waveforms (c and d) at the virtual station on the Kafue (b) using the GPOD dataset with the Samosa+ retracker applying a double (c) and triple (d) extension of the receiving window and the SciHub dataset. The misfit parameter is provided with the GPOD dataset and is a measure of fit of the waveform model from the Samosa+ retracker to the actual model.

The tracker range from the SciHub dataset suggests that the range window was correctly positioned within +/- 10 m of the surface elevation at around 1111 m (Le Gac et al., 2019). The discrepancy can instead be attributed to the waveform processing, as illustrated in Fig. 11. After the OLTC update a target is defined for the VS at 1113 m and the transition occurs earlier on the pass. The altimeter reception window has shifted just enough that the VS elevation is within the receiving window for all three datasets, including the GPOD dataset with the double extended receiving window.

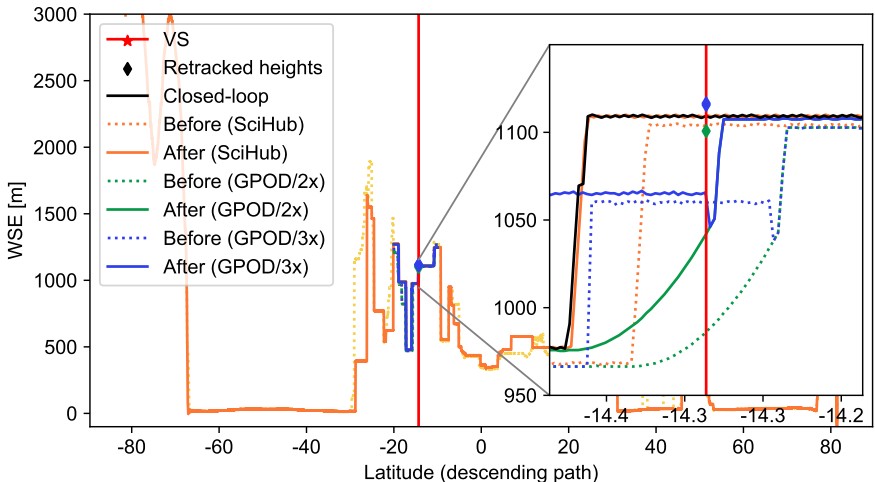

**Figure 11.** Tracker range for the three possible processing setups before and after the OLTC update.

The standard processing produces an abrupt change in the tracker range, consistent with the Sentinel-3 operating mode, where a target is retained until a new one is defined. If we consider the standard GPOD processing, the transition between two targets is smoothed. Increasing the window preserves the Level-1b echoes leading edge (Dinardo et al., 2018), creating a stepwise transition (Fig. 11). The effect of the pseudo-DEM is akin to that of sharp changes in elevation in coastal or

mountainous areas for closed-loop operations (Dinardo et al., 2018). While a smoother transition may be an advantage for closed-loop processing, as demonstrated in Dinardo et al. (2018), however in open-loop processing, where the targets are immediately known, an unnecessary delay is introduced and the window extension becomes necessary to mitigate this.

In such cases, mitigation options include ensuring that the target is defined early enough on the track to avoid consequences for the dataset and extending the receiving window and thus making sure that the full echo can fit in the receiving window and

that the leading edge is preserved (Dinardo et al., 2018). In the example above, the latter is necessary when using the GPOD dataset, and although not critical to data retrieval, the position of the target was also shifted in the OLTC update of March 2019. Based on these findings, we recommend using the triple window extension when processing catchment scale datasets on GPOD to maximize the number of VS.

## 4.3 River-floodplain interaction

The Zambezi is home to several significant wetlands, e.g. the Barotse floodplain, Kafue Flats and Chobe floodplain. At some VS, the WSE reflects the river-floodplain interaction. Fig. 12 is an example from the Barotse floodplain (VS B074). The crossing tracks are both close to the river. The ascending track directly crosses the river, the waveforms and backscatter coefficients closely support a good target. We do see multiple peaks in the waveform as the target nears the edge of the river. The other track crosses the floodplain. When considering the two tracks separately, the interaction is clearly visible: the

river level rises until it reaches the floodplain level. Subsequently, the river floods and water levels in the floodplain increase,

before decreasing again after the wet season. The river level decreases further to its original level, 1.5 m below the floodplain. The coupling is particularly visible during the flood recession phase. The increase of the floodplain appears to begin right before the river reaches the floodplain level. It is unclear whether this is due to the local topography or to artifacts due to the track orientation.

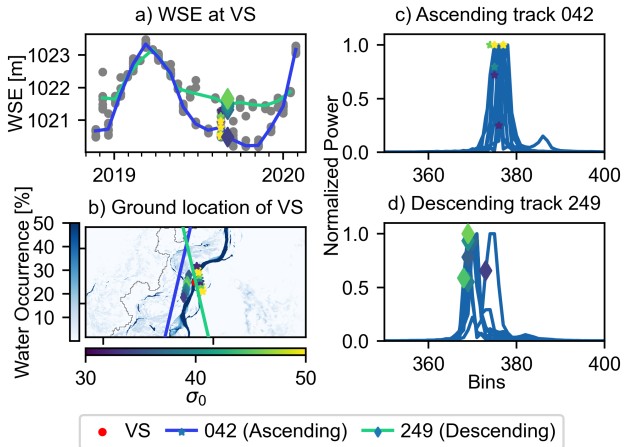

**Figure 12.** Demonstration of river-floodplain interactions at virtual station (VS) B074 on the Barotse floodplain. a) water surface elevation (WSE) at VS with highlighted observations showed in b) along with the water extent map and ground tracks, c) and d) waveforms from the observations from the two respective passes. The observations plotted are the same in a-d to illustrate the different data dimensions (ground location, WSE and backscatter coefficient).

The findings in Fig. 12 motivate an analysis of entire tracks crossing the floodplains. Rather than grouping by coordinates, we here assess all unique passes, known to cross floodplains. The seasonal flooding dynamics are clearly visible at all three evaluated floodplains. Darker blue colors indicate higher water occurrence. The tracks in Fig. 12 correspond to track 498 (descending) and 85 (ascending) in Fig. 13. It is interesting to note the drought in 2019, which is clearly visible in all three wetlands, particularly at the Sentinel-3A VS on track 741, which appears to suggest the level has remained 2 m below the mean

well into the 2019-2020 wet season. Furthermore, there are several valuable observations along Sentinel-3B ground track 498, although the water occurrence is 0%. This is likely due to the frequent cloud cover over the floodplain or vegetation masking the water surface in optical images, stressing the importance of integrating SAR imagery into water mask processing.

The datasets also hold valuable information regarding slope in the wetland. This is a particular feature of the orientation of the wetland compared to the satellite tracks, which creates a spatially dense sampling pattern along the river line and floodplain.

This could be useful in hydrologic/hydrodynamic modelling of the river in the region.

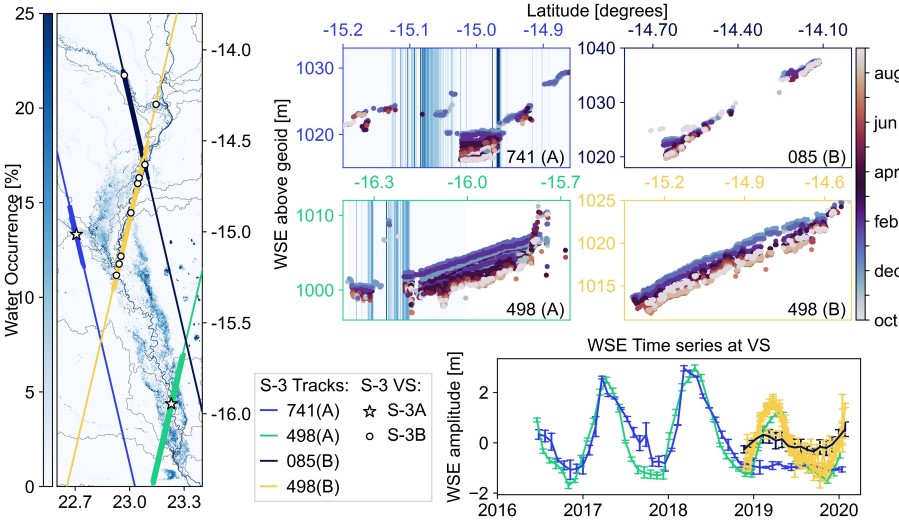

**Figure 13.** Floodplain dynamics in the Barotse floodplain as observed by Sentinel-3A (ground track 741 and 498) and Sentinel-3B (ground track 85 and 498). The water surface elevation (WSE) cross-sections are in order of crossing tracks from West to East. The cyclical color scheme shows the WSE amplitude over the hydrological year. The line colors in the time series correspond to the track colors from the map and the width of the line indicates the observations shown in the scatter plots. The error bars reflect the standard deviation for each pass used in the time series. The water occurrence thresholds are deliberately set from to 0-10% to enable the visualization of the floodplain.

In the Kafue Flats, we see seasonal patterns, with high flow occurring in spring and low flow starting in the late summer (Fig. 14). We also see gradual smoothing in the WSE time series as the distance to the Itezhi-Tezhi reservoir upstream decreases. The upstream WSE is driven by reservoir release with sharp changes in WSE, whereas wetland processes smooth the downstream WSE. There are no valid VS on the tributaries located very close to frequently flooded areas (Sentinel-3B track 298 and Sentinel-3A track 541). The time series at the VS on Sentinel-3A track 070 and Sentinel-3B track 184 both present a sharp increase in January 2019 followed by a sharp return to the previously low level. The pass standard deviation is also larger in the upstream part. The tracks are both in the upstream part of the wetland, with nearby seasonally flooded areas. Both findings are coherent with the results from Jiang et al. (2020), which identified nearby bright targets such as small lakes and ponds as a key source of errors for Sentinel-3. In this case, there are either no observations or unlikely artifacts in the time series.

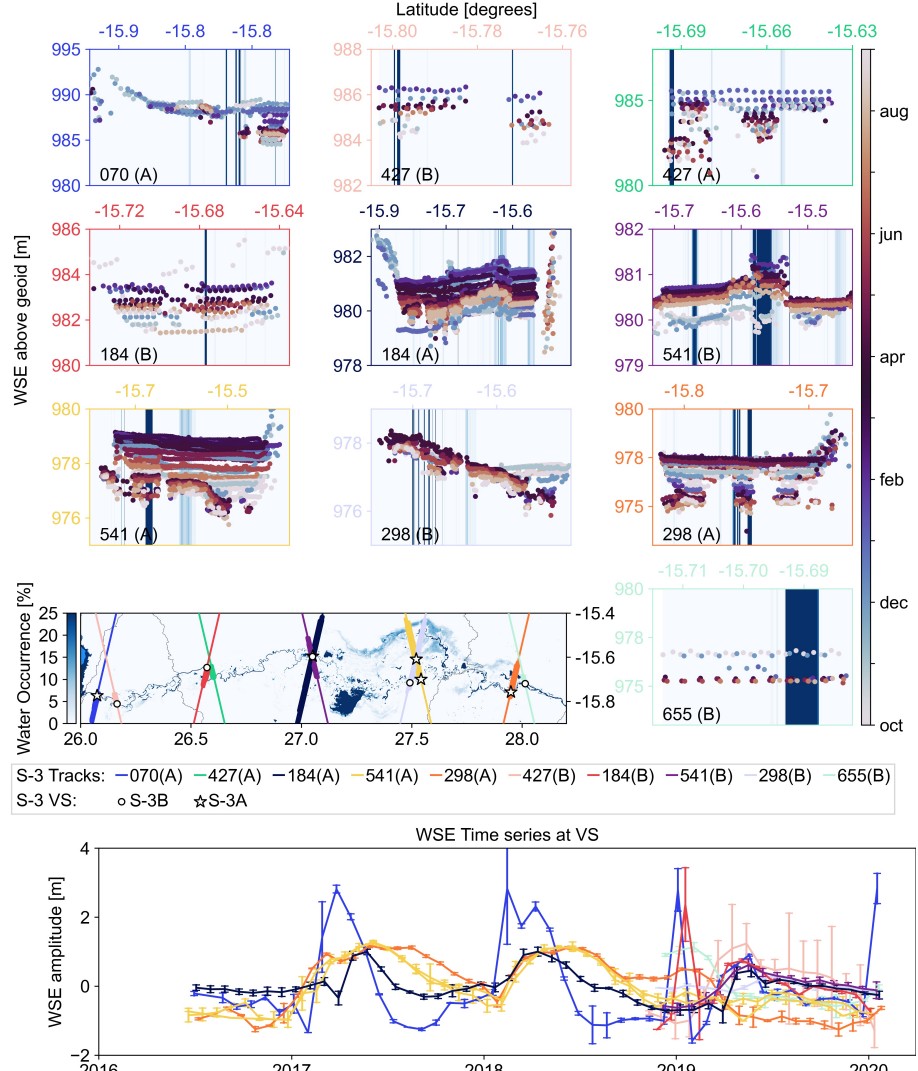

**Figure 14.** Floodplain dynamics in the Kafue Flats as observed by Sentinel-3A and Sentinel-3B. The tracks are ordered by longitude moving left to right on the map. The water surface elevation (WSE) cross-sections are in order of crossing tracks from West to East. The line colors in the time series correspond to the track colors from the map and the width of the line indicates the observations shown in the scatter plots. The frame colors and line colors in the time series correspond to the track colors from the map. The error bars reflect the standard deviation for each pass used in the time series. The water occurrence thresholds are deliberately set from to 0-10% to enable the visualization of the floodplain.

The WSE in the Chobe region reflect the dry 2019 wet season – the peak WSE is lower than the previous two years on record (Fig. 15). We see two different behaviors at the VS in the wetland region. In the Southern portion, prior to confluence with the Zambezi River, the amplitude is smaller. The wet season is slightly delayed with a more gradual decrease after the

peak WSE height; however, in winter 2019-2020, the level has continued to decrease, as seen at all VS in the region. On the Zambezi, the annual amplitude is closer to 5m and there is a clear attenuation in the maximum water level in the 2019-2020 season compared to previous years. This was clear at station 3045 in Figure 6 as well compared to records from 2000-2010.

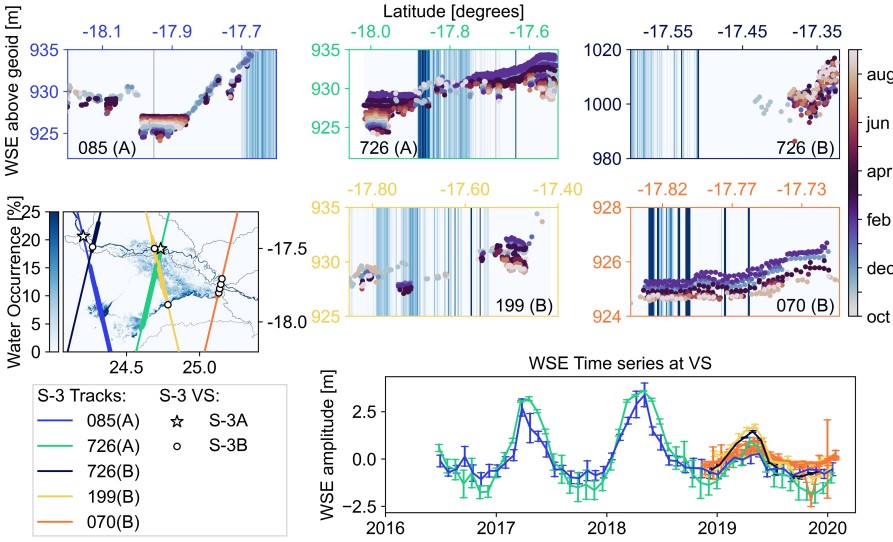

**Figure 15.** Floodplain dynamics in Chobe as observed by Sentinel-3A and Sentinel-3B. The water surface elevation (WSE) cross-sections are in order of crossing tracks from West to East. The cyclical color scheme shows the WSE amplitude over the hydrological year. The line colors in the time series correspond to the track colors from the map and the width of the line indicates the observations shown in the scatter plots. The error bars reflect the standard deviation for each pass used in the time series. The water occurrence thresholds are deliberately set from to 0-10% to enable the visualization of the floodplain.

## 4.4  Hydrological applications

This study explores the potential for extracting Sentinel-3 water surface elevation (WSE) at catchment-level. We present a dense monitoring network for the Zambezi basin, with high spatial coverage and monthly observations of WSE. In this study, the availability of in-situ records is limited relative to the catchment size. Although the performance of the Sentinel-3 satellites cannot be fully validated in the entire catchment, previous studies confirm the performance observed in this study (Jiang et al., 2020). The potential value of altimetry virtual stations (VS) is high in poorly gauged catchments and subcatchments, where altimetry may be the only source of water level observations. The high number of VS throughout the basin can form the basis of a dense monitoring network. Michailovsky et al. (2012) assessed the number of VS in the Zambezi from Envisat and found 423 crossing points against 731 with Sentinel-3, and after careful evaluation, 31 VS had useful records. Although all 156 VS are not manually checked, the results in this study confirm that this number is greatly increased with Sentinel-3. The spatio-temporal sampling of altimetry missions often constrains monitoring capabilities. Particularly the dual-satellite configuration of Sentinel-3 thus offers new, interesting possibilities in a hydrological context. It is important to note that the success is entirely dependent on the accuracy of the OLTC tables as data is missing from the Sentinel-3A records in large part due to the latency

between mission start and OLTC update. At present, the database must be updated manually, automatic download can be set up from SciHub using existing open-source tools. GPOD requires users to submit processing requests through their user accounts. Thus, it is not currently possible to implement an operational observation network using Sentinel-3 data processed on GPOD.

Satellite observations of WSE have been used in several studies to obtain information on river dynamics and to calibrate and update hydrological models (e.g. Domeneghetti, 2016; Dubey et al., 2015; Finsen et al., 2014; Schneider et al., 2018a). Schneider et al. (2018b) and Jiang et al. (2019b) have explored the value of high spatial resolution in calibrating hydrodynamic model parameters by using altimetry WSE observations. Jiang et al. (2019b) evaluated the value of calibrating with different spatio-temporal densities and their results reveal a high benefit from the high spatial distribution of CryoSat-2 and Envisat observations as opposed to the Jason missions. The Sentinel-3 orbit is similar to the Envisat orbit with the added benefit of the two-satellite constellation; therefore, we expect similar results from the integration of Sentinel-3 WSE observations in a similar setup. Furthermore, calibration and assimilation approaches created for CryoSat-2 can also be applied where Sentinel-3 runs parallel to the river line.

Similarly, we show that Sentinel-3 can be used to provide spatio-temporal characterization of floodplains, as clear seasonal patterns can be seen where the satellite ground tracks cross wetlands and floodplains. The connectivity between river and floodplains is an important hydrogeomorphic process, which can significantly alter the floodplain landscape. Several studies have characterized wetland dynamics using satellite altimetry; for instance Park (2020), Zakharova et al. (2014). Sentinel-3 is an interesting candidate for similar studies due to the closer ground-track spacing and reduced footprint from the SAR altimeter. For this application as well, the spatio-temporal sampling of the dual-satellite configuration provides a uniquely advantageous compromise between spatial and temporal resolution. Furthermore, the accuracy achieved at in-situ station Kalabo in the Barotse floodplain (2.9 cm with the GPOD dataset) is promising in terms of characterizing level variations in the decimeter range. This has important implications for successful monitoring of wetlands and floodplains with smaller level fluctuations (Dettmering et al., 2016).

## 5   Conclusions

Satellite radar altimetry has been widely used in the past decades to bridge the gap between data requirements in hydrologic/hydrodynamic simulations and in-situ data availability. The dual satellite mission Sentinel-3 joins a new generation of satellites carrying high-resolution synthetic aperture radar (SAR) altimeters. It is the first mission in a near-polar orbit to carry SAR altimeters and use open-loop tracking with over 65,000 hydrological targets globally. In this study, we explore the capabilities of Sentinel-3 to provide catchment-scale water surface elevation (WSE) observations for hydrological applications. The network can be used to supplement limited in-situ records for monitoring applications and to inform hydrologic/hydrodynamic models.

We extract all Sentinel-3A and Sentinel-3B virtual station datasets over the Zambezi river basin in Africa and developed an automatic workflow to remove outliers, retaining only clear water targets and provide WSE at all possible locations in the basin. In total, the spatial coverage of the dual-satellite mission consists of 731 potential virtual stations in the Zambezi, of which

156 show consistently promising results based on the evaluation of Level-1b waveforms and Level-2 WSE observations across datasets. Where in-situ gauging stations are available, the RMSD is less than 32 cm and there is good coherence with expected hydrological patterns throughout the catchment. A uniquely dense, WSE monitoring network can be extracted at catchment scale by using publicly available processing tools to process the Sentinel-3. The dataset can be used to monitor river WSE and river-floodplain interactions. In particular, we see significant potential for wetlands parallel to satellite tracks, e.g. the Barotse floodplain in the Zambezi.

The proposed approach illustrates the potential of considering the full Sentinel-3 records to achieve complete basin coverage, a substantial supplement to the WSE time series available on global altimetry databases. We demonstrate how this can be achieved on publicly available processing platforms and provide an example for the Zambezi. The dual satellite constellation provides a useful and unique spatio-temporal coverage of river and wetland WSE with important implications for future hydrology-oriented missions.

*Code and data availability.* The python code used in this study is publicly available on GitHub: https://github.com/KittelC/s3_catch. All data sets used in this study are derived from publicly available resources. The database of the Zambezi virtual stations is available on the GitHub repository.

# Appendix A: Supplementary information on VS location

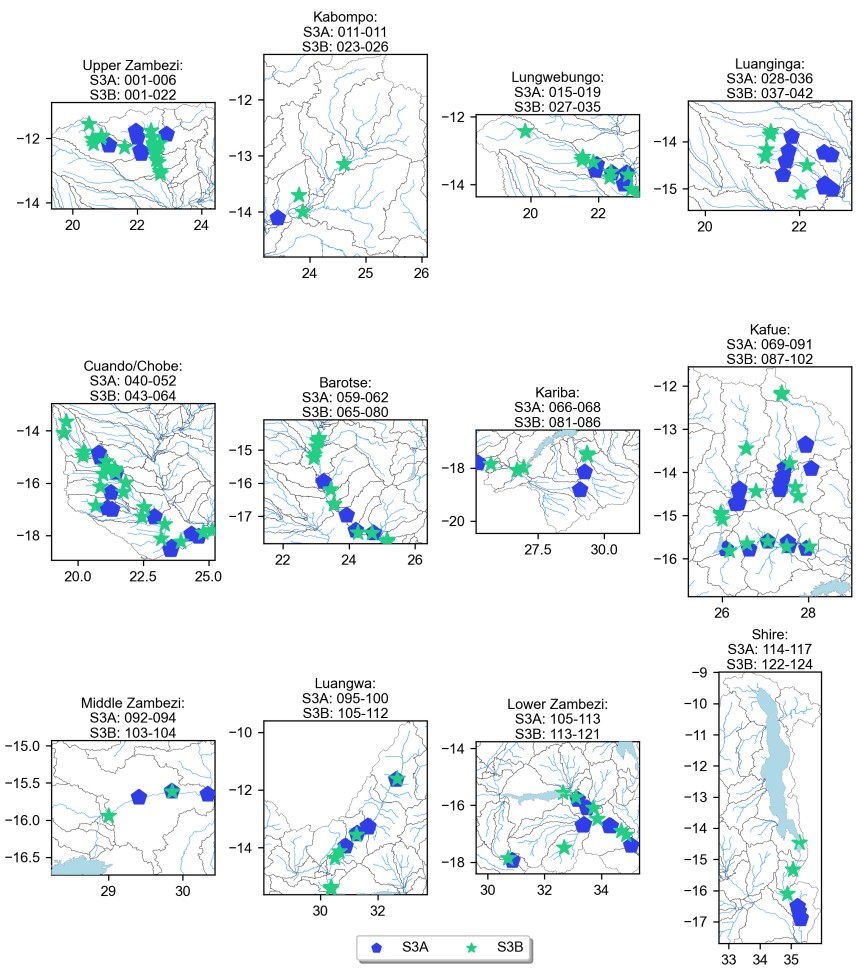

**Figure A1.** All Sentinel-3 VS considered in this study.

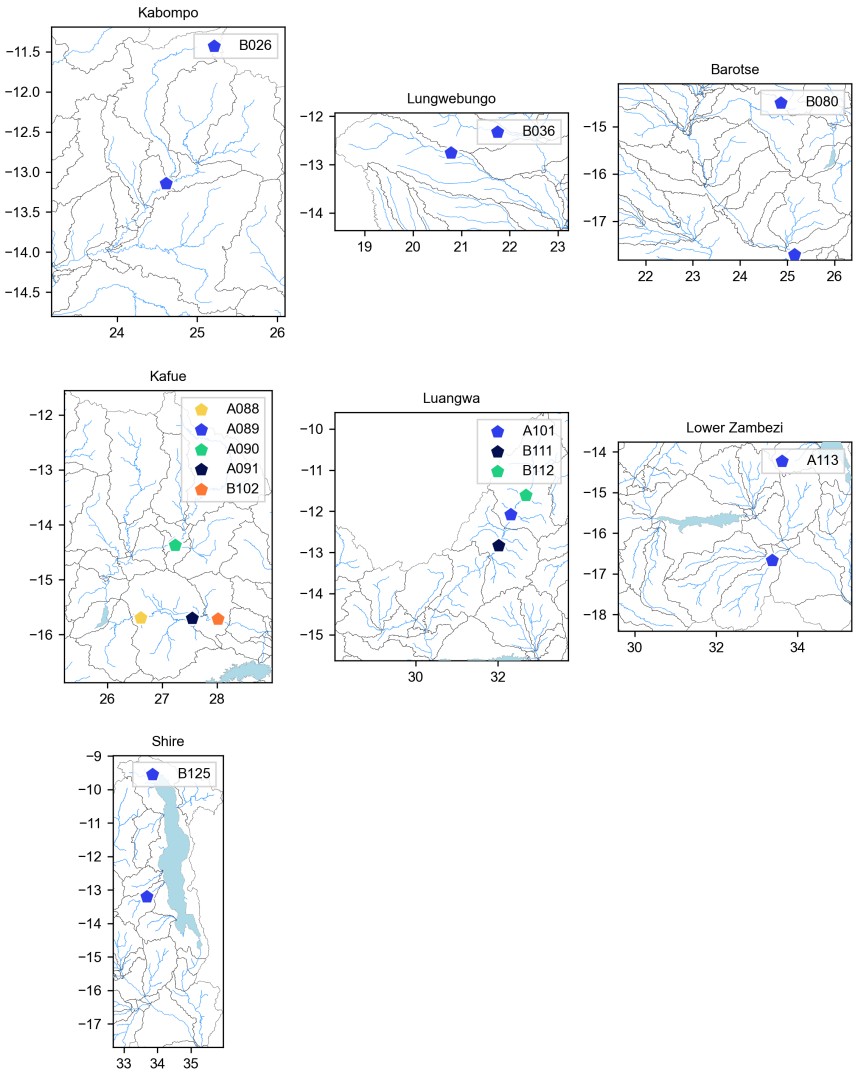

**Figure A2.** S3A and S3B VS improved by the 3x window extension in the GPOD processing options.

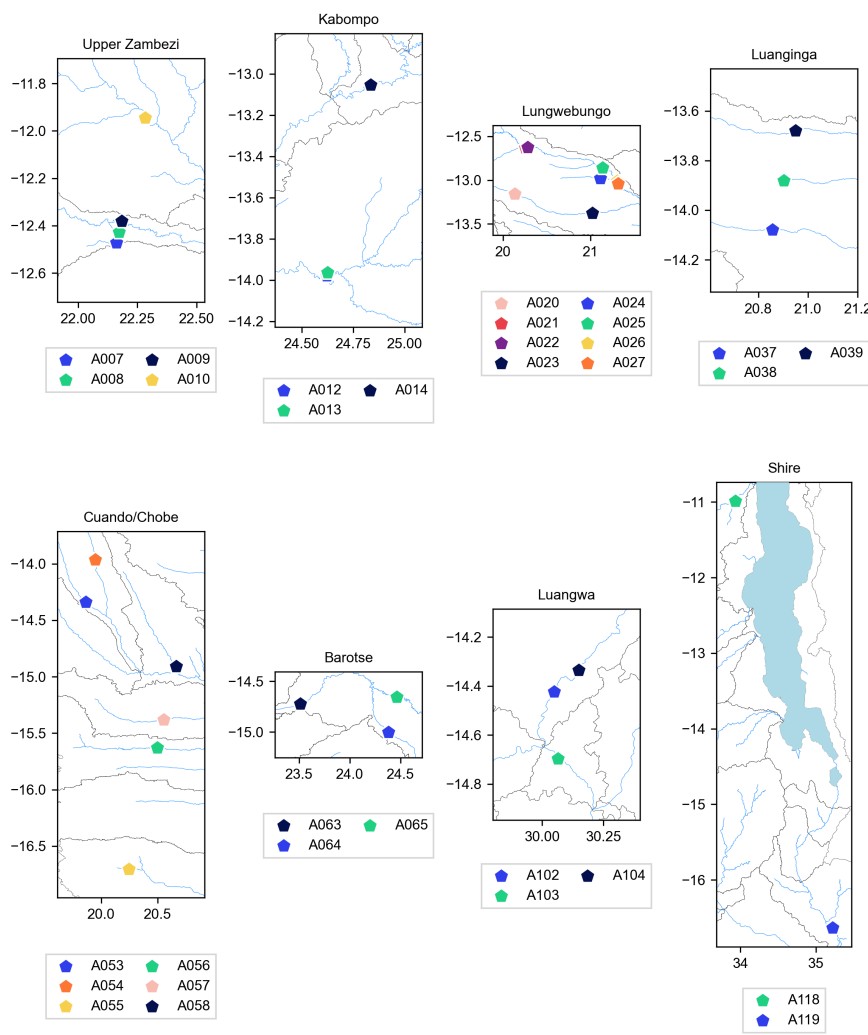

**Figure A3.** Sentinel-3A VS improved by the OLTC update in the GPOD and SciHub datasets.

*Author contributions.* Cécile M. M. Kittel and developed the methodology and code with inputs from Liguang Jiang. The work was supervised by Peter Bauer-Gottwein and Christian Tøttrup. Cécile M. M. Kittel prepared the manuscript including the figures and tables with contributions from all the co-authors.

*Competing interests.* The authors declare that they have no conflict of interest

*Acknowledgements.* This work has been conducted in the context of the European Space Agency (ESA) project Earth Observations for Sustainable Development (EO4SD, contract number 4000117094/16/I-NB). The authors wish to thank the Zambezi River Authority (ZRA) for providing in-situ water level observations. We would like to acknowledge Salvatore Dinardo for his assistance on the GPOD/SARvatore service, and the two anonymous reviewers for their constructive and useful reviews.

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
