# Peer review of "Sentinel-3 radar altimetry for river monitoring - a catchment-scale evaluation of satellite water surface elevation from Sentinel-3A and Sentinel-3B"

_Hydrology and Earth System Sciences, 2020_

## Referee Comment (RC1) · Anonymous Referee #1 · 11 Jun 2020

The manuscript from Kittel et al. presents a validation study of Sentinel-3A/B (S3-A/B) SAR altimeters measurement over the whole Zambezi basin. Time series at 175 virtual stations data have been extracted on the river network, floodplain, reservoirs and wetland. Only 6 in situ gages could be used to validate this database, showing a RMSD between 3 to 31 cm. However, no direct validation can be done for the remaining 169 VS and especially over wetlands (except that the seasonal cycle is well captured and coherent with past in situ observation or nearby VS). Some discussions on the benefits and drawbacks from 1. the open loop tracking mode and 2. the different processing

available on two platforms (SciHub and GPOD) to process S3 data complement the manuscript.

General comments: It should be noted the important work done by the authors to extract this unprecedented database of WSE time series over the whole Zambezi basin and the interesting discussion on the open loop mode and the SAR altimetry processing. However, the authors should better highlight the new discovery from their manuscript and why it has to be published in HESS and not in a more specific remote sensing journal. This is my main concern and the reason why I suggest major revision. As stated by the authors: "The objectives of the study are to evaluate the density of valuable observations and establish a WSE monitoring network. Additionally, we demonstrate the potential application of Sentinel-3 for monitoring river interactions with wetlands and floodplains." The issue is that validation and discussion on SAR and open loop mode have already been done in Jiang et al. (2020) over rivers in China. The submitted manuscript confirms some conclusions from this paper over another basin, but does not bring new information concerning S3 measurements, nor on the hydrology of the Zambezi basin. The application of radar altimetry for monitoring interactions between river, floodplains and wetlands has already been investigated by other studies with different radar altimetry missions. Another previous study from this group (Michailovsky et al., 2012), also studied the Zambezi basin with the Envisat radar altimeter and derived discharges from these WSE with different methods. The main benefit of the submitted manuscript is the important database of WSE over the Zambezi basin derived from S3 missions. So, according to me, the submitted manuscript is a database presentation paper, but the database does not seem to be freely accessible, like other global altimetry database (e.g. Hydroweb, DAHITI...).

Specific comments:

Few clarifications are needed in the abstract. For example, give the name of the two datasets the first time you mention them (line 4). Especially, the sentence "Additional VS are available in both the Copernicus Open Access Hub and GPOD", seems to

suggest that these two datasets are different from the two platform mentioned on line 4, which is not the case. That's why, when reading only the abstract, this sentence is confusing, especially the term "additional". It is not clear from which dataset the Copernicus Hub and GPOD provide additional information.

Line 8: Give the meaning of RMSD acronym.

Line 17: I have some doubts about using S3A/B as a SWOT surrogate. SWOT will do quasi global observations over two swaths, providing not just WSE, but also water extent and surface water slope, which could not be derived from S3A/B only. Besides, the temporal/space resolutions are coarser for S3A/B.

Line 18/19: This sentence is quite general. Similar conclusions were also reached in Jiang et al. (2020) for rivers in China. Besides, in the submitted manuscript, there is no comparison with other mission that does not have SAR mode. So it is difficult to conclude from this manuscript only that SAR mode brings more information than mission with LRM mode.

Lines 27-30: References provided here correspond to only few studies linked to these subjects. That's why I suggest to add "e.g." before the references in brackets.

Line 36: Getting "up-to-date" reference for the databases is very difficult (for example Cretaux et al., 2011 corresponds to the old "lake" version of Hydroweb). To overcome this issue, you could rather point out to the web link for each database. It's just a suggestion, so I let the authors decide if they want to do that or not. There are other altimetry databases than the ones cited in this sentence, like HydroSat (http://hydrosat.gis.uni-stuttgart.de/php/index.php) and GRRATS (Coss et al. 2020, https://doi.org/10.5194/essd-12-137-2020; https://podaac.jpl.nasa.gov/dataset/PRESWOT_HYDRO_GRRATS_L2_VIRTUAL_STATION_HEIGHTS_V1). And for lakes, there is the G-REALM database (https://ipad.fas.usda.gov/cropexplorer/global_reservoir/).

Line 47: For S3 mission, you should rather cite the S3 mission requirements document (S3 MRD), available at http://esamultimedia.esa.int/docs/GMES/GMES_Sentinel3_MRD_V2.0_update.pdf, rather than Jiang et al. (2020).

Line 44: "Sentinel-3 mission is a marine and land mission" This sentence is of course true, but it could give the feeling that both ocean and land requirements are considered equally, which is not the case for the altimeter part of the mission. Indeed, it is worth pointing out that for the topography component of S3 mission "Altimetric gauging of river and lake water levels is a secondary mission objective [...]. This requirement shall not compromise the ability of the altimeter to meet the primary ocean and ice topographic mission objectives." (section 4.4.2 in S3 MRD).

Line 62: "To allow continuation of the historical ERS/Envisat time series, the Sentinel-3 orbit is similar to the orbit of Envisat" This sentence is confusing, as S3A/B cannot continue VS from ERS-1/2 and Envisat, as the orbit and its phasing is not the same. You can argue that S3 provide more spatial sampling than some other missions (e.g. Jason series), but it is not a direct continuation of previous ERS and Envisat ones.

Line 64-67: These sentences are confusing for people who know nothing about the OLTC. It should be clearly stated that the "on board Hydrology Database (HDB) targets" is part of OLTC table. It should be introduced earlier, in the OLTC description section.

Line 137: Could you provide more information on this receiving window? The explanation provided in the current manuscript is interesting, but it is still difficult to understand clearly what this receiving window is and why it is needed. In the manuscript, it is written that it should "not to be confused with the on board reception window", but it's not clearly defined. It is important to better explain it for readers not familiar with SAR altimeter processing (and even more, for those not familiar with altimetry at all).

In section 2.3.3, please cite briefly the corrections taken into account in the two datasets.
Line 170: According to Jiang et al. (2020), the RIP is in Watt, so please indicate the unit in "(>10ˆ-13)"

Lines 174-175: Could you provide some estimates of the two DEM errors (provided in the DEM reference paper or in the DEM quality matrix, for the Zambezi basin). It would help the reader to assess if the 30m threshold is much above the DEM accuracy.

Line 178: According to Jiang et al. (2020), the fit is a gaussian fit, isn't it? It could be worthwhile to mention it (and maybe to add a sentence to explain why the fit is needed).

Line 195: "Retrieving the untracked range gives an assessment of whether the expected WSE was within the on board reception window", I agree, but this statement is very general and you could better explain how you will use this information. If you don't know the expected WSE (which is the case for 169 of your VS), I don't see how you can really make use of this information. Will you compare it to DEM?

Line 211: "WRMSD (Weighted RMSD) by dividing with the residuals with the in-situ standard deviation" it not clear, please rephrase

Equation 3, to be coherent with the text change $D_{RMS}$ with RMSD

Line 217: "We correct for datum shifts by using the WSE amplitude and therefore expect a bias of 0 cm." You need to provide more explanation. First, how did you use the amplitude and to compute what? Second, I don't understand why you need to correct datum shifts, as you already removed to time series "mean level at overlapping sensing dates is subtracted". So why is it needed to add any other bias correction?

Lines 244-251: All the criteria used by the authors are not easy to follow, as they depend of the dataset and the product level. It should be better explained in the methodology section, with a clear flowchart of the process and a more in depth explanation of all the criteria used.

Line 250: "we use NP alone as the L1b selection criterion" but how did you use NP? Using which threshold? It is somewhat difficult to understand why NP is a good criterion,

as multiple targets could be in the waveform and it does not mean that the altimeter is not observing the target of interest. It is especially true for small tributaries, where there are a lot of missing data on Figure 2. The missing data could also be due to your criteria. Could you discuss it in more details?

Line 254: "The rejection rate is higher in the SciHub dataset, with rejected stations throughout the basin." This sentence seems to meet the concern expressed in my previous comment.

Table 2: What is the line "OLTC" in Table 2? It is not explained in the table legend, nor in the text.

Figure 4: This figure does not seem useful, except to state that after OLTC update there is mainly 1 peak in the waveform. But as the NP before the update is not provided, it is difficult to estimate the improvement.

Lines 266-267: "The OLTC contains targets based on elevation information from hydrology databases (e.g. Hydroweb), virtual stations networks and the global ACE2 DEM (Altimeter Corrected Elevations v.2 Digital Elevation Model)" Actually it depends of the OLTC version you are considering, as stated later in your paragraph. According to https://www.altimetry-hydro.eu/ here are the different OLTC table versions over inland waters: - For S3A: * DEM: v5 (Date start: 2019-03-09) * DEM: v4_2 (Date start: 2016-05-24, Date end: 2019-03-01) * DEM: v4_1 (Date start: 2016-04-18, Date end: 2016-05-24) - For S3B: * DEM: v2_0 (Date start: 2018-11-27) * DEM: v1(tandem) (Date start: 2018-06-06, Date end: 2018-10-16) Especially, on the https://www.altimetry-hydro.eu/ you can see that ACE2 DEM is heavily used in v4_2 for S3A, but not used at all in v5 over the Zambezi basin, as shown on Table 3 but not clearly stated in the text. Besides, at line 268 and in other part of the manuscript, it is written that the table has been updated in March 2019. It is true for S3A, but not for S3B, which has been updated sooner (after the end of the tandem phase in November 2018). The OLTC versions are given in Table 3, but never really explained in the text. A good reference for OLTC tables'

generation is (with some validation): Le Gac S., F. Boy, D. Blumstein, L. Lasson and N. Picot (in press). Benefits of the Open-Loop Tracking Command (OLTC): Extending conventional nadir altimetry to inland waters monitoring. Advances in Space Research, https://doi.org/10.1016/j.asr.2019.10.031 I think putting a table to summarized all these OLTC versions and dates could be useful in the manuscript, with some information on OLTC generation (see Le Gac et al., in press). These information should be put somewhere in section 2.

Figure 5: On the map, the black line (sub-basin boundaries?) are not defined, does not seem to be useful and make the map difficult to read. I suggest removing them. Where they are close, S3A and S3B VS are difficult to differentiate. Maybe use different color or level of grey between the two missions. On the sub-plots, write when it is S3A or S3B. In the legend, write to refer to figure 2 for the location of the map within the Zambezi basin (blue polygon on figure 2).

Line 280: "no new targets were uploaded to the OLTC in March 2019 near the two S3A VS" Just to be sure, even if no new targets has been added in march 2019 near these VS, it does not mean that the OLTC table has not been updated in March 2019 for these VS. Is it the case? From figure 5 even if it is the case, the updated value should be pretty similar, as the time series seems pretty stable before and after March 2019.

Line 288: "Samosa+ retracker outperforms the OGOC retracker", first replace OGOC with OCOG. Second, from this sentence, I was expecting much better results with Samosa+ than with OCOG, whereas on table 4, SAMOSA+ is better only by few cm (or %, even most of the times few tenth of %). So I would encourage the authors to add this quantitative information to alleviate this sentence. Besides, Samosa+ comes from GPOD, whereas OCOG comes from SciHub, and processing between these two platforms are different (not just the used retracker, but also the data selection and probably other processing, corrections...) as described on sections 2.3.1 and 2.3.2. How these differences could impact the results shown on Table 4?

Table 4: For Chavuma station, Samosa+ RMSD is equal to 15.8cm and 3.3%, whereas for OCOG the RMSD is 25.6cm and 3.6%. How an almost 10 cm difference in RMSD between Samosa+ and OCOG only translates into 0.3% increase? I think there is an issue with the % computation (or with the RMSD value). Besides, the 9th column entitled "Relative RMSD" corresponds to WRMSD in the text, please replace "Relative RMSD" with "WRMSD" for consistency.

Line 315: "If we consider the stations, which are valid across datasets", how do you define "valid" here? Could you recall the criteria here?

Line 316: "The number of VS is quadrupled compared to using the global database Hydroweb", it is impressive. However, it should be noted that all Zambezi VS on Hydroweb have an "expert validation criteria" (see http://hydroweb.theia-land.fr/?lang=en&basin=ZAMBEZI and https://theia.sedoo.fr/wp-content-theia/uploads/sites/2/2020/04/Handbook_Hydroweb-V2.0-1.pdf). Are all the 145 VS being individually checked and validated (coherent seasonal cycle and amplitude from upstream to downstream VS)? Coherent amplitude and seasonal cycle has been shown only for 10 VS (and compared to in situ gage data only for 6 gages) in the manuscript.

Line 320: "At four stations in the Upper Zambezi, there are no valid observations at any of the VS prior to the OLTC update (Fig. 8)" I don't see how figure 8 shows that there is no valid observation before OLTC update, as figure 8 is only showing data after (S3A) OLTC update.

Figure 9: Concerning the zoom on the WSE vs. latitude plot (between -12.01°N and -11.81°N), it might be because of the color code, but it seems "after Schihub" WSE is in between 1050m and 1100m, whereas "After (GPOD/3x" WSE is in between 1000m and 1050m and "After (GPOD/2x" WSE is below 950m. I don't understand why there are not more consistent. I understand it is near the transition, which affect GPOD when changing the receiving window, but why is it that different, especially why "GPOD/3x"

is above "GPOD/2x" and not below (by tripling the window, you should have more data after the transition)? Besides, on the WSE vs. latitudes plots, I would suggest to draw all "Before" curves with dashed lines, to make them easier to differentiate with the "After" curves.

Line 350: "According to the OLTC website," please give the URL of this website (I guess it is https://www.altimetry-hydro.eu/)

Section 4.2, which VS is considered here? A86 VS on figure B1? Besides, it to better see the impact of the platform processing versus the OLTC value, it could be good to show the OLTC table value (i. e. position of the tracking window), rather than the retracked value converted to WSE. It would help to show the transition and why you need to extend the receiving window with GPOD and then you can discuss the difference between the two platforms.

Lines 370-372: The second option is the one chosen during the March 2019 update, isn't it?

Line 389: "This is likely due to the frequent cloud cover over the floodplain." Or maybe due to vegetation cover masking water?

Line 412: the references provided here are just examples of studies using altimetry to calibrate and update hydrology model, so I suggest putting "e.g." before the references.

Lines 423-428: There is not just Park (2020) and your study which investigated connectivity between river and floodplains. Could you increase your references list?

Line 429: "The cross-sections extracted over floodplains are similar to observations expected from the future SWOT" I disagree with this statement. Even if Sentinel-3 mission provides much more spatial observations than other altimetry missions (like Jason series), it is not comparable to SWOT measurements, which will provides images of WSE. So rephrase this sentence accordingly.

Line 430: Concerning SWOT mission, I think a better reference for the interested reader will the SWOT Science Requirements Document (SRD) rather than Domeneghetti et al. (2018). SWOT SRD could be accessed with the following link: https://swot.jpl.nasa.gov/system/documents/files/2176_2176_D-61923_SRD_Rev_B_20181113.pdf

Line 432: "Similar information can already be extracted from the Sentinel-3 dataset in selected locations" Similarly to my previous comment, I think this sentence should be rephrased. S3 is providing WSE, but not water mask and of course slope could be computed between close VS, but it is far from being the one expected from SWOT images...

Lines 445-446: "We extract over 360 virtual stations from each satellite of which over 70 are validated based on the waveforms and temporal coverage for each Sentinel-3 satellite" Why stating this in the conclusion and not in the core of the manuscript? In the abstract 170 VS are mentioned. The same goes for the 70 validated VS.

Section 4.4 and 5: I find it strange to have perspectives before conclusions...

---

## Referee Comment (RC2) · Anonymous Referee #2 · 18 Jun 2020

The paper describes the computation and exploitation of satellite altimetry water level time series in the Zambezi basin. According to the authors, the aim of the study is "to assess the potential of the Sentinel-3 mission in hydrological applications". For that purpose, they compare different satellite altimetry pre-processing options (from two different databases) and they analyze the impact of open loop processing. Moreover, a validation by comparison with (few) in-situ ground stations is performed. For three different wetlands within the study basin, the potential of Sentinel-3 for monitoring the interaction of river and floodplain is shown.

[Figure]

General comments:

This is an interesting topic worth publishing. However, some aspects of the paper are not innovative and had been published before by some of the same authors (e.g. OLTC impact by Jiang et al., 2020). Moreover, some parts of the manuscript are quite technically without providing the (less experienced) readers a recommendation on which processing option to use. In my opinion, the most interesting and innovative part of the study is the approach of automatically processing all possible VS of both satellites in the entire basin with the aim to use these time series for assessment of wetland-river interactions. Thus, my recommendation would be to focus on this part of the study by adding a bit more statistics (how many potential VS, how many valid VS, how many VS gained by OLTC,...) and some citations of existing work on wetlands based on satellite altimetry (e.g. Zakharova et al., 2014; Dettmering et al., 2016; Park, 2020). In addition, a (at least theoretical) comparison to classical missions can be added discussing the benefit of the dual satellite constellation (with respect to spatial and temporal coverage) and the measurement mode (SAR/OLTC).

Specific comments:

Line 5: If the objective of the study is to "evaluate the density of valuable observations", you should add some more statistics on the number of VS (see general comment).

Line 18: In my opinion, the paper is not showing the benefits of SAR (with respect to what? LRM?). The denser track network is due to the orbit configuration not the measurement mode, and there is not comparison to LRM data. The RMSD values are similar to those from LRM missions. So, how is the benefit demonstrated?

Line 44: Sentinel-3 is not only an ESA mission => Copernicus

Section 2.2: What about adding additional information on in-situ validation observations and OLTC targets?

Line 104/105: Please add some more information on the stream burning. I'm not sure

what is meant here.

Section 2.3.3: Some detailed info on the corrections is missing (e.g. which models).

Line 171: Sigma0==backscatter?

Section 2.3: I recommend to provide also the web addresses of GPOD and SciHub (in the text or alternatively in Refs or Acknowledgements.

Line 174/175: Are these DEMs good enough to be used in this context. My personal experience is that at least ACE2 includes really large outliers in some regions.

Line 200: "are processed" => how? Median/mean?

Line 204: "six". Where are these stations located. Maybe you can reference to a figure.

Line 210: RMSD or D_{RMS}? Please make consistent

Line 231: "two the" => "the" or "the two"

Figure 2: I can't find any black cycle in the plot. On the other hand blue lines (which I assume to be rivers) not covered by data. The black lines are a bit confusing here. I guess these are sub-basin borders. Please indicate or remove. The additional maps seems to be in the Annex, not in the supplementary material.

Title of 3.1: This is quite technically. What about using a title indicating the aim of this section, e.g. comparison of different L1b pre-processing

Figure 3: What does OLTC stands for here (black and orange)? Before/after OLTC update? Please clarify.

Table 3: Please provide the sum over the entire basin. Include description of GPOD/SciHub version for VS no (I guess it should be 2x, 3x?)

Section 3.3.1. What about adding a discussion on the impact of low number and distribution of the validation sites. Are the validation numbers representative for the entire basin?

Figure 5: in-situ (black) lines are not visible. Are they always available for the whole period? Are there more than one observation available per epoch (=> single along-track measurements instead of mean/median?) Can you add RMSD here?

Line 288: OGOC => OCOG

Table 4: is the difference only due to the retracker? Might the pre-processing play a role? Is the Relative RMSD == WRMSD?

Figure 6: I can not find any orange lines here...

3.3.3: What about adding some more information and interpretation here.

Line 323-324 (and in some other parts of the manuscript): I'm not sure whether it is fair to compare with global WSE databases. Since these databases aim in providing input for hydrological research, the focus is on long time-series. For sure, they are also able to process these VS - however, this has no priority given the short time series of less than 2 years.

Line 342: Is there any statistics available on the percentage of improve-ment/degradation by OLTC in this region?

Line 349ff: "mamsl": all other heights are provided with respect to a geoid. Why not these ones? At least you should explain the abbreviation.

Line 369: options to mitigate: Do you have any recommendation for the users? What preprocessing should I use?

Figure 12: Is there any color change in c) and d) depending on waveform misfit?

Line 385/386: Are there no unique track numbers?

Figure 13: What are the vertical blue lines in crossing tracks 741 and 498? Where are the VS located for tracks 498 and 085? What are the stars and cycles in the left hand plot?

Figure 15: left and right?

4.4 This is more a summery than perspective... Moreover, perspective should be placed after conclusions...

Line 409: "first" => where is second?

Line 429-434: Please reformulate this paragraph: SWOT will provide much more information than S3, especially in cross-track direction. Also CS2 can already extract similar information in selected locations.

Line 441: "should"? => is or is not improving!

Line 446: I don't think that you should name that a "validation"

Line 447: Again: My feeling is, that this is not a fair comparison. Hydroweb is a global database not aiming in complete coverage of entire basins.

Line 452-458: I suggest shifting this paragraph to line 443 (as second paragraph of this section). This would make the paper end with the application, which is the overall focus of your paper according to line 69.

---

## Author Comment (AC1) · 3 Aug 2020

Replies to the comments by anonymous referee #1:

We would like to thank the reviewer for their interest and comments on the manuscript. Below the reviewer's comments are in italic and the replies in normal font.

*The manuscript from Kittel et al. presents a validation study of Sentinel-3A/B (S3- A/B) SAR altimeters measurement over the whole Zambezi basin. Time series at 175 virtual stations data have been extracted on the river network, floodplain, reservoirs and wetland. Only 6 in situ gages could be used to validate this database, showing a RMSD between 3 to 31 cm. However, no direct validation can be done for the remaining 169 VS and especially over wetlands (except that the seasonal cycle is well captured and coherent with past in situ observation or nearby VS). Some discussions on the benefits and drawbacks from 1. the open loop tracking mode and 2. the different processing available on two platforms (SciHub and GPOD) to process S3 data complement the manuscript.*

*General comments: It should be noted the important work done by the authors to extract this unprecedented database of WSE time series over the whole Zambezi basin and the interesting discussion on the open loop mode and the SAR altimetry processing. However, the authors should better highlight the new discovery from their manuscript and why it has to be published in HESS and not in a more specific remote sensing journal. This is my main concern and the reason why I suggest major revision. As stated by the authors: "The objectives of the study are to evaluate the density of valuable observations and establish a WSE monitoring network. Additionally, we demonstrate the potential application of Sentinel-3 for monitoring river interactions with wetlands and floodplains." The issue is that validation and discussion on SAR and open loop mode have already been done in Jiang et al. (2020) over rivers in China. The submitted manuscript confirms some conclusions from this paper over another basin, but does not bring new information concerning S3 measurements, nor on the hydrology of the Zambezi basin. The application of radar altimetry for monitoring interactions between river, floodplains and wetlands has already been investigated by other studies with different radar altimetry missions. Another previous study from this group (Michailovsky et al., 2012), also studied the Zambezi basin with the Envisat radar altimeter and derived discharges from these WSE with different methods. The main benefit of the submitted manuscript is the important database of WSE over the Zambezi basin derived from S3 missions. So, according to me, the submitted manuscript is a database presentation paper, but the database does not seem to be freely accessible, like other global altimetry database (e.g. Hydroweb, DAHITI...).*

We thank the reviewer for the summary and general comments about the manuscript.

Indeed the validation is challenged by the lack of concurrent in-situ data and only 6 in-situ gauges were located close enough to VS to ensure direct validation of the satellite performance. We expanded the validation by considering the hydrological patterns at additional stations with historical records. However, extracting the full Sentinel-3 dataset has the largest potential value in supplementing ground observations in poorly gauged catchments.

As the reviewer correctly points out an important aim of the study is to demonstrate the extraction of a catchment-scale WSE monitoring network from Sentinel-3 observations. To

address these main concerns, we propose to rewrite the introduction and objectives of the study to more clearly reflect this (see also specific suggestions below).

The database will be publicly available in conjunction with the paper and a link will be provided in the reviewed version. The code will also be publicized. The purpose is not the specific Zambezi-database (although that is a non-negligible product of the study) but rather to demonstrate that by using the publicly available processing platforms, such databases can be created for any catchment globally. This will greatly facilitate non-experts in altimetry to utilize altimetry dataset. We believe that a framework to extract catchment-scale monitoring networks to suit specific study areas has a wide range of applications in hydrology, which is why we believe HESS is a good target journal for publication.

*Specific comments:*

*Few clarifications are needed in the abstract. For example, give the name of the two datasets the first time you mention them (line 4). Especially, the sentence "Additional VS are available in both the Copernicus Open Access Hub and GPOD", seems to suggest that these two datasets are different from the two platform mentioned on line 4, which is not the case. That's why, when reading only the abstract, this sentence is confusing, especially the term "additional". It is not clear from which dataset the Copernicus Hub and GPOD provide additional information.*

We agree with the suggestions and will move the introduction of the processing platform names to line 4 and clarify the later statement, so there is no confusion: the datasets refer to their respective processing platforms. We will also clarify that both datasets provide unique VS, where useful data can only be extracted from one platform due to various processing options.

*Line 8: Give the meaning of RMSD acronym.*

Will be changed to RMSD (Root Mean Square Deviation)

*Line 17: I have some doubts about using S3A/B as a SWOT surrogate. SWOT will do quasi global observations over two swaths, providing not just WSE, but also water extent and surface water slope, which could not be derived from S3A/B only. Besides, the temporal/space resolutions are coarser for S3A/B.*

This is a reasonable concern and we propose to remove the sentence from the abstract and instead highlight that the S3A/B constellation provides a useful and unique spatio-temporal coverage of wetland WSE with important implications for future hydrology-oriented missions.

*Line 18/19: This sentence is quite general. Similar conclusions were also reached in Jiang et al. (2020) for rivers in China. Besides, in the submitted manuscript, there is no comparison with other mission that does not have SAR mode. So it is difficult to conclude from this manuscript only that SAR mode brings more information than mission with LRM mode.*

We suggest modifying the part of the final sentence of the abstract referencing the SAR instrument, to instead reflect the reformulated objectives of the study: extracting a uniquely dense Sentinel-3 WSE monitoring network at catchment scale and the importance of

considering the pros and cons of the processing options on publicly available data processing platforms.

*Lines 27-30: References provided here correspond to only few studies linked to these subjects. That's why I suggest to add "e.g." before the references in brackets.*

We agree.

*Line 36: Getting "up-to-date" reference for the databases is very difficult (for example Cretaux et al., 2011 corresponds to the old "lake" version of Hydroweb). To overcome this issue, you could rather point out to the web link for each database. It's just a suggestion, so I let the authors decide if they want to do that or not. There are other altimetry databases than the ones cited in this sentence, like HydroSat (http://hydrosat.gis.uni-stuttgart.de/php/index.php) and GRRATS (Coss et al. 2020, https://doi.org/10.5194/essd-12-137-2020; https://podaac.jpl.nasa.gov/dataset/PRESWOT_HYDRO_GRRATS_L2_VIRTUAL_STATION_HEIGHTS_V1). And for lakes, there is the G-REALM database (https://ipad.fas.usda.gov/cropexplorer/global_reservoir/).*

Thank you for the suggestion and the additional databases – we agree that the references are not up-to-date, we will add the links as suggested.

*Line 47: For S3 mission, you should rather cite the S3 mission requirements document (S3 MRD), available at http://esamultimedia.esa.int/docs/GMES/GMES_Sentinel3_MRD_V2.0_update.pdf, rather than Jiang et al. (2020).*

Yes, the citation will be corrected.

*Line 44: "Sentinel-3 mission is a marine and land mission" This sentence is of course true, but it could give the feeling that both ocean and land requirements are considered equally, which is not the case for the altimeter part of the mission. Indeed, it is worth pointing out that for the topography component of S3 mission "Altimetric gauging of river and lake water levels is a secondary mission objective [. . .]. This requirement shall not compromise the ability of the altimeter to meet the primary ocean and ice topographic mission objectives." (section 4.4.2 in S3 MRD).*

Thank you for pointing out this detail – we propose to change the text to reflect this. Mentioning the effort put into updating the OLTC hydrology targets may be a more important point to be made here (e.g. by citing the paper by LeGac et al. suggested further down in the comments).

*Line 62: "To allow continuation of the historical ERS/Envisat time series, the Sentinel3 orbit is similar to the orbit of Envisat" This sentence is confusing, as S3A/B cannot continue VS from ERS-1/2 and Envisat, as the orbit and its phasing is not the same. You can argue that S3 provide more spatial sampling than some other missions (e.g. Jason series), but it is not a direct continuation of previous ERS and Envisat ones.*

Indeed, the statement can appear unclear. In the mission summary, it is stated: 'The mission provides data continuity for the ERS, ENVISAT and SPOT satellites'; however it is true that the altimeter does not provide a direct continuation of the ERS/ENVISAT VS on land. We propose to

modify the text to clarify this statement and include the comment about the higher spatial density compared to the Jason series.

*Line 64-67: These sentences are confusing for people who know nothing about the OLTC. It should be clearly stated that the "on board Hydrology Database (HDB) targets" is part of OLTC table. It should be introduced earlier, in the OLTC description section.*

We propose to reorganize the text to introduce HBD earlier with the OLTC as suggested.

*Line 137: Could you provide more information on this receiving window? The explanation provided in the current manuscript is interesting, but it is still difficult to understand clearly what this receiving window is and why it is needed. In the manuscript, it is written that it should "not to be confused with the on board reception window", but it's not clearly defined. It is important to better explain it for readers not familiar with SAR altimeter processing (and even more, for those not familiar with altimetry at all).*

We propose to expand this section and the altimetry processing section to better differentiate the two. To avoid confusion, we will use "range window" to refer to the on-board reception window.

The range window is the vertical window that is recorded by the altimeter whereas the receiving window is the matrix within which the pulses are temporally stored prior to processing. Shifts in topography may mean that the 128 bin radar window cannot store the elevation of all the samples in the echogram. Using a larger window to store the range samples ensures that the signals of all the echoes can be stored inside the same radar window and that the leading edge (which is later retracked to obtain the WSE) is not truncated. Examples and further details can be found in Dinardo et al. (2018 - Advances in Space Research 62 (2018) 1371–1404).

*In section 2.3.3, please cite briefly the corrections taken into account in the two datasets.*

This will be added. In the Scihub dataset, the files contain the "corrected altimeter elevation from OCOG (ice-1) retracker" and only the geoid needs to be subtracted. In GPOD the instrumental corrections are applied already and only the geophysical corrections need to be handled and they are already aggregated. The corrections include:
- Instrumental corrections: USO drift correction, internal path correction, distance antenna-COG and Doppler-slope correction
- Geophysical corrections: GIM-derived ionospheric correction, model dry tropospheric correction, model wet tropospheric correction, solid earth tide height, geocentric pole tide height and ocean loading tide.

The corrections are also provided individually, but the formats above greatly simplify the task for the less experienced user.

*Line 170: According to Jiang et al. (2020), the RIP is in Watt, so please indicate the unit in "(>10ˆ-13)"*

Correct.

*Lines 174-175: Could you provide some estimates of the two DEM errors (provided in the DEM reference paper or in the DEM quality matrix, for the Zambezi basin). It would help the reader to assess if the 30m threshold is much above the DEM accuracy.*

The two DEM have the following expected errors:

⇨ MERIT DEM – 2 m or better vertical accuracy and based on SRTM (16 m – RMSD around 9-10 m)

⇨ ACE-2 – better than 16 m but large regional differences. Considering the quality filter, ACE-2 has an accuracy better than 10m at over half of the VS in the Zambezi basin. MERIT DEM is expected to have a 2 m or better vertical accuracy.

*Line 178: According to Jiang et al. (2020), the fit is a gaussian fit, isn't it? It could be worthwhile to mention it (and maybe to add a sentence to explain why the fit is needed).*

In this paper, we calculate the Stack Peakiness using the maximum and mean RIP, therefore no fit is applied to the RIP beforehand. We will remove "fitted" from the text too. Indeed, in Jiang et al. (2020) a Gaussian fit was used to smooth the 64 sample RIP. We did not do this here as the fitting is not very crucial, and we do not use the SP to classify the data only to describe it.

*Line 195: "Retrieving the untracked range gives an assessment of whether the expected WSE was within the on board reception window", I agree, but this statement is very general and you could better explain how you will use this information. If you don't know the expected WSE (which is the case for 169 of your VS), I don't see how you can really make use of this information. Will you compare it to DEM?*

We propose to expand this sentence slightly to better introduce the use of the untracked range. It allows us to track how the untracked range changes when new targets are uploaded and to identify whether the OLTC is at the source of problems with the data.

We use the DEM as a reference, but most importantly, plotting the untracked range can help explain why some VS fail. If we are very far from the DEM it is unlikely that the target was sensed at all. Of course there is a risk in some cases that the DEM is so wrong (e.g. due to dam construction) that we lose information due to this filtering. We do not expect this to be the case in the Zambezi basin. With regards to the untracked range, the examples where we see large errors, the untracked range is off by far more than the expected DEM error.

*Line 211: "WRMSD (Weighted RMSD) by dividing with the residuals with the in-situ standard deviation" it not clear, please rephrase Equation 3, to be coherent with the text change D_{RMS} with RMSD*

Indeed, it should be: "WRMSD (Weighted RMSD) by dividing the residuals with the in-situ standard deviation"

Equation 3 was modified in line with HESS recommendations to avoid abbreviations in equations – for clarity and because it is a widely used abbreviation, we prefer to retain WRMSD/RMSD in the text body but will use D_{RMS} when referring to the equation.

*Line 217: "We correct for datum shifts by using the WSE amplitude and therefore expect a bias of 0 cm." You need to provide more explanation. First, how did you use the amplitude and to compute what? Second, I don't understand why you need to correct datum shifts, as you already removed to time series "mean level at overlapping sensing dates is subtracted". So why is it needed to add any other bias correction?*

The amplitudes are used for comparison with the ground observations – the section is indeed unclear, as it is the same bias correction mentioned in two different ways. This will be rephrased for clarity. We only correct the bias once by subtracting the mean WSE at overlapping sensing dates.

*Lines 244-251: All the criteria used by the authors are not easy to follow, as they depend of the dataset and the product level. It should be better explained in the methodology section, with a clear flowchart of the process and a more in depth explanation of all the criteria used.*

Thank you for the suggestion – we propose to add the following flowchart to the paper to illustrate the processing steps, and to reorganize the methods section accordingly. For the particular section, we suggest to already mention in the methods that the NP is used as a selection criteria. The criteria are the same across datasets, however MP and SP are not calculated for the SciHub dataset as the RIP is not available.

[Figure]

Figure 1 Data selection and processing flow chart. Highlighted in red are the output products used to assess the Sentinel-3 performance in the basin.

*Line 250: "we use NP alone as the L1b selection criterion" but how did you use NP? Using which threshold? It is somewhat difficult to understand why NP is a good criterion, as multiple targets could be in the waveform and it does not mean that the altimeter is not observing the target of interest. It is especially true for small tributaries, where there are a lot of missing data on Figure 2. The missing data could also be due to your criteria. Could you discuss it in more details?*

As mentioned above, we will clarify section 3.1. Indeed multiple targets could be in the waveform, however that increases the risk of retracking errors if there are multiple high power targets. The missing data in Figure 2 is not due to this criteria as it is only based on what WSE observations could be extracted after corrections, water mask selection and filtering.
We base the evaluation on the along-track average number of peaks and select the 90th percentile as the selection criteria, meaning at least 90% of the data should come from single-peak waveforms. We did notice some small errors in the number of stations using this criteria. The error affects the number of valid stations but not the conclusions.

We also checked all rejected stations and found the following cases:
- o Most have very little valid data at all, or several outliers (rejected on criteria of 80% data should be available). For some of these stations, dedicated (most likely manual) processing could help retrieve information if they were located in areas of interest
- o Data loss due to OLTC update
- o A few stations have seasonal water observations – but with a very wide spread – this is the case for VS on narrow river targets in wetlands.
- o The stations rejected based on the single peak criteria mostly have very large across-track standard deviations, suggesting it is not unlikely that the waveform is contaminated by other bright targets and justifying the rejection of the VS.

Some of the retained VS might also require some degree of manual validation or outlier removal, however the proposed filtering greatly reduces the task (> 200 VS to check versus just over 100). It also allows users to group VS that they wish to further inspect and validate and to provide tools for pre-selection and evaluation. We propose to summarize this information in the revised manuscript.

*Line 254: "The rejection rate is higher in the SciHub dataset, with rejected stations throughout the basin." This sentence seems to meet the concern expressed in my previous comment.*

Several of the stations rejected had missing data in the GPOD dataset because the retracker failed to fit a model waveform to the observed waveform, suggesting the target is not a good water target. The OCOG retracker is less sensitive to the shape of the waveform. The higher rejection rate balances this.

*Table 2: What is the line "OLTC" in Table 2? It is not explained in the table legend, nor in the text.*

We will modify the line in the table: it is the stations with data only after the OLTC update in March 2019.

*Figure 4: This figure does not seem useful, except to state that after OLTC update there is mainly 1 peak in the waveform. But as the NP before the update is not provided, it is difficult to estimate the improvement.*

> We suggest removing the figure and state in the text that the OLTC update also improves the NP statistics (as shown already in Table 2).

*Lines 266-267: "The OLTC contains targets based on elevation information from hydrology databases (e.g. Hydroweb), virtual stations networks and the global ACE2 DEM (Altimeter Corrected Elevations v.2 Digital Elevation Model)" Actually it depends of the OLTC version you are considering, as stated later in your paragraph. According to https://www.altimetry-hydro.eu/ here are the different OLTC table versions over inland waters: - For S3A: \* DEM: v5 (Date start: 2019-03-09) \* DEM: v4_2 (Date start: 2016- 05-24, Date end: 2019-03-01) \* DEM: v4_1 (Date start: 2016-04-18, Date end: 2016- 05-24) - For S3B: \* DEM: v2_0 (Date start: 2018-11-27) \* DEM: v1(tandem) (Date start: 2018-06-06, Date end: 2018- 10-16) Especially, on the https://www.altimetry-hydro.eu/ you can see that ACE2 DEM is heavily used in v4_2 for S3A, but not used at all in v5 over the Zambezi basin, as shown on Table 3 but not clearly stated in the text. Besides, at line 268 and in other part of the manuscript, it is written that the table has been updated in March 2019. It is true for S3A, but not for S3B, which has been updated sooner (after the end of the tandem phase in November 2018). The OLTC versions are given in Table 3, but never really explained in the text. A good reference for OLTC tables' generation is (with some validation): Le Gac S., F. Boy, D. Blumstein, L. Lasson and N. Picot (in press). Benefits of the Open-Loop Tracking Command (OLTC): Extending conventional nadir altimetry to inland waters monitoring. Advances in Space Research, https://doi.org/10.1016/j.asr.2019.10.031 I think putting a table to summarized all these OLTC versions and dates could be useful in the manuscript, with some information on OLTC generation (see Le Gac et al., in press). These information should be put somewhere in section 2.*

> We thank the reviewer for the citation suggestions – we will incorporate them in the methods section as suggested. The update indeed only refers to the Sentinel-3A OLTC as the Sentinel-3B update as made prior to the beginning of the datasets considered. We will make sure this is clear in the manuscript.

> Indeed the targets mainly consist of HDB targets after the update – however they still rely on high resolution DEMs – as we understand, ACE-2 is still used to define many hydrology targets. We are very interested in further information if other high resolution DEMs are used instead of ACE-2 for the HDB targets.

*Figure 5: On the map, the black line (sub-basin boundaries?) are not defined, does not seem to be useful and make the map difficult to read. I suggest removing them. Where they are close, S3A and S3B VS are difficult to differentiate. Maybe use different color or level of grey between the two missions. On the sub-plots, write when it is S3A or S3B. In the legend, write to refer to figure 2 for the location of the map within the Zambezi basin (blue polygon on figure 2).*

> Agreed – we will remove the lines, increase the difference between the two mission markers and refer back to Figure 2.

*Line 280: "no new targets were uploaded to the OLTC in March 2019 near the two S3A VS" Just to be sure, even if no new targets has been added in march 2019 near these VS, it does not mean that the*

*OLTC table has not been updated in March 2019 for these VS. Is it the case? From figure 5 even if it is the case, the updated value should be pretty similar, as the time series seems pretty stable before and after March 2019.*

> Based on the online OLTC webpage, the existing targets were only updated with no significant change in height, suggesting there is no point in splitting the time series in before and after the OLTC update.

*Line 288: "Samosa+ retracker outperforms the OGOC retracker", first replace OGOC with OCOG. Second, from this sentence, I was expecting much better results with Samosa+ than with OCOG, whereas on table 4, SAMOSA+ is better only by few cm (or %, even most of the times few tenth of %). So I would encourage the authors to add this quantitative information to alleviate this sentence. Besides, Samosa+ comes from GPOD, whereas OCOG comes from SciHub, and processing between these two platforms are different (not just the used retracker, but also the data selection and probably other processing, corrections...) as described on sections 2.3.1 and 2.3.2. How these differences could impact the results shown on Table 4?*

> Thank you for pointing this out. The difference is indeed only slightly better – we agree that the sentence could be misleading. We will nuance the sentence with the range of improvement from using Samosa+ versus OCOG and instead refer to GPOD and SciHub as the L2 WSE is compared in terms of platform, as it is not necessarily the retracker that drives the difference in performance. Because we use the processing options available on the two platforms, comparing the full processing packages and not the retrackers alone makes most sense in light of the objectives of the study.

*Table 4: For Chavuma station, Samosa+ RMSD is equal to 15.8cm and 3.3%, whereas for OCOG the RMSD is 25.6cm and 3.6%. How an almost 10 cm difference in RMSD between Samosa+ and OCOG only translates into 0.3% increase? I think there is an issue with the % computation (or with the RMSD value). Besides, the 9th column entitled "Relative RMSD" corresponds to WRMSD in the text, please replace "Relative RMSD" with "WRMSD" for consistency.*

> Indeed, there was an error in the table as the values for Chavuma and Ngonye Falls were exchanged – thank you for pointing this inconsistency out. The correct RMSD relative to the yearly amplitude is 5.3% at Chavuma and 3.6% at Ngonye Falls.

*Line 315: "If we consider the stations, which are valid across datasets", how do you define "valid" here? Could you recall the criteria here?*

> We will add detail and nuance the term "valid" – we consider the stations with single peak waveforms and a low degree of missing data as more reliable than those with multi-peak waveforms and a high degree of missing data.

*Line 316: "The number of VS is quadrupled compared to using the global database Hydroweb", it is impressive. However, it should be noted that all Zambezi VS on Hydroweb have an "expert validation criteria" (see http://hydroweb.theia-land.fr/?lang=en&basin=ZAMBEZI and https://theia.sedoo.fr/wpcontent-theia/uploads/sites/2/2020/04/Handbook_Hydroweb-V2.0-1.pdf). Are all the 145 VS being individually checked and validated (coherent seasonal cycle and amplitude from upstream to downstream VS)? Coherent amplitude and seasonal cycle has been shown only for 10 VS (and compared to in situ gage data only for 6 gages) in the manuscript.*

Of course, this is why this number can be increased this dramatically at catchment scale. We are fully aware that the goal of global database can not be to provide all VS for all catchments, therefore we see a value in providing tools and lessons-learned in processing the data at catchment-scale, as highly valuable data may be available. Data is also accessible sooner when new satellites are launched (e.g. S3-B,) allowing faster uptake of the data.

*Line 320: "At four stations in the Upper Zambezi, there are no valid observations at any of the VS prior to the OLTC update (Fig. 8)" I don't see how figure 8 shows that there is no valid observation before OLTC update, as figure 8 is only showing data after (S3A) OLTC update.*

The figure shows no data as no data could be processed before the update: i.e. due to no-data values, or too far from the DEM or very low backscatter. We will clarify in the text.

*Figure 9: Concerning the zoom on the WSE vs. latitude plot (between -12.01°N and -11.81°N), it might be because of the color code, but it seems "after Schihub" WSE is in between 1050m and 1100m, whereas "After (GPOD/3x" WSE is in between 1000m and 1050m and "After (GPOD/2x" WSE is below 950m. I don't understand why there are not more consistent. I understand it is near the transition, which affect GPOD when changing the receiving window, but why is it that different, especially why "GPOD/3x" is above "GPOD/2x" and not below (by tripling the window, you should have more data after the transition)? Besides, on the WSE vs. latitudes plots, I would suggest to draw all "Before" curves with dashed lines, to make them easier to differentiate with the "After" curves.*

We suspect it is due to too much weight being given to the next target thus over-smoothing the transition. The algorithm as we understand is developed for closed-loop mode, where you want a faster transition, which is what happens. In open-loop the fast transition already occurs and the algorithm introduces an artificial transition. We have been in contact with GPOD who confirmed that the results were as expected. The performance at this VS explains why processing options are significant at certain locations.

Thank you for the suggestion for the figure – we will redraw the "before" curves.

*Line 350: "According to the OLTC website," please give the URL of this website (I guess it is https://www.altimetry-hydro.eu/)*

Indeed – it will be added.

*Section 4.2, which VS is considered here? A86 VS on figure B1? Besides, it to better see the impact of the platform processing versus the OLTC value, it could be good to show the OLTC table value (i. e. position of the tracking window), rather than the retracked value converted to WSE. It would help to show the transition and why you need to extend the receiving window with GPOD and then you can discuss the difference between the two platforms.*

We will add detail about the VS considered. We noted in this context that a number of VS where erroneously left out from, so a slight renumbering is necessary. The shown height is the tracking window position – we will correct the y-axis label accordingly.

*Lines 370-372: The second option is the one chosen during the March 2019 update, isn't it?*

Actually, both options were chosen – the target was defined earlier, but for the GPOD dataset, an extension of the receiving window is still necessary.

*Line 389: "This is likely due to the frequent cloud cover over the floodplain." Or maybe due to vegetation cover masking water?*

This is a very good point – yes.

*Line 412: the references provided here are just examples of studies using altimetry to calibrate and update hydrology model, so I suggest putting "e.g." before the references.*

Yes.

*Lines 423-428: There is not just Park (2020) and your study which investigated connectivity between river and floodplains. Could you increase your references list?*

The citation is a single example of course, we propose to increase the reference list, including by adding a paragraph in the introduction better showing this.

*Line 429: "The cross-sections extracted over floodplains are similar to observations expected from the future SWOT" I disagree with this statement. Even if Sentinel-3 mission provides much more spatial observations than other altimetry missions (like Jason series), it is not comparable to SWOT measurements, which will provides images of WSE. So rephrase this sentence accordingly.*

This is a fair point and in light of other comments about the reference to the SWOT mission, we propose to instead focus the discussion on the benefit of the spatio-temporal coverage of the dual-satellite constellation for hydrological applications.

*Line 430: Concerning SWOT mission, I think a better reference for the reader will the SWOT Science Requirements Document (SRD) rather than Domeneghetti et al. (2018). SWOT SRD could be accessed with the following link:*
[https://swot.jpl.nasa.gov/system/documents/files/2176_2176_D61923_SRD_Rev_B_20181113.pdf](https://swot.jpl.nasa.gov/system/documents/files/2176_2176_D61923_SRD_Rev_B_20181113.pdf)

We will correct.

*Line 432: "Similar information can already be extracted from the Sentinel-3 dataset in selected locations" Similarly to my previous comment, I think this sentence should be rephrased. S3 is providing WSE, but not water mask and of course slope could be computed between close VS, but it is far from being the one expected from SWOT images. . .*

Same as above.

*Lines 445-446: "We extract over 360 virtual stations from each satellite of which over 70 are validated based on the waveforms and temporal coverage for each Sentinel-3 satellite" Why stating this in the conclusion and not in the core of the manuscript? In the abstract 170 VS are mentioned. The same goes for the 70 validated VS.*

We state it here as a concluding remark. The 70 stations are for each mission, totaling 72 S3B stations and 73 S3A stations which pass all criteria across datasets. The 169 stations are stations which pass the criteria in one or the other dataset. We will ensure that a more consistent reference to the number of stations is done in the revised manuscript.

*Section 4.4 and 5: I find it strange to have perspectives before conclusions...*

In light of other comments in addition to this one, we propose rewriting this section and the conclusion in order to ensure perspectives are placed after the conclusion.

---

## Author Comment (AC2) · 3 Aug 2020

Replies to the comments by anonymous referee #2:

We would like to thank the reviewer for their interest and comments on the manuscript. Below the reviewer's comments are in italic and the replies in normal font.

*The paper describes the computation and exploitation of satellite altimetry water level time series in the Zambezi basin. According to the authors, the aim of the study is "to assess the potential of the Sentinel-3 mission in hydrological applications". For that purpose, they compare different satellite altimetry pre-processing options (from two different databases) and they analyze the impact of open loop processing. Moreover, a validation by comparison with (few) in-situ ground stations is performed. For three different wetlands within the study basin, the potential of Sentinel-3 for monitoring the interaction of river and floodplain is shown.*

*General comments: This is an interesting topic worth publishing. However, some aspects of the paper are not innovative and had been published before by some of the same authors (e.g. OLTC impact by Jiang et al., 2020). Moreover, some parts of the manuscript are quite technically without providing the (less experienced) readers a recommendation on which processing option to use. In my opinion, the most interesting and innovative part of the study is the approach of automatically processing all possible VS of both satellites in the entire basin with the aim to use these time series for assessment of wetland-river interactions. Thus, my recommendation would be to focus on this part of the study by adding a bit more statistics (how many potential VS, how many valid VS, how many VS gained by OLTC,. . .) and some citations of existing work on wetlands based on satellite altimetry (e.g. Zakharova et al., 2014; Dettmering et al., 2016; Park, 2020). In addition, a (at least theoretical) comparison to classical missions can be added discussing the benefit of the dual satellite constellation (with respect to spatial and temporal coverage) and the measurement mode (SAR/OLTC).*

> We thank the reviewer for the interest in the manuscript and the comments. We agree on the analysis of the major contribution and propose to better highlight this in the introduction. We also propose to add a flowchart to the methods section, allowing a better overview of the different processing steps for reproduction. Finally, we plan to publish the Zambezi network and generic processing tools together with the final manuscript.

> In terms of the statistics requested, we will develop the text a bit more to better highlight Table 2 (which contains the requested numbers) and include basin-scale summaries where these are missing.

> The suggestions for additional citations will be added in text. We thank the reviewer also for the suggestions for the discussion, which we will incorporate in the discussion.

*Specific comments:*

*Line 5: If the objective of the study is to "evaluate the density of valuable observations", you should add some more statistics on the number of VS (see general comment).*

The objective is to show the value of processing Sentinel-3 at catchment scale, illustrating the performance for the Zambezi basin. We will add the requested statistics from the general comment.

*Line 18: In my opinion, the paper is not showing the benefits of SAR (with respect to what? LRM?). The denser track network is due to the orbit configuration not the measurement mode, and there is not comparison to LRM data. The RMSD values are similar to those from LRM missions. So, how is the benefit demonstrated?*

The results support the progress in results observed in past papers as well, where the RMSD is lower with SAR missions than LRM (e.g. in comparison to Michailovsky's results with Envisat). Direct comparison is difficult due to the lack of overlap in space and time.

We propose to instead highlight the benefit of the spatio-temporal sampling achieved by the dual-satellite constellation and orbit.

*Line 44: Sentinel-3 is not only an ESA mission => Copernicus*

Thank you for pointing this out – indeed, Sentinel-3 is part of the Copernicus program and the mission is developed by ESA in this context.

*Section 2.2: What about adding additional information on in-situ validation observations and OLTC targets?*

We will add two separate sections for the in-situ stations (taken from 2.3.6) and the OLTC table, thank you for pointing this out.

*Line 104/105: Please add some more information on the stream burning. I'm not sure what is meant here.*

This is part of the data processing for the river network database, as detailed in the cited paper.

We selected the dataset from Yan et al. (2019) because they defined the river networks globally (thus the same dataset can be used for other study cases) and because in addition to a river delineation algorithm, they burnt in a river line to the DEM, increasing the accuracy of the river location, particularly in flat areas.

*Section 2.3.3: Some detailed info on the corrections is missing (e.g. which models).*

This will be added. In the Scihub dataset, the files contain the already "corrected altimeter elevation from OCOG (ice-1) retracker" and only the geoid needs to be subtracted. In GPOD the instrumental corrections are applied already and only the geophysical corrections need to be handled and they are already aggregated. The corrections include:
- Instrumental corrections: USO drift correction, internal path correction, distance antenna-COG and Doppler-slope correction
- Geophysical corrections: GIM-derived ionospheric correction, model dry tropospheric correction, model wet tropospheric correction, solid earth tide height, geocentric pole tide height and and ocean loading tide.

*Line 171: Sigma0==backscatter?*

Yes, it is the backscatter coefficient

*Section 2.3: I recommend to provide also the web addresses of GPOD and SciHub (in the text or alternatively in Refs or Acknowledgements.*

Good point.

*Line 174/175: Are these DEMs good enough to be used in this context. My personal experience is that at least ACE2 includes really large outliers in some regions.*

ACE-2 has an accuracy of 5-10 m at most VS in the basin (and almost always less than 16 m). Indeed the choice of DEM might bias the selection. The +/- 30 m window should not be a problem.

*Line 200: "are processed" => how? Median/mean*

Reformulated: "All observations retained at a given virtual station are processed by taking the along-track mean to produce a WSE time series."

*Line 204: "six". Where are these stations located. Maybe you can reference to a figure.*

Indeed the locations are not presented until Figure 5 – we can add the stations to the catchment basemap.

*Line 210: RMSD or D_{RMS}? Please make consistent*

$D_{RMS}$ is used in accordance with HESS guidelines for equations – we can streamline this in text – we do think that RMSD is a frequently used acronym justifying using RMSD elsewhere in the text.

*Line 231: "two the" => "the" or "the two"*

"the two"

*Figure 2: I can't find any black cycle in the plot. On the other hand blue lines (which I assume to be rivers) not covered by data. The black lines are a bit confusing here. I guess these are sub-basin borders. Please indicate or remove. The additional maps seems to be in the Annex, not in the supplementary material.*

The reference to the additional maps should indeed be the Annex. We agree that the subbasin borders do not carry significant information in this case.

Thank you for pointing out that the circles were missing. The map below shows where the stations with no data are located. There are parts of the river, which fall between tracks and are thus not sensed by either satellite.

[Figure]

Figure 1 Zambezi Sentinel-3A and 3B VS after outlier filtering. Stations which improved by modifying processing steps (either on board through the OLTC update or on-ground by extending the receiving window on GPOD) are highlighted separately. The frames indicate examples highlighted in the next sections of this study. Additional maps are included in the supplementary material (Fig. A1, A2 and A3).

*Title of 3.1: This is quite technically. What about using a title indicating the aim of this section, e.g. comparison of different L1b pre-processing*

> We propose to change the title to "Evaluation of VS at Level-1b level" – the section does not compare the L1b pre-processing but rather evaluates whether the L1b data supports that the VS are good inland hydrology targets.

*Figure 3: What does OLTC stands for here (black and orange)? Before/after OLTC update? Please clarify.*

> In legend: OLTC indicates Sentinel-3A stations where observations are only available after the OLTC update in March 2019.

*Table 3: Please provide the sum over the entire basin. Include description of GPOD/SciHub version for VS no (I guess it should be 2x, 3x?)*

> We propose adding column titles in the table to avoid confusion so the different versions are clear for the two databases and so that the information is available at basin scale as well.

*Section 3.3.1. What about adding a discussion on the impact of low number and distribution of the validation sites. Are the validation numbers representative for the entire basin?*

> Of course the validation is limited by the low number of validation sites. However, section 3.3.2 confirms that the hydrological patterns are reliable in other parts of the basin as well. This pattern of data availability is also why S3 holds high value in a catchment with low gauging density.

> We propose to mention in the text that the validation is mainly focused on the Upper Zambezi where data was made available and thus the full dataset cannot by extension be considered valid. However, the stations are located on rivers of very varying width (quite representative of the basin in general), which is encouraging in spite of the stations being close to each other.

*Figure 5: in-situ (black) lines are not visible. Are they always available for the whole period? Are there more than one observation available per epoch (=> single alongtrack measurements instead of mean/median?) Can you add RMSD here?*

> We will try stippling the S3 lines to make the underlying black lines visible. The in-situ observations are available until April 2019 (we will add the time of observation to Table 4).

> In some cases, there are more than one observation – indicated by the points – whereas the line indicates the mean WSE, which is compared to the in-situ observations. The RMSD is given in Table 4.

*Line 288: OGOC => OCOG*

> Thank you for pointing this out.

*Table 4: is the difference only due to the retracker? Might the pre-processing play a role? Is the Relative RMSD == WRMSD?*

> Indeed the pre-processing might also play a role, although both are intrinsically linked to the processing platform and to each other. For more clarity, we will refer to the datasets by the platform rather than the retracker here.

> Yes it should be WRMSD.

*Figure 6: I can not find any orange lines here. . .*

> There are indeed no observations from those decades – we will remove the orange lines from the legend.

*3.3.3: What about adding some more information and interpretation here.*

> We propose to add a discussion linking back to the in-situ stations discussed in the two previous sections, which indicate annual amplitudes in the order of 5-10 m. Furthermore, Figure 7 provides a summary of the Sentinel-3 observations, suggesting that in some cases further manual validation might be necessary, i.e. to remove large outliers or confirm that the patterns are hydrologically consistent.

*Line 323-324 (and in some other parts of the manuscript): I'm not sure whether it is fair to compare with global WSE databases. Since these databases aim in providing input for hydrological research, the focus is on long time-series. For sure, they are also able to process these VS - however, this has no priority given the short time series of less than 2 years.*

> The comparison should be seen as an encouragement to explore the public processing platforms, which provide access to the full Sentinel-3 dataset, beyond what is available on the databases. The databases provide an excellent starting point, however, at catchment scale (including for smaller rivers) or where short time series would have useful applications there may be more information available. This paper illustrates how much additional data can be obtained through automatic extraction from the full dataset.

*Line 342: Is there any statistics available on the percentage of improvement/degradation by OLTC in this region?*

> We are not sure we understand the question – to obtain statistics a simultaneous closed-loop mission would be necessary. What we do see is cases where the time series stops after the update and a loss of data due to the time lag between mission launch and table update. This is quite significant as large amounts of data are potentially useless when the OLTC is not up to date.

*Line 349ff: "mamsl": all other heights are provided with respect to a geoid. Why not these ones? At least you should explain the abbreviation.*

> The elevation is from the OLTC database (altimetry-hydro.eu) and thus actually relative to a geoid. We will correct the unit to avoid confusion.

*Line 369: options to mitigate: Do you have any recommendation for the users? What preprocessing should I use?*

> This is a tricky question with no clear single answer. In some cases, the dedicated inland water options outperform the standard processing (as would be expected), in others they appear to actually worsen the results. The take-home message is that the choice of preprocessing does indeed matter and based on the virtual station and its location it might be worthwhile to consider several.

*Figure 12: Is there any color change in c) and d) depending on waveform misfit?*

> Indeed – in this particular case, the misfit is generally quite low with no significant change, making the misfit information superfluous.

*Line 385/386: Are there no unique track numbers?*

> The given track numbers are the relative track numbers – and all data will belong to those same tracks. Alternatively, there are absolute track numbers, however the point made here is that the dataset will contain two data groups.

*Figure 13: What are the vertical blue lines in crossing tracks 741 and 498? Where are the VS located for tracks 498 and 085? What are the stars and cycles in the left hand plot?*

The vertical blue lines are the water occurrence (we will add this to the figure caption) as seen in the basemap on the left. The VS are the cycles and stars in the left hand plot and are indeed missing from the legend.

*Figure 15: left and right?*

Left and right are erroneous in this case and will be removed.

*4.4 This is more a summery than perspective. . . Moreover, perspective should be placed after conclusions. . . Line 409: "first" => where is second?*

We propose to rewrite this section and the conclusion, in order to ensure perspectives are placed after the conclusion.

*Line 429-434: Please reformulate this paragraph: SWOT will provide much more information than S3, especially in cross-track direction. Also CS2 can already extract similar information in selected locations.*

Agreed – we will reformulate to better highlight the point.

*Line 441: "should"? => is or is not improving!*

We suggest reformulating to "is expected to improve" – the point addressed here is the expectation to OLTC vs. closed-loop rather than the conclusion of this study.

*Line 446: I don't think that you should name that a "validation"*

We propose to refer to the selection process as an evaluation of the data at multiple levels.

*Line 447: Again: My feeling is, that this is not a fair comparison. Hydroweb is a global database not aiming in complete coverage of entire basins.*

We completely agree that a comparison would not be fair – our intent with the comparison of the number of VS is to highlight the benefit of extracting data beyond what is already available on such databases, when looking at altimetry data at catchment scale. This is important for regional to local hydrological studies. Furthermore, we ease the WSE data retrieval for hydrologists. By showing the numbers together, we hope to encourage interested users to also consider the full dataset on publically available processing platforms.

*Line 452-458: I suggest shifting this paragraph to line 443 (as second paragraph of this section). This would make the paper end with the application, which is the overall focus of your paper according to line 69.*

Thank you for the suggestion – we will shift the paragraph.

---

## Author Response (AR1)

**Replies to the comments by anonymous referee #1:**

We would like to thank the reviewer for their interest and comments on the manuscript. Below the reviewer's comments are in italic and the replies in normal font.

The manuscript from Kittel et al. presents a validation study of Sentinel-3A/B (S3- A/B) SAR altimeters measurement over the whole Zambezi basin. Time series at 175 virtual stations data have been extracted on the river network, floodplain, reservoirs and wetland. Only 6 in situ gages could be used to validate this database, showing a RMSD between 3 to 31 cm. However, no direct validation can be done for the remaining 169 VS and especially over wetlands (except that the seasonal cycle is well captured and coherent with past in situ observation or nearby VS). Some discussions on the benefits and drawbacks from 1. the open loop tracking mode and 2. the different processing available on two platforms (SciHub and GPOD) to process S3 data complement the manuscript.

General comments: It should be noted the important work done by the authors to extract this unprecedented database of WSE time series over the whole Zambezi basin and the interesting discussion on the open loop mode and the SAR altimetry processing. However, the authors should better highlight the new discovery from their manuscript and why it has to be published in HESS and not in a more specific remote sensing journal. This is my main concern and the reason why I suggest major revision. As stated by the authors: "The objectives of the study are to evaluate the density of valuable observations and establish a WSE monitoring network. Additionally, we demonstrate the potential application of Sentinel-3 for monitoring river interactions with wetlands and floodplains." The issue is that validation and discussion on SAR and open loop mode have already been done in Jiang et al. (2020) over rivers in China. The submitted manuscript confirms some conclusions from this paper over another basin, but does not bring new information concerning S3 measurements, nor on the hydrology of the Zambezi basin. The application of radar altimetry for monitoring interactions between river, floodplains and wetlands has already been investigated by other studies with different radar altimetry missions. Another previous study from this group (Michailovsky et al., 2012), also studied the Zambezi basin with the Envisat radar altimeter and derived discharges from these WSE with different methods. The main benefit of the submitted manuscript is the important database of WSE over the Zambezi basin derived from S3 missions. So, according to me, the submitted manuscript is a database presentation paper, but the database does not seem to be freely accessible, like other global altimetry database (e.g. Hydroweb, DAHITI...).

We thank the reviewer for the summary and general comments about the manuscript.

Indeed the validation is challenged by the lack of concurrent in-situ data and only 6 in-situ gauges were located close enough to VS to ensure direct validation of the satellite performance. We expanded the validation by considering the hydrological patterns at additional stations with historical records. However, extracting the full Sentinel-3 dataset has the largest potential value in supplementing ground observations in poorly gauged catchments.

As the reviewer correctly points out an important aim of the study is to demonstrate the extraction of a catchment-scale WSE monitoring network from Sentinel-3 observations. To address these main concerns, we rewrote the introduction and objectives of the study to more

clearly reflect this, and the discussion in the following sections has been adapted accordingly (see also specific suggestions below).

The database will be publicly available in conjunction with the paper and a link will be provided in the reviewed version. The python code used for processing will also be published. The purpose is not the specific Zambezi-database (although that is an important product of the study) but rather to demonstrate that by using the publicly available processing platforms, such databases can be created for any catchment globally. We believe that a framework to extract catchment-scale monitoring networks to suit specific study areas has a wide range of applications in hydrology, which is why we believe HESS is a good target journal for publication.

**Specific comments:**

Few clarifications are needed in the abstract. For example, give the name of the two datasets the first time you mention them (line 4). Especially, the sentence "Additional VS are available in both the Copernicus Open Access Hub and GPOD", seems to suggest that these two datasets are different from the two platform mentioned on line 4, which is not the case. That's why, when reading only the abstract, this sentence is confusing, especially the term "additional". It is not clear from which dataset the Copernicus Hub and GPOD provide additional information.

We have reformulated the abstract accordingly including:

- Moving the introduction of the processing platforms to l. 6
- Clarified the number of stations referred to in the abstract on I. 9

**Line 8: Give the meaning of RMSD acronym.**

The acronym has been defined. L. 10: "The Root Mean Square Deviation (RMSD)"

Line 17: I have some doubts about using S3A/B as a SWOT surrogate. SWOT will do quasi global observations over two swaths, providing not just WSE, but also water extent and surface water slope, which could not be derived from S3A/B only. Besides, the temporal/space resolutions are coarser for S3A/B.

Indeed S3A/B could be used as a partial surrogate until SWOT launch. This has already been explored using CryoSat-2; however, Sentinel-3 operating in SAR mode and the two-satellite constellation has a good spatial resolution as well – as seen in the Zambezi – which could provide similar information where the tracks are appropriately located. Of course, the expectation is that SWOT will provide unique information compared to existing missions, but there is value in exploring existing missions as well as synthetic SWOT data in preparation for mission launch. We have removed the reference to SWOT in light of this and other comments.

Line 18/19: This sentence is quite general. Similar conclusions were also reached in Jiang et al. (2020) for rivers in China. Besides, in the submitted manuscript, there is no comparison with other mission that does not have SAR mode. So it is difficult to conclude from this manuscript only that SAR mode brings more information than mission with LRM mode.

We modified the part of the final sentence of the abstract referencing the SAR instrument, to instead reflect the reformulated objectives of the study: extracting a uniquely dense Sentinel-3 WSE monitoring network at catchment scale and the importance of considering the pros and cons of the processing options on publically available data processing platforms.

L. 18-19: "These results highlight the benefit of the high spatio-temporal resolution of the dualsatellite constellation, which holds important implications for future hydrology-oriented missions."

*Lines 27-30: References provided here correspond to only few studies linked to these subjects. That's why I suggest to add "e.g." before the references in brackets.*

L. 28-38: We agree and "e.g." has been added to the list of references where relevant.

Line 36: Getting "up-to-date" reference for the databases is very difficult (for example Cretaux et al., 2011 corresponds to the old "lake" version of Hydroweb). To overcome this issue, you could rather point out to the web link for each database. It's just a suggestion, so I let the authors decide if they want to do that or not. There are other altimetry databases than the ones cited in this sentence, like HydroSat (http://hydrosat.gis.uni-stuttgart.de/php/index.php) and GRRATS (Coss et al. 2020, https://doi.org/10.5194/essd-12-137-2020; https://podaac.jpl.nasa.gov/dataset/PRESWOT\_HYDRO\_GRRATS\_L2\_VIRTUAL\_STATION\_HEIGHTS\_V1). And for lakes, there is the G-REALM database (https://ipad.fas.usda.gov/cropexplorer/global\_reservoir/).

Thank you for the suggestion and the additional databases – we agree that the references are not up-to-date, we use the links instead as suggested.

L. 81-85: "Several databases provide global, ready-to-use and publicly available time series of WSE for inland water bodies derived from satellite altimetry observations, including from Sentinel-3 e.g. Hydroweb (http://hydroweb.theia-land.fr/), DAHITI (https://dahiti.dgfi.tum.de/en/) and HydroSat (http://hydrosat.gis.uni-stuttgart.de/php/index.php)."

Line 47: For S3 mission, you should rather cite the S3 mission requirements document (S3 MRD), available at http://esamultimedia.esa.int/docs/GMES/GMES\_Sentinel3\_MRD\_V2.0\_update.pdf, rather than Jiang et al. (2020).

We agree, the citation has been be corrected:

L. 49: "The satellites both carry dual-frequency (Ku- and C-band) Synthetic Aperture Radar Altimeters (SRAL) on board, building on the heritage of the CryoSat-2 and Jason missions (Drinkwater and Rebhan, 2007)."

L. 68-69: "Sentinel-3 is a marine and land mission, with the altimetric gauging of inland water being a secondary objective to the ocean and ice topographic mission objectives (Drinkwater and Rebhan, 2007)."

Line 44: "Sentinel-3 mission is a marine and land mission" This sentence is of course true, but it could give the feeling that both ocean and land requirements are considered equally, which is not the case for the altimeter part of the mission. Indeed, it is worth pointing out that for the topography component of S3 mission "Altimetric gauging of river and lake water levels is a secondary mission objective [...]. This requirement shall not compromise the ability of the altimeter to meet the primary ocean and ice topographic mission objectives." (section 4.4.2 in S3 MRD).

Thank you for pointing out this detail – we changed the text to reflect this and to mention the effort put into updating the OLTC hydrology targets to highlight the focus on inland water applications:

L. 68-74: "Sentinel-3 is a marine and land mission, with the altimetric gauging of inland water being a secondary objective to the ocean and ice topographic mission objectives (Drinkwater and Rebhan, 2007). However, the OLTC tables on-board Sentinel-3A and Sentinel-3B contain over 65,000 virtual stations, or hydrological targets, defined using state-of-the-art water surface masks and high resolution Digital Elevation Models. The OLTC is expected to be a key factor in establishing global databases of water level and to be integrated on future altimetry missions (Le Gac et al., 2019). It is therefore important to understand the implications of the open-loop tracking mode and interactions between the OLTC and post-processing choices on the WSE datasets."

Line 62: "To allow continuation of the historical ERS/Envisat time series, the Sentinel3 orbit is similar to the orbit of Envisat" This sentence is confusing, as S3A/B cannot continue VS from ERS-1/2 and Envisat, as the orbit and its phasing is not the same. You can argue that S3 provide more spatial sampling than some other missions (e.g. Jason series), but it is not a direct continuation of previous ERS and Envisat ones.

Indeed, the statement can appear unclear. In the mission summary, it is stated: 'The mission provides data continuity for the ERS, ENVISAT and SPOT satellites'; however it is true that the altimeter does not provide a direct continuation of the ERS/ENVISAT VS on land. To avoid confusion, the sentence has been removed, and instead focus is put on the spatial sampling as suggested.

L.75-80 : "The Sentinel-3 tracks are spaced 52 km apart at the Equator, offering a high spatial density of potential virtual stations (VS) on rivers globally, with a return period of 27 days. This is interesting when compared to traditional short-repeat missions such as the Jason mission (10 days repeat period and 315 km inter-track interval) or Envisat (35 days and 80 km) and geodetic missions such as CryoSat-2 (369 days and 7.5 km). Sentinel-3 could potentially provide a much denser VS network than Jason-2 while maintaining a relatively short return period. This creates interesting possibilities for monitoring rivers and wetlands at catchment scale."

Line 64-67: These sentences are confusing for people who know nothing about the OLTC. It should be clearly stated that the "on board Hydrology Database (HDB) targets" is part of OLTC table. It should be introduced earlier, in the OLTC description section.

We reorganized the text to introduce HBD earlier with the OLTC as suggested. The OLTC and HDB are now introduced together, and section 2.2.4 more clearly introduces the OLTC and its targets.

L.69-72: "However, the OLTC tables on-board Sentinel-3A and Sentinel-3B contain over 65,000 virtual stations, or hydrological targets, defined using state-of-the-art water surface masks and high resolution Digital Elevation Models. The OLTC is expected to be a key factor in establishing global databases of water level and to be integrated on future altimetry missions (Le Gac et al., 2019)."

Line 137: Could you provide more information on this receiving window? The explanation provided in the current manuscript is interesting, but it is still difficult to understand clearly what this receiving window is and why it is needed. In the manuscript, it is written that it should "not to be confused with the on board reception window", but it's not clearly defined. It is important to better explain it for readers not familiar with SAR altimeter processing (and even more, for those not familiar with altimetry at all).

We expanded this section and the altimetry processing section to better differentiate between the two. The on-board reception window is the vertical window that is recorded by the altimeter whereas the receiving window is the matrix within which the pulses are stored prior to processing. Shifts in topography may mean that the 128 bin radar window cannot store the elevation of all the samples in the echogram. Using a larger window to store the range samples ensures that the return power of all the echoes can be stored inside the same radar window and that the leading edge (which is later retracked to obtain the WSE) is not truncated. Examples and further details can be found in Dinardo et al. (2018 - Advances in Space Research 62 (2018) 1371–1404).

L. 179-189: "The range window is the vertical window during which the altimeter records the return echo from the emitted pulse. For satellites operating in closed-loop mode, there may be a transition phase before the range window is correctly positioned in regions with rapidly changing topography (Dinardo et al., 2018). If the topography is too steep, the standard fixed-size receiving window of 256 samples cannot store the elevation of all the samples in the echogram prior to Level-1b processing (e.g. Figure 5 in Dinardo et al. (2018)). By extending the receiving window, all the echoes can be stored in the same matrix without truncating the leading edge, which will be retracked to obtain the WSE. In open-loop mode, truncation might occur close to changes in the OLTC, where the receiving window may still be positioned according to the previous target. OLTC targets might be far apart due to space limitations of the OLTC (Le Gac et al., 2019), resulting in steep changes when a new target is introduced. Extending the receiving window can accommodates these sudden shifts in the position of the range window as well. We therefore process all tracks using a double and triple receiving window, to identify where the extension might be useful. "

In section 2.3.3, please cite briefly the corrections taken into account in the two datasets.

In the Scihub dataset, the files contain the already "corrected altimeter elevation from OCOG (ice-1) retracker" and only the geoid needs to be subtracted. In GPOD the instrumental

corrections are applied already and only the geophysical corrections need to be handled and they are already aggregated. The corrections include:

- Instrumental corrections: USO drift correction, internal path correction, distance antenna-COG and Doppler-slope correction
- Geophysical corrections: GIM-derived ionospheric correction, model dry tropospheric correction, model wet tropospheric correction, solid earth tide height, geocentric pole tide height and and ocean loading tide.

The corrections are provided individually, but this greatly simplifies the task for the less experienced user. We have added the following details in the manuscript.

L. 208-212: "In both datasets, instrumental corrections have already been applied to the 20Hz retracked range,  $R_{unc}$  (i.e. USO (Ultrastable Oscillator) drift correction, internal path correction, distance antenna-COG (Center of Gravity) and Doppler corrections).  $R_{unc}$  must also be corrected for geophysical and propagation effects (i.e. pole tides, solid Earth tides, ionosphere, and dry and wet troposphere), here summed into  $R_{geo}$  to obtain the corrected range,  $R_c$  (Eq. 1)"

L. 214-219: "In the GPOD dataset, the geophysical corrections are aggregated and provided as a single variable to be subtracted from the retracked range. In the SciHub dataset, the geophysical corrections have already been subtracted from the OCOG-retracked elevation. In both cases, all corrections are also available separately."

Line 170: According to Jiang et al. (2020), the RIP is in Watt, so please indicate the unit in "(>10-13)"

L. 204: Unit added to RIP criteria

Lines 174-175: Could you provide some estimates of the two DEM errors (provided in the DEM reference paper or in the DEM quality matrix, for the Zambezi basin). It would help the reader to assess if the 30m threshold is much above the DEM accuracy.

We have added the following to the manuscript:

L. 227-231: "The expected uncertainty of the MERIT DEM is less than 2 m for 58% of land pixels globally (Yamakazi et al., 2017). Based on the project accuracy matrix, ACE-2 has an accuracy better than 10 m for over half of the virtual stations in the basin and better than 16 m throughout the catchment (Berry et al., 2019, 2010). Thus, we do not expect a significant number of false negative outliers due to DEM accuracy based on the allowed window of uncertainty. One exception may be new dams and reservoirs, altering the surface elevation by more than 30 m; however, this does not appear to be an issue in this catchment."

Line 178: According to Jiang et al. (2020), the fit is a gaussian fit, isn't it? It could be worthwhile to mention it (and maybe to add a sentence to explain why the fit is needed).

In this paper, we calculate the Stack Peakiness using the maximum and mean RIP, therefore no fit is applied to the RIP beforehand. We will remove "fitted" from the text to avoid confusion.

Line 195: "Retrieving the untracked range gives an assessment of whether the expected WSE was within the on board reception window", I agree, but this statement is very general and you could better explain

how you will use this information. If you don't know the expected WSE (which is the case for 169 of your VS), I don't see how you can really make use of this information. Will you compare it to DEM?

We expand this sentence slightly to better introduce the use of the untracked range. It allows us to track how the range changes when new targets are uploaded and to identify whether the OLTC is at the source of problems with the data.

The first outlier filtering is indeed through comparison with the DEM but using the retracked WSE, not the untracked range. If the difference is too large, it can be assumed that the WSE will be outside of the listening window. This of course only allows very coarse filtering. We use the DEM as a reference, but most importantly it can help explain why some VS fail. If we are very far from the DEM it is unlikely that the target was sensed at all. Of course there is a risk in some cases that the DEM is so wrong that we lose information due to this filtering (e.g. due to dam construction). We do not expect this to be the case in the Zambezi basin.

With regards to the untracked range, the examples where we see large errors, the untracked range is off by far more than the expected DEM error.

L. 246-251: "The tracker range is the on board positioning of the expected leading edge according to the OLTC. Plotting the along-track tracker range reveals how the range window position changes based on the OLTC targets and updates to the OLTC. The on-board surface elevation must be correct and the surface elevation must be within the range window to obtain useful observations of the water surface. The tracker range also provides insight into the Level-1b processing options of the two datasets, particularly where the range window is repositioned. If this occurs close to a virtual station, there may be impacts on the tracker range depending on how the transition is handled, e.g. by extending the receiving window."

*Line 211: "WRMSD (Weighted RMSD) by dividing with the residuals with the in-situ standard deviation" it not clear, please rephrase Equation 3, to be coherent with the text change D\_{RMS} with RMSD*

L. 264: Indeed, it should be: "WRMSD (Weighted RMSD) by dividing the residuals with the in-situ standard deviation"

Equation 3 was modified in line with HESS recommendations to avoid abbreviations in equations – for clarity and because it is a widely used abbreviation, we chose to retain WRMSD/RMSD in the text.

Line 217: "We correct for datum shifts by using the WSE amplitude and therefore expect a bias of 0 cm." You need to provide more explanation. First, how did you use the amplitude and to compute what? Second, I don't understand why you need to correct datum shifts, as you already removed to time series "mean level at overlapping sensing dates is subtracted". So why is it needed to add any other bias correction?

The amplitudes are used for comparison with the ground observations – the section is indeed unclear, as it is the same bias correction mentioned in two different ways. This will be rephrased for clarity.

L. 261-263: "In order to account for any vertical bias between the two ground and satellite observations, the mean level at overlapping sensing dates is subtracted from the in-situ and satellite WSE respectively."

Lines 244-251: All the criteria used by the authors are not easy to follow, as they depend of the dataset and the product level. It should be better explained in the methodology section, with a clear flowchart of the process and a more in depth explanation of all the criteria used.

Thank you for the suggestion – we have added a flowchart to the paper to better illustrate the processing steps and to reorganize the methods section accordingly, which will hopefully clarify the following sections. For the particular section, we suggest to already mention in the methods that the NP is used as a selection criteria. The criteria are the same across datasets, however MP and SP are not calculated for the SciHub dataset as the RIP is not available.

Figure 2 added to the manuscript.

Line 250: "we use NP alone as the L1b selection criterion" but how did you use NP? Using which threshold? It is somewhat difficult to understand why NP is a good criterion, as multiple targets could be in the waveform and it does not mean that the altimeter is not observing the target of interest. It is especially true for small tributaries, where there are a lot of missing data on Figure 2. The missing data could also be due to your criteria. Could you discuss it in more details?

We have clarified the wording in section 2.3.6:

L. 241-245: "MP and NP are indicators of the presence and number of bright targets respectively, while SP and PP provide information on the shape of the waveform. A river-like surface is typically smooth and highly reflective, resulting in quasi-specular reflections. This will typically translate into narrow, peaky waveforms and consequently high SP and PP values. We use NP to classify the VS at Level-1b, assuming stations with over 90% single-peak waveforms are likely to be good water targets with useful time series."

Indeed multiple targets could be in the waveform, however that increases the risk of retracking errors if there are multiple high power targets. The missing data in Figure 2 is not due to this criteria as it is only based on what WSE observations could be extracted after corrections, water mask selection and filtering.

We also checked all rejected stations and found the following cases:

- Most have very little valid data at all, or several outliers (rejected on criteria of 80% data should be available). For some of these stations, dedicated (most likely manual) processing could help retrieve information if they were located in areas of interest
- Data loss due to OLTC update
- A few stations have seasonal water observations but with a very wide spread this is the case for VS on narrow river targets in wetlands.

 The stations rejected based on the single peak criteria mostly have very large acrosstrack standard deviations, suggesting it is not unlikely that the waveform is contaminated by other bright targets and justifying the rejection of the VS.

Some of the retained VS might also require some degree of manual validation or outlier removal, however the proposed filtering greatly reduces the task (> 200 VS to check versus just over 100). It also allows users to group VS that they wish to further inspect and validate and to provide tools for pre-selection and evaluation. We propose to summarize this information in the revised manuscript.

To clarify this, we have re-written section 3.1. including adding

L. 303-306: "Furthermore, as the SP and PP cannot be calculated based on the waveforms processed on SciHub, the VS are evaluated at Level-1b based on the NP. We select stations with predominantly single-peak waveforms (along-track median NP = 1 in over 90% of the observations associated to the VS). In total, 101 Sentinel-3A and 103 Sentinel-3B have complete records and promising waveform statistics."

L. 311-319: "The rejection rate is higher in the SciHub dataset, with rejected stations throughout the basin. This is mainly due to the lower percentage of missing data in the Level-2 data. OCOG is an empirical retracker, less likely to fail on non-water waveforms. Samosa+ is a physical retracker developed for coastal regions but suited to inland water targets. If the model misfit is too high, the retracker fails and the VS is rejected based on this missing data.

A closer look at the stations with a large fraction of missing observations or multi-peak waveforms in both datasets revealed that at some stations, outliers caused the rejection but could be removed with dedicated, manual post-processing if the stations were located in areas of interest. In several cases, the rejected stations were located on narrow rivers crossing seasonal floodplains, with along-track standard deviations exceeding the seasonal variation. This was mostly the case when the station was rejected based on the single-peak criteria, justifying the rejection of the station. The proposed approach allows users to group the VS for further inspection, e.g. starting out with the VS most likely to hold useful river WSE observations."

*Line 254: "The rejection rate is higher in the SciHub dataset, with rejected stations throughout the basin." This sentence seems to meet the concern expressed in my previous comment. "*

Several of the stations rejected had missing data in the GPOD dataset because the retracker failed to fit a model waveform to the observed waveform, suggesting the target is not a good water target. The OCOG retracker is less sensitive to the shape of the waveform. The higher rejection rate balances this. The response to the previous comment also address this concern.

Table 2: What is the line "OLTC" in Table 2? It is not explained in the table legend, nor in the text.

We modified the line in the table (now Table 3): it is the stations with data only after the OLTC update in March 2019.

Table 3 caption: "We consider S3A VS with data only after the OLTC update in March 2019 (line "OLTC v. 5") as well as the two processing settings on GPOD (line "3x window extension") separately."

Figure 4: This figure does not seem useful, except to state that after OLTC update there is mainly 1 peak in the waveform. But as the NP before the update is not provided, it is difficult to estimate the improvement.

We have removed the figure and instead written in the text that the OLTC update also improves the NP statistics (as shown already in Table 2).

L. 292-296: "At 30 Sentinel-3A stations, no observations were available in the either dataset before the March 2019 OLTC update, suggesting the water surface elevation was outside the range window prior to the update causing the poor results prior to the update. Indeed, at over 90% of these stations, the Level-1b statistics are consistent with water targets."

Lines 266-267: "The OLTC contains targets based on elevation information from hydrology databases (e.g. Hydroweb), virtual stations networks and the global ACE2 DEM (Altimeter Corrected Elevations v.2 Digital Elevation Model)" Actually it depends of the OLTC version you are considering, as stated later in your paragraph. According to https://www.altimetry-hydro.eu/ here are the different OLTC table versions over inland waters: - For S3A: \* DEM: v5 (Date start: 2019-03-09) \* DEM: v4 2 (Date start: 2016-05-24, Date end: 2019-03-01) \* DEM: v4\_1 (Date start: 2016-04-18, Date end: 2016-05-24) - For S3B: \* DEM: v2\_0 (Date start: 2018-11-27) \* DEM: v1(tandem) (Date start: 2018-06-06, Date end: 2018-10-16) Especially, on the https://www.altimetry-hydro.eu/ you can see that ACE2 DEM is heavily used in v4\_2 for S3A, but not used at all in v5 over the Zambezi basin, as shown on Table 3 but not clearly stated in the text. Besides, at line 268 and in other part of the manuscript, it is written that the table has been updated in March 2019. It is true for S3A, but not for S3B, which has been updated sooner (after the end of the tandem phase in November 2018). The OLTC versions are given in Table 3, but never really explained in the text. A good reference for OLTC tables' generation is (with some validation): Le Gac S., F. Boy, D. Blumstein, L. Lasson and N. Picot (in press). Benefits of the Open-Loop Tracking Command (OLTC): Extending conventional nadir altimetry to inland waters monitoring. Advances in Space Research, https://doi.org/10.1016/j.asr.2019.10.031 I think putting a table to summarized all these OLTC versions and dates could be useful in the manuscript, with some information on OLTC generation (see Le Gac et al., in press). These information should be put somewhere in section 2.

We thank the reviewer for the citation suggestions – we incorporated them in the methods section as suggested. The update indeed only refers to the Sentinel-3A OLTC as the Sentinel-3B update as made prior to the beginning of the datasets considered. We will make sure this is clear in the manuscript.

Indeed the targets mainly consist of HDB targets after the update – however they still rely on high resolution DEMs – as we understand, ACE-2 is still used to define many hydrology targets. We are very interested in further information if other high resolution DEMs are used instead of ACE-2 for the HDB targets.

Section 2.2.4 now introduces the OLTC separately.

Figure 5: On the map, the black line (sub-basin boundaries?) are not defined, does not seem to be useful and make the map difficult to read. I suggest removing them. Where they are close, S3A and S3B VS are difficult to differentiate. Maybe use different color or level of grey between the two missions. On the subplots, write when it is S3A or S3B. In the legend, write to refer to figure 2 for the location of the map within the Zambezi basin (blue polygon on figure 2).

Agreed – we will remove the lines, increase the difference between the two mission markers and refer back to Figure 2 (now 3).

Figure 5 has been modified in line with both reviewers' suggestions.

Line 280: "no new targets were uploaded to the OLTC in March 2019 near the two S3A VS" Just to be sure, even if no new targets has been added in march 2019 near these VS, it does not mean that the OLTC table has not been updated in March 2019 for these VS. Is it the case? From figure 5 even if it is the case, the updated value should be pretty similar, as the time series seems pretty stable before and after March 2019.

Based on the online OLTC webpage, the existing targets were only updated with no significant change in height, suggesting there is no point in splitting the time series in before and after the OLTC update.

L. 330-331: "The OLTC did not significantly change at the VS considered, meaning WSE observations are available for the entire Sentinel-3 sensing period."

Line 288: "Samosa+ retracker outperforms the OGOC retracker", first replace OGOC with OCOG. Second, from this sentence, I was expecting much better results with Samosa+ than with OCOG, whereas on table 4, SAMOSA+ is better only by few cm (or %, even most of the times few tenth of %). So I would encourage the authors to add this quantitative information to alleviate this sentence. Besides, Samosa+ comes from GPOD, whereas OCOG comes from SciHub, and processing between these two platforms are different (not just the used retracker, but also the data selection and probably other processing, corrections...) as described on sections 2.3.1 and 2.3.2. How these differences could impact the results shown on Table 4?

Thank you for pointing out the spelling mistake.

The main difference is linked to a few outliers in the SciHub dataset, which might skew the along-track mean slightly. Indeed, it is more accurate to state that the GPOD processing package is slightly better than the SciHub standard product.

Section 3.3.1: Modified text to refer to GPOD/SciHub instead of Samosa+/OCOG L. 339-341: "The GPOD dataset performs better than the SciHub dataset at all stations, improving the RMSD with between 1.1 cm (7.5%, at Kalabo) and 10.2 cm (39.2% at Chavuma); except Matongo Platform, where the SciHub dataset improves the RMSD by 1.4 cm (4.5%)."

Table 4: For Chavuma station, Samosa+ RMSD is equal to 15.8cm and 3.3%, whereas for OCOG the RMSD is 25.6cm and 3.6%. How an almost 10 cm difference in RMSD between Samosa+ and OCOG only translates into 0.3% increase? I think there is an issue with the % computation (or with the RMSD value).

*Besides, the 9th column entitled "Relative RMSD" corresponds to WRMSD in the text, please replace "Relative RMSD" with "WRMSD" for consistency.*

Indeed, there was an error in the table as the values for Chavuma and Ngonye Falls were exchanged – thank you for pointing this inconsistency out.

Table 5: The correct RMSD relative to the yearly amplitude is 5.4% at Chavuma and 3.6% at Ngonye Falls.

*Line 315: "If we consider the stations, which are valid across datasets", how do you define "valid" here? Could you recall the criteria here?*

We have clarified how we refer to the selected stations in order to nuance the term "valid" – we consider the stations with single peak waveforms and a low degree of missing data as more reliable than those with multi-peak waveforms and a high degree of missing data. These are now referred to as "selected".

L. 362-363: "Fig. 7 shows boxplots of all selected VS based on the evaluation of the Level-1b and Level-2 data (< 20% missing data and along-track NP = 1 for 90% of the tracks)."

Line 316: "The number of VS is quadrupled compared to using the global database Hydroweb", it is impressive. However, it should be noted that all Zambezi VS on Hydroweb have an "expert validation criteria" (see http://hydroweb.theia-land.fr/?lang=en&basin=ZAMBEZI and https://theia.sedoo.fr/wpcontent-theia/uploads/sites/2/2020/04/Handbook\_Hydroweb-V2.0-1.pdf). Are all the 145 VS being individually checked and validated (coherent seasonal cycle and amplitude from upstream to downstream VS)? Coherent amplitude and seasonal cycle has been shown only for 10 VS (and compared to in situ gage data only for 6 gages) in the manuscript.

Yes indeed. Of course, this is why this number can be increased this dramatically at catchment scale. We are aware that the goal of global database can not be to provide all VS for all catchments, therefore we see a value in providing tools and lessons-learned in processing the data at catchment-scale as highly valuable data may be available. The data access is also faster when new satellites are launched (e.g. S3-B) allowing faster uptake of the data.

L. 375-381: "If we consider the stations with less than 20% missing data and over 90% singlepeak waveforms, there are 204 Sentinel-3 VS in the Zambezi, which contain potentially valuable information about WSE. Thus, automatically processing all Sentinel-3 observations within an area of interest can provide a highly valuable addition to global altimetric WSE databases, by increasing the spatial density of VS at catchment scale. The assessment based on the degree of missing data and on single-peak waveforms constitutes a preliminary validation of the virtual stations, although dedicated outlier filtering and validation might be necessary at some stations to ensure consistency with the catchment dynamics."

Line 320: "At four stations in the Upper Zambezi, there are no valid observations at any of the VS prior to the OLTC update (Fig. 8)" I don't see how figure 8 shows that there is no valid observation before OLTC update, as figure 8 is only showing data after (S3A) OLTC update.

The figure shows no data as no data could be processed before the update: i.e. due to no-data values, or too far from the DEM or very low backscatter.

L. 384: "At four stations in the Upper Zambezi, there are no observations prior to the OLTC update (Fig. 8)."

Figure 9: Concerning the zoom on the WSE vs. latitude plot (between -12.01°N and -11.81°N), it might be because of the color code, but it seems "after Schihub" WSE is in between 1050m and 1100m, whereas "After (GPOD/3x" WSE is in between 1000m and 1050m and "After (GPOD/2x" WSE is below 950m. I don't understand why there are not more consistent. I understand it is near the transition, which affect GPOD when changing the receiving window, but why is it that different, especially why "GPOD/3x" is above "GPOD/2x" and not below (by tripling the window, you should have more data after the transition)? Besides, on the WSE vs. latitudes plots, I would suggest to draw all "Before" curves with dashed lines, to make them easier to differentiate with the "After" curves.

We suspect it is before too much weight is given to the next target, over-smoothing the transition. The algorithm as we understand is developed closed-loop where you want a faster transition, which is what happens. In open-loop the fast transition already occurs and the algorithm introduces an artificial transition. We have been in contact with GPOD who confirmed that the results where as expected. The performance at this VS explains why processing options are so significant at certain locations.

Thank you for the suggestion for the figure.

Figure 9 has been updated accordingly: the "Before" curves with dashed lines and the y-axis label has been corrected.

*Line 350: "According to the OLTC website," please give the URL of this website (I guess it is* https://www.altimetry-hydro.eu/)

This line has been removed to ensure that the section about the OLTC is concise and reflects the findings of this study in particular.

Section 4.2, which VS is considered here? A86 VS on figure B1? Besides, it to better see the impact of the platform processing versus the OLTC value, it could be good to show the OLTC table value (i. e. position of the tracking window), rather than the retracked value converted to WSE. It would help to show the transition and why you need to extend the receiving window with GPOD and then you can discuss the difference between the two platforms.

We have added detail about the VS considered. We noted in this context that a number of VS where erroneously left out from, so a slight renumbering is necessary. The shown height is the tracking window position – we will correct the y-axis label accordingly.

L. 407: "at is A102 on the Kafue"

Lines 370-372: The second option is the one chosen during the March 2019 update, isn't it?

Actually, both options are true in this case – the target was defined earlier, but for the GPOD dataset, an extension of the receiving window is still necessary.

L. 425-428: "In the example above, the latter is necessary when using the GPOD dataset, and although not critical to data retrieval, the position of the target was also shifted in the OLTC update of March 2019. Based on these findings, we recommend using the triple window extension when processing catchment scale datasets on GPOD to maximize the number of VS."

*Line 389: "This is likely due to the frequent cloud cover over the floodplain." Or maybe due to vegetation cover masking water?*

This is a very good point – yes.

L. 446-448: "This is likely due to the frequent cloud cover over the floodplain or vegetation masking the water surface in optical images, stressing the importance of integrating SAR imagery into water mask processing."

*Line 412: the references provided here are just examples of studies using altimetry to calibrate and update hydrology model, so I suggest putting "e.g." before the references.*

L. 482: e.g. has been added.

*Lines 423-428: There is not just Park (2020) and your study which investigated connectivity between river and floodplains. Could you increase your references list?*

The citation is a single example of course, we have increased the reference list, including by adding a paragraph in the introduction better showing this.

L. 38-46: "Wetlands and floodplains provide important economic and ecological services and are intrinsically linked to river dynamics. Several studies have used altimetry WSE to characterize river-floodplain interactions (e.g. Park et al., 2020, Zakharova et al., 2014, Ovando et al., 2018, DaSilva et al., 2012). Park et al. (2020) recently showed the potential in using satellite altimetry for this purpose using Jason-2 WSE in the Amazon and Zakharova et al. (2014) assessed the seasonal variability of boreal wetlands in Western Siberia using Envisat altimetry. Due to the temporal resolution of Envisat (35 days), an interannual characterization of the wetland processes was not possible. By definition, the satellite orbit is a compromise between spatial and temporal sampling. Dettmering et al. (2016) used Envisat altimetry to characterize water levels in the Pantanal Wetlands but their methods were constrained by the accuracy of the method compared to the level variations in large regions of the Pantanal. They cited SAR technology as a potential solution to overcome these limitations."

L. 496-499: "Furthermore, the accuracy achieved at in-situ station Kalabo in the Barotse floodplain (2.9 cm with the GPOD dataset) is promising in terms of characterizing level variations in the decimeter range. This has important implications for successful monitoring of wetlands and floodplains with smaller level fluctuations (Dettmering et al., 2016)."

Line 429: "The cross-sections extracted over floodplains are similar to observations expected from the future SWOT" I disagree with this statement. Even if Sentinel-3 mission provides much more spatial observations than other altimetry missions (like Jason series), it is not comparable to SWOT measurements, which will provides images of WSE. So rephrase this sentence accordingly.

Line 430: Concerning SWOT mission, I think a better reference for the reader will the SWOT Science Requirements Document (SRD) rather than Domeneghetti et al. (2018). SWOT SRD could be accessed with the following link:

https://swot.jpl.nasa.gov/system/documents/files/2176\_2176\_D61923\_SRD\_Rev\_B\_20181113.pdf

Line 432: "Similar information can already be extracted from the Sentinel-3 dataset in selected locations" Similarly to my previous comment, I think this sentence should be rephrased. S3 is providing WSE, but not water mask and of course slope could be computed between close VS, but it is far from being the one expected from SWOT images. . .

In response to the three comments above:

Based on reviewers comments, we have removed the reference to SWOT and instead focus on the spatio-temporal sampling and performance of Sentinel-3 for hydrological applications. We have entirely re-written section 4.4 to focus on hydrological applications.

Lines 445-446: "We extract over 360 virtual stations from each satellite of which over 70 are validated based on the waveforms and temporal coverage for each Sentinel-3 satellite" Why stating this in the conclusion and not in the core of the manuscript? In the abstract 170 VS are mentioned. The same goes for the 70 validated VS.

We state it here as a concluding remark and have reviewed the numbers mentioned. We instead cite the 204 promising Sentinel-3 VS in the conclusion out of 731 total VS.

L. 509-511 "In total, the spatial coverage of the dual-satellite mission consists of 731 potential virtual stations in the Zambezi, of which 204 show promising results based on the evaluation of Level-1b waveforms and Level-2 WSE observations across datasets."

Section 4.4 and 5: I find it strange to have perspectives before conclusions...

We have rewritten the section and the conclusion in order to ensure perspectives are placed after the conclusion.

**Replies to the comments by anonymous referee #2:**

We would like to thank the reviewer for their interest and comments on the manuscript. Below the reviewer's comments are in italic and the replies in normal font.

The paper describes the computation and exploitation of satellite altimetry water level time series in the Zambezi basin. According to the authors, the aim of the study is "to assess the potential of the Sentinel-3 mission in hydrological applications". For that purpose, they compare different satellite altimetry pre-processing options (from two different databases) and they analyze the impact of open loop processing. Moreover, a validation by comparison with (few) in-situ ground stations is performed. For three different wetlands within the study basin, the potential of Sentinel-3 for monitoring the interaction of river and floodplain is shown.

General comments: This is an interesting topic worth publishing. However, some aspects of the paper are not innovative and had been published before by some of the same authors (e.g. OLTC impact by Jiang et al., 2020). Moreover, some parts of the manuscript are quite technically without providing the (less experienced) readers a recommendation on which processing option to use. In my opinion, the most interesting and innovative part of the study is the approach of automatically processing all possible VS of both satellites in the entire basin with the aim to use these time series for assessment of wetland-river interactions. Thus, my recommendation would be to focus on this part of the study by adding a bit more statistics (how many potential VS, how many valid VS, how many VS gained by OLTC,...) and some citations of existing work on wetlands based on satellite altimetry (e.g. Zakharova et al., 2014; Dettmering et al., 2016; Park, 2020). In addition, a (at least theoretical) comparison to classical missions can be added discussing the benefit of the dual satellite constellation (with respect to spatial and temporal coverage) and the measurement mode (SAR/OLTC).

We thank the reviewer for the interest in the manuscript and the comments. We agree on the analysis of the major contribution and propose to better highlight this in the introduction. We also propose to add a flowchart to the methods section, allowing a better overview of the different processing steps for reproduction. Finally, the Zambezi network and generic processing tools will be published with the final manuscript.

In terms of the statistics requested, we have updated Table 3 and Table 4 and rewritten section 3.1.

The suggestions for additional citations regarding wetland studies have been added to the text in the introduction:

L. 38-46: "Wetlands and floodplains provide important economic and ecological services and are intrinsically linked to river dynamics. Several studies have used altimetry WSE to characterize river-floodplain interactions (e.g. Park et al., 2020, Zakharova et al., 2014, Ovando et al., 2018, DaSilva et al., 2012). Park et al. (2020) recently showed the potential in using satellite altimetry for this purpose using Jason-2 WSE in the Amazon and Zakharova et al. (2014) assessed the seasonal variability of boreal wetlands in Western Siberia using Envisat altimetry. Due to the

temporal resolution of Envisat (35 days), an interannual characterization of the wetland processes was not possible. By definition, the satellite orbit is a compromise between spatial and temporal sampling. Dettmering et al. (2016) used Envisat altimetry to characterize water levels in the Pantanal Wetlands but their methods were constrained by the accuracy of the method compared to the level variations in large regions of the Pantanal. They cited SAR technology as a potential solution to overcome these limitations."

We thank the reviewer also for the suggestions for the discussion. Section 4.4 has been rewritten with a focus on the density compared to Envisat and a mention of the potential accuracy of Sentinel-3 in terms of studying wetlands.

L. 473-480: "The high number of VS throughout the basin can form the basis of a dense monitoring network. Michailovsky et al. (2012) assessed the number of VS in the Zambezi from Envisat and found 423 crossing points against 731 with Sentinel-3, and after careful evaluation, 31 VS had useful records. Although all 204 VS were not manually checked, the results in this study confirm that this number is greatly increased with Sentinel-3. The spatio-temporal sampling of altimetry missions often constrains monitoring capabilities. Particularly the dualsatellite configuration of Sentinel-3 thus offers new, interesting possibilities in a hydrological context. It is important to note that the success is entirely dependent on the accuracy of the OLTC tables as data is missing from the Sentinel-3A records in large part due to the latency between mission start and OLTC update."

L. 496-499: "Furthermore, the accuracy achieved at in-situ station Kalabo in the Barotse floodplain (2.9 cm with the GPOD dataset) is promising in terms of characterizing level variations in the decimeter range. This has important implications for successful monitoring of wetlands and floodplains with smaller level fluctuations (Dettmering et al., 2016)."

**Specific comments:**

*Line 5: If the objective of the study is to "evaluate the density of valuable observations", you should add some more statistics on the number of VS (see general comment).*

The objective is to show the value of processing Sentinel-3 at catchment scale, illustrating the performance for the Zambezi basin. We have added the requested statistics from the general comment.

We have updated Tables 3 and 4 for this purpose (previously 2 and 3).

Line 18: In my opinion, the paper is not showing the benefits of SAR (with respect to what? LRM?). The denser track network is due to the orbit configuration not the measurement mode, and there is not comparison to LRM data. The RMSD values are similar to those from LRM missions. So, how is the benefit demonstrated?

The results support the progress in results observed in past papers as well, where the RMSD is lower with SAR missions than LRM (e.g. in comparison to Michailovsky's results with Envisat). Direct comparison is difficult due to the lack of overlap in space and time. But we agree that this

is not a key investigation in this paper and instead, we now highlight the benefit of the spatiotemporal sampling achieved by the dual-satellite constellation and orbit.

L. 18-19: "These results highlight the benefit of the high spatio-temporal resolution of the dualsatellite constellation, which holds important implications for future hydrology-oriented missions."

**Line 44: Sentinel-3 is not only an ESA mission => Copernicus**

Thank you for pointing this out – indeed, Sentinel-3 is part of the Copernicus program and the mission is developed by ESA in this context.

L.47: "The Sentinel-3 mission was developed by the European Space Agency (ESA) mission for the Copernicus program."

Section 2.2: What about adding additional information on in-situ validation observations and OLTC targets?

Section 2.2.4 and 2.2.5 have been added, incorporating the information about the in-situ stations and OLTC table from other sections.

*Line 104/105: Please add some more information on the stream burning. I'm not sure what is meant here.*

This is part of the data processing for the river network database, as detailed in the cited paper.

We selected the dataset from Yan et al. (2019) because they defined the river networks globally (thus the same dataset can be used for other study cases) and because in addition to a river delineation algorithm, they burnt in a river line to the DEM, increasing the accuracy of the river location, particularly in flat areas.

L. 122-124: "Yan et al. (2019) included a stream burning step prior to the application of the river delineation algorithm to improve the river localization compared to the DEM processing alone particularly over plain areas."

Section 2.3.3: Some detailed info on the corrections is missing (e.g. which models).

We have added details about the different corrections as well as how they are provided in each dataset. In both cases we use the recommended corrections from each processing platform.

L. "In both datasets, instrumental corrections have already been applied to the 20Hz retracked range, Runc (i.e. USO (Ultrastable Oscillator) drift correction, internal path correction, distance antenna-COG (Center of Gravity) and Doppler corrections). Runc must also be corrected for geophysical and propagation effects (i.e. pole tides, solid Earth tides, ionosphere, and dry and wet troposphere), here summed into Rgeo to obtain the corrected range, Rc (Eq. 3)"

L. "In the GPOD dataset, the geophysical corrections are aggregated and provided as a single variable to be subtracted from the retracked range. In the SciHub dataset, the geophysical

corrections have already been subtracted from the OCOG-retracked elevation. In both cases, all corrections are also available separately."

Line 171: Sigma0==backscatter?

Yes, it is the backscatter coefficient, now indicated L. 204.

Section 2.3: I recommend to provide also the web addresses of GPOD and SciHub (in the text or alternatively in Refs or Acknowledgements.

Good point.

L. 163 and L. 164 web addresses have been added.

*Line 174/175: Are these DEMs good enough to be used in this context. My personal experience is that at least ACE2 includes really large outliers in some regions.*

ACE-2 has an accuracy of 5-10 m at most VS in the basin (and almost always less than 16 m). The choice of DEM might bias the selection, however the +/- 30 m window should not be a problem.

L. 227-231: "The expected uncertainty of the MERIT DEM is less than 2 m for 58% of land pixels globally (Yamakazi et al., 2017). Based on the project accuracy matrix, ACE-2 has an accuracy better than 10 m for over half of the virtual stations in the basin and better than 16 m throughout the catchment (Berry et al., 2019, 2010). Thus, we do not expect a significant number of false negative outliers due to DEM accuracy based on the allowed window of uncertainty. One exception may be new dams and reservoirs, altering the surface elevation by more than 30 m; however, this does not appear to be an issue in this catchment."

Line 200: "are processed" => how? Median/mean

L. 260: "We calculate the along-track mean of all observations retained at a given virtual station to produce a WSE time series. "

*Line 204: "six". Where are these stations located. Maybe you can reference to a figure.*

Indeed the locations are not presented until Figure 5 – we have added the stations to the catchment basemap (Figure 1) to show the geographical coverage of the gauging stations.

Added to Figure 1.

**Line 210: RMSD or D\_{RMS}? Please make consistent**

Equation 3 was modified in line with HESS recommendations to avoid abbreviations in equations – for clarity and because it is a widely used abbreviation, we chose to retain WRMSD/RMSD in the text.

L. 263-265: "Performance is evaluated by calculating the RMSD (Root Mean Square Deviation),  $D_{RMS}$ , between the relative in-situ ( $w_g$ ) and satellite ( $w_s$ ) levels (Eq. 3), and the WRMSD (Weighted RMSD)."

Line 231: "two the" => "the" or "the two"

L. 287: "the two"

Figure 2: I can't find any black cycle in the plot. On the other hand blue lines (which I assume to be rivers) not covered by data. The black lines are a bit confusing here. I guess these are sub-basin borders. Please indicate or remove. The additional maps seems to be in the Annex, not in the supplementary material.

The reference to the additional maps should indeed be the Annex. We agree that the subbasin borders do not carry significant information in this case and have removed them.

Thank you for pointing out that the circles were missing. There are parts of the river, which fall between tracks and are thus not sensed by either satellite.

Figure 3 has been updated.

*Title of 3.1: This is quite technically. What about using a title indicating the aim of this section, e.g. comparison of different L1b pre-processing*

As we have rewritten section 3 and 3.1, the title now refers to the evaluation of the VS in the catchment.

3.1 Evaluation of Sentinel-3 VS in the Zambezi catchment.

*Figure 3: What does OLTC stands for here (black and orange)? Before/after OLTC update? Please clarify.*

Figure 3 caption: ""OLTC" indicates Sentinel-3A stations where observations are only available after the OLTC update in March 2019."

Table 3: Please provide the sum over the entire basin. Include description of GPOD/SciHub version for VS no (I guess it should be 2x, 3x?)

Table 4 has been updated with the basin totals and with new lines to better identify the source of the number of VS.

Section 3.3.1. What about adding a discussion on the impact of low number and distribution of the validation sites. Are the validation numbers representative for the entire basin?

Of course the validation is limited by the low number of validation sites. However, section 3.3.2 confirms that the hydrological patterns are reliable in other parts of the basin as well. This pattern of data availability is also why S3 holds high value in a catchment with low gauging density.

L. 348-350: "The in-situ stations are mainly located in the Upper Zambezi, therefore the validation is geographically constrained. However, the river morphology at the ground stations is diverse, ranging from 95 m wide rivers to 35-600 m on the Barotse floodplain. Therefore the validation is presumed to be an encouraging indication of the performance basin-wide."

Figure 5: in-situ (black) lines are not visible. Are they always available for the whole period? Are there more than one observation available per epoch (=> single alongtrack measurements instead of mean/median?) Can you add RMSD here?

We have stippled the S3 lines to make the underlying black lines visible. The in-situ observations are available until April 2019 at all six stations.

In some cases, there are more than one observation – indicated by the points – whereas the line indicates the mean WSE, which is compared to the in-situ observations. The RMSD is given in Table 4.

Figure 5 has been updated.

**Line 288: OGOC => OCOG**

Thank you for pointing this out.

Table 4: is the difference only due to the retracker? Might the pre-processing play a role? Is the Relative RMSD == WRMSD?

Indeed the pre-processing might also play a role, although both are intrinsically linked to the processing platform and to each other. For more clarity, we refer to the datasets by the platform rather than the retracker in 3.3.1.

We have corrected to WRMSD in Table 4.

**Figure 6: I can not find any orange lines here. . .**

There are indeed no observations from those decades. We have removed orange lines from the legend of Figure 6.

**3.3.3: What about adding some more information and interpretation here.**

We have added a discussion linking back to the in-situ stations discussed in the two previous sections, which indicate annual amplitudes in the order of 5-10 m. Furthermore, Figure 7 provides a summary of the Sentinel-3 observations, suggesting that in some cases further manual validation might be necessary, i.e. to remove large outliers or confirm that the patterns are hydrologically consistent.

L. 365-367: "We note that for Sentinel-3B, the amplitudes are smaller than for Sentinel-3A. This is due to the length of records, with indications of 2019 being a dryer year than 2016-2018, as seen in Fig. 5 at Senanga and Kalabo, and when comparing the Sentinel-3B records to in-situ records in Fig. 6."

L. 371-381: "If we consider the stations with less than 20% missing data and over 90% singlepeak waveforms, there are 204 Sentinel-3 VS in the Zambezi, which contain potentially valuable information about WSE. Thus, automatically processing all Sentinel-3 observations within an area of interest can provide a highly valuable addition to global altimetric WSE databases, by increasing the spatial density of VS at catchment scale. The assessment based on the degree of missing data and on single-peak waveforms constitutes a preliminary validation of the virtual stations, although dedicated outlier filtering and validation might be necessary at some stations to ensure consistency with the catchment dynamics." Line 323-324 (and in some other parts of the manuscript): I'm not sure whether it is fair to compare with global WSE databases. Since these databases aim in providing input for hydrological research, the focus is on long time-series. For sure, they are also able to process these VS - however, this has no priority given the short time series of less than 2 years.

The comparison should be seen as an encouragement to explore the public processing platforms, which provide access to the full Sentinel-3 dataset, beyond what is available on the databases. The databases provide an excellent starting point, however, at catchment scale (including for smaller rivers) or where short time series would have useful applications there may be more information available. This paper illustrates how much additional data can be obtained through automatic extraction from the full dataset.

L. 377-378: "Thus, automatically processing all Sentinel-3 observations within an area of interest can provide a highly valuable addition to global altimetric WSE databases, by increasing the spatial density of VS at catchment scale."

*Line 342: Is there any statistics available on the percentage of improvement/degradation by OLTC in this region?*

We are not sure we understand the question – to obtain statistics a simultaneous closed-loop mission would be necessary. What we do see is cases where the time series stops after the update and a loss of data due to the time lag between mission launch and table update. This is quite significant as large amounts of data are potentially useless when the OLTC is not up to date.

**Line 349ff: "mamsl": all other heights are provided with respect to a geoid. Why not these ones? At least you should explain the abbreviation.**

The elevation is from the OLTC database (altimetry-hydro.eu) and thus actually relative to a geoid. We have corrected the unit to avoid confusion.

L. 411-415: "The tracker range from the SciHub dataset suggests that the range window was correctly positioned within +/- 10 m of the surface elevation at around 1111 m (Le Gac et al., 2019). The discrepancy can instead be attributed to the waveform processing, as illustrated in Fig. 11. After the OLTC update a target is defined for the VS at 1113 m and the transition occurs earlier on the pass. The altimeter reception window has shifted just enough that the VS elevation is within the receiving window for all three 415 datasets, including the GPOD dataset with the double extended receiving window.

**Line 369: options to mitigate: Do you have any recommendation for the users? What preprocessing should I use?**

This is a tricky question with no clear single answer. In some cases, the dedicated inland water options outperform the standard processing (as would be expected), in others they appear to actually worsen the results. The take-home message is that the choice of preprocessing does indeed matter and based on the virtual station and its location it might be worthwhile to

consider several. We do recommend using the 3x extension for GPOD processing to maximize the number of VS.

L. 427-428: "Based on these findings, we recommend using the triple window extension when processing catchment scale datasets on GPOD to maximize the number of VS."

**Figure 12: Is there any color change in c) and d) depending on waveform misfit?**

Indeed – in this particular case, the misfit is generally quite low with no significant change, making the misfit information superfluous.

Figure 12 has been modified.

**Line 385/386: Are there no unique track numbers?**

The given track numbers are the relative track numbers which are the same across cycles – and all data will belong to those same tracks.

L. 440-441: "Rather than grouping by coordinates, we here assess all unique passes, known to cross floodplains."

**Figure 13: What are the vertical blue lines in crossing tracks 741 and 498? Where are the VS located for tracks 498 and 085? What are the stars and cycles in the left hand plot?**

The vertical blue lines are the water occurrence (we will add this to the figure caption) as seen in the basemap on the left. The VS are the cycles and stars in the left hand plot and are indeed missing from the legend (in Figure 14 and 15 as well).

**Figure 15: left and right?**

Left and right are erroneous in this case and have been removed.

4.4 This is more a summery than perspective. . . Moreover, perspective should be placed after conclusions. . . Line 409: "first" => where is second?

We have rewritten section 4.4 and changed the title to "Hydrological applications" as it is part of section 4 Discussion.

*Line 429-434: Please reformulate this paragraph: SWOT will provide much more information than S3, especially in cross-track direction. Also CS2 can already extract similar information in selected locations.*

The section has been reformulated and there is no longer a reference to SWOT in line with comments received on this part.

**Line 441: "should"? => is or is not improving!**

The point addressed here is the expectation to OLTC vs. closed-loop rather than the conclusion of this study. The conclusion has been reformulated, removing this particular sentence.

Line 446: I don't think that you should name that a "validation"

L. 509-510: "In total, the spatial coverage of the dual-satellite mission consists of 731 potential virtual stations in the Zambezi, of which 204 show promising results based on the evaluation of Level-1b waveforms and Level-2 WSE observations."

*Line 447: Again: My feeling is, that this is not a fair comparison. Hydroweb is a global database not aiming in complete coverage of entire basins.*

We completely agree that a comparison would not be fair – our intent with the comparison of the number of VS is to highlight the benefit of extracting data beyond what is already available on such databases, when looking at altimetry data at catchment scale. This is important for regional to local hydrological studies. Furthermore, we ease the WSE data retrieval for hydrologists. By showing the numbers together, we hope to encourage interested users to also consider the full dataset on publically available processing platforms.

L. 516-517: "The proposed approach illustrates the potential of considering the full Sentinel-3 records to achieve complete basin coverage, a substantial supplement to the WSE time series available on global altimetry databases."

*Line 452-458: I suggest shifting this paragraph to line 443 (as second paragraph of this section). This would make the paper end with the application, which is the overall focus of your paper according to line 69.*

Indeed, thank you for this suggestion. We have rewritten the conclusion, effectively removing this paragraph to better highlight the focus of the paper.

[revised manuscript text omitted]

105 wetlands at catchment scale.

Several databases provide global, ready-to-use and publicly available time series of WSE for inland water bodies derived from satellite altimetry observations, including from Sentinel-3 e.g. Hydroweb (http://hydroweb.theia-land.fr/). Furthermore, the , DAHITI (https://dahiti.dgfi.tum.de/en/) and HydroSat (http://hydrosat.gis.uni-stuttgart.de/php/index.php). However, they do not provide full catchment-scale coverage and there is a time-lag between data acquisition and the inclusion of the VS

- 110 in the database. The Sentinel-3 dataset is available on public processing platforms with dedicated tools for WSE extraction over inland water. In order to benefit from the high spatio-temporal coverage of Sentinel-3 and large number of hydrological targets, automatic processing workflows and evaluation tools are necessary. For instance, the mission has operated in dualsatellite constellation since November 2018, providing at least one over a year of non-time critical data from Sentinel-3B not yet available on the aforementioned databases.
- 115 The aim of this study is to assess demonstrate the potential of the Sentinel-3 mission in hydrological applications (e.g. monitoring, modelling and river-floodplain interactions) by extracting a catchment-scale WSE monitoring network of Sentinel-3 VS. Where ground observations are available, we VS using the full Sentinel-3 records. We evaluate the satellite performance directly against in-situ data - We where these are available and investigate the impact of processing choices - by evaluating the implications of the open-loop tracking mode and on board OLTC for hydrological applications on the WSE time series at
- 120 selected VS. Finally, we explore the potential of the dual-satellite constellation for spatio-temporal monitoring of wetlands and floodplainsusing Sentinel-3. The purpose of these investigations is to confirm that the network can serve as a useful supplement to the in-situ gauging stations by capturing temporal dynamics across the catchment.

. To address these objectives, we extract all available Sentinel-3A and Sentinel-3B observations over the Zambezi basin using-use two publicly accessible databases - We and present an automatically extracted catchment-scale river WSE monitoring network based on Sentinel-3 radar altimetry for the Zambezi. All processing steps are performed on publicly accessible

databases or using open-access code.

125

**2 Data and study area**

**2.1 The Zambezibasin**

The Zambezi basin is the largest river in Southern Africa and drains 1,390,000 km2 km2 stretching over eight countries (Fig. 1). Water resources in the basin are crucial for human consumption, hydropower production, irrigation and ecosystem services (Beilfuss, 2012). There are three distinct seasons: the wet and warm season from November to April, the cool and dry season from May to July and the hot and dry season between August and October. The river and its tributaries display a strong seasonal signal, which should be reflected by the satellite altimetry dataset.

---

## Referee Report (RR1)

**Review of "Sentinel-3 radar altimetry for river monitoring – a catchment-scale evaluation of satellite water surface elevation from Sentinel-3A and Sentinel-3B" by Kittel et al.**

The authors did a great job in revising the paper. All my previous comments have been taken into account and the manuscript improved significantly.

I only found a few minor points that should be corrected before the paper can be published.

- I suggest to add a note on the possibility of wrong a-priori heights by the OLTC in case of inaccurate DEMs in the introduction (around line 67)
- Description of blue points in Figure 1 is missing. Moreover, the Fig. is quite small – an enlargement might improve the readability
- Is there an latitude limit in OLTC? If yes, please indicate in 2.2.4
- Legend of Fig. 2 must be corrected: SciHub is on the left and GPOD on the right hand side
- Sect. 2.3.4: Are you sure that the same models for geophysical corrections are used in GPOD and SciHub? If not: please indicate in line 216. This might be an explanation for the biases mentioned in line 222. However, in that case, these offsets will not be completely time-independent…
- Sect. 2.3.5: Please explain possible implications of not using identical DEMs for both processing chains.
- Line 243: Where can I find the classification step in Fig. 2?
- Line 278: how is the rejection done? Explain or give reference to explanation (Sect. 2.3.5 or 2.3.6?)
- Line 306: Where can I find the numbers (101 and 103) in Tab. 3?
- Table 4: Is Hydroweb also using GPOD data? Or what is the reason for placing Hydroweb below GPOD? Shouldn't that be a separate column?
- Legend of Fig.8: Please describe what is shown in column 1,2,3; What is the observation period in the middle column? 2016-2020?
- Section titles 4.1 and 4.2: please consider to change one of them; currently "processing options" are used twice.

---

## Author Response (AR2)

Response to Reviewer 1 – hess-2020-165

The reviewed version of the manuscript has been quite improved by the authors. I only have minor suggestions below (line number refers to the manuscript version with changes highlighted, at the end of the authors replies to reviewers' comments).

The authors thank the reviewer for reviewing the revised manuscript and for their useful comments throughout this process.

Specific comments:

Line 1: "Sentinel-3 is the first satellite altimetry mission to operate in Synthetic Aperture Radar (SAR) mode and in open-loop tracking mode nearly globally", maybe replace "and in open-loop" with "and one of the first to have open-loop". Indeed, Jason-3 also has open-loop targets nearly globally…

Indeed, the sentence could be misread. Sentinel-3 is particular as it is the first to operate with both (CryoSat-2 operates in SAR mode as well) near globally:

L. 1: "Sentinel-3 is the first satellite altimetry mission to operate both in Synthetic Aperture Radar (SAR) mode and in open-loop tracking mode nearly globally."

In their reply to my main concern, the authors wrote (among other):

"The purpose is not the specific Zambezi-database (although that is an important product of the study) but rather to demonstrate that by using the publicly available processing platforms, such databases can be created for any catchment globally. We believe that a framework to extract catchment-scale monitoring networks to suit specific study areas has a wide range of applications in hydrology".

It is a fair purpose. I think it would be worth to state it even more clearly in the abstract (put similar words than what you wrote in your reply to my comment), maybe around line 4 of the abstract. It would be more impactful than a sentence like "demonstrate the potential application of Sentinel-3 for monitoring river interactions with wetlands and floodplains", because 1- it was expected that S3A/B will indeed bring valuable information on these interactions, like previous altimetry mission and 2- you did not demonstrate much these interactions in the study, to my point of view.

We thank the reviewer for this suggestion and have reformulated the abstract accordingly, removing the mentioned sentence:

L. 7-8: "The objectives of the study are to demonstrate that by using publicly available processing platforms, such databases can be created for any catchment globally, to suit specific study areas and with a wide range of applications in hydrology."

Line 18-20: "This was largely related to the open-loop tracking mode: while correct on board elevation information is crucial, steep changes in the receiving window position can have detrimental effects on the WSE observations if post-processing options are not adapted." This sentence bothers me. Indeed,

the issue is that "nominal" parameters for the post-processing have been defined for close-loop cases. Your study is just showing that parameters has to be changed for open-loop mode, which makes sense. However, the open-loop mode itself is not "detrimental" to WSE estimation. As written, the sentence could give a false idea to readers unfamiliar with close-loop/open-loop modes. Please rewrite to acknowledge that as implemented, GPOD parameters (i.e. receiving window size) needs to be adjusted for open-loop mode.

The sentence has been reformulated:

L. 14-16: "This was largely related to the implementation of GPOD parameters: while correct on board elevation information is crucial, the post-processing options must be adapted to handle the steep changes in the receiving window position."

Line 25: I would remove "which holds important implications for future hydrology-oriented missions", because it is speculative and honestly, I don't see how S3A/B coverage could have important implications for SWOT (just to cite the mission you were referring to in the previous version of the manuscript).

We have removed the sentence.

Line 104: "Sentinel-3 could potentially provide a much denser VS network than Jason-2 while maintaining a relatively short return period" I agree that S3A/B have a much denser VS network than J2/3, but writing that "maintaining a relatively short return period" is somewhat excessive. S3A/B repeat period is 27 days (to be compared to 10 days for J2/3 and to less than an hour for some automatic gauges). 27 days means one measurement and rarely two measurements per month... not well adapted even to compute monthly mean. It is better than Envisat, but it could not be labelled as "short return period" (it is overselling S3A/B time sampling).

We agree with the comment – the point refers to the compromise between spatial and temporal sampling. We have reformulated to better reflect this:

L. 78-79: "Sentinel-3 provides a denser VS network than Jason-2 and Envisat, with a shorter return period than Envisat."

Figure 1, in legend, contemporary in situ gauge corresponds to orange large dots, whereas on the map there are only green large dots.

Indeed, this is a mistake – the figure has been corrected.

Line 170: replace "global ACE2 DEM" with "global DEM", as I think ACE2 is not used at all in the last versions of the OLTC. The sentence will be therefore more general.

We have reformulated the sentence:

L. 136-137: "The OLTC contains targets based on elevation information from either hydrology databases (e.g. Hydroweb), virtual stations networks and a global DEM (e.g. the Altimeter Corrected Elevations v.2 Digital Elevation Model, ACE-2)."

Line 216-220: "the standard fixed-size receiving window of 256 samples cannot store the elevation of all the samples in the echogram prior to Level-1b processing (e.g. Figure 5 in Dinardo et al. (2018)" The new explanation of the receiving window is still not clear to me. I don't really see how the "receiving window" is different from the "range window".

We have added a sentence to introduce the receiving window as well:

L. 182-184: "During processing, the return echoes from several pulses are stored in a so-called "receiving window" (i.e. a temporal matrix) and combined to form an echogram."

Section 3.1 and especially Table 3 deals with the SV fulfilling criteria on L1b and L2 in the two datasets. Among the ~250 VS that do not fulfill these criteria, how many are in open-loop mode and in close-loop mode?

All VS are in open-loop. The only difference is whether there is a target defined at the specific VS or whether the a-priori elevation is set to the previous target in the OLTC. To clarify, we have added to the OLTC section:

L. 145-148, section 2.2.4: "Sentinel-3A and Sentinel-3B operate in open-loop between 60° N and 60° S since March 9th 2019 and since the beginning of mission life respectively. Prior to March 9th 2019, Sentinel-3A followed a mode mask, switching between closed- and open-loop, and operated in open-loop over the Zambezi catchment, with the exception of a short transition phase in March 2019, when the OLTC was updated."

Hydroweb has Jason-3 and S3A/B virtual stations that are "operational" (for some validated VS, after a thorough quality control), meaning that there are automatically updated. Will it be the case for your database? Is it possible with SciHub and/or GPOD to do such automatic update? Whatever the answer is, it will be good to add it in section 4 (especially in section 4.1).

We agree, this is valuable information. We have added this as a comment to Section 4.4. where hydrological applications are discussed:

L. 485-487: "At present, the database must be updated manually, automatic download can be set up from SciHub using existing open-source tools. GPOD requires users to submit processing requests through their user accounts. Thus, it is not currently possible to implement an operational observation network using Sentinel-3 data processed on GPOD."

Response to Reviewer 2 – hess-2020-165

The authors did a great job in revising the paper. All my previous comments have been taken into account and the manuscript improved significantly. I only found a few minor points that should be corrected before the paper can be published.

The authors thank the reviewer for reviewing the revised manuscript and for their useful comments throughout this process.

I suggest to add a note on the possibility of wrong a-priori heights by the OLTC in case of inaccurate DEMs in the introduction (around line 67)

We have added in the text:

L. 67-68: "The OLTC is based on DEM information, which must be accurate and up-to-date, as new dams for instance, can alter the surface elevation significantly (Zhang et al., 2020)."

Description of blue points in Figure 1 is missing. Moreover, the Fig. is quite small – an enlargement might improve the readability

We thank the reviewer for pointing this out, the green points are the orange ones described in the legend and the figure has been enlarged.

Is there an latitude limit in OLTC? If yes, please indicate in 2.2.4

We have added the following to clarify where S3 operates in open-loop:

L. 145-148: "Sentinel-3A and Sentinel-3B operate in open-loop between 60$^\circ$ N and 60$^\circ$ S since March 9th 2019 and since the beginning of mission life respectively. Prior to March 9th 2019, Sentinel-3A followed a mode mask, switching between closed- and open-loop, and operated in open-loop over the Zambezi catchment, with the exception of a short transition phase in March 2019, when the OLTC was updated."

Legend of Fig. 2 must be corrected: SciHub is on the left and GPOD on the right hand side

Thank you for noticing this.

Sect. 2.3.4: Are you sure that the same models for geophysical corrections are used in GPOD and SciHub? If not: please indicate in line 216. This might be an explanation for the biases mentioned in line 222. However, in that case, these offsets will not be completely time-independent…

We have found a processing error of the geoid, which must be interpolated from 1Hz observations to 20Hz observations in the SciHub dataset, which led to the offset described in the manuscript. We have corrected this. The consequences are more SciHub VS are selected based on having a full record (as was already the case), but the final number based on Level-2 and Level-1b performance remains the same.

The geophysical corrections are the same, the only difference is the way the models are computed. We have checked this by small samples and found that the difference is negligible.

The error affects the number of VS retrieved from the SciHub dataset. More VS can be extracted (as there are fewer missing observations within the tracks, probably due to the empirical retracker, which are now within the +/- 30 m DEM outlier filtering), but there are also a lot more VS rejected based on the Level-1b criteria – as already addressed in the manuscript. The error was largest at the extremities of the study area, where the tributaries are narrower, and thus harder to observe by altimetry.

We have modified the total count of VS accordingly (204 to 156) and corrected Tables 3 and 4 and Figures affected by this error.

Sect. 2.3.5: Please explain possible implications of not using identical DEMs for both processing chains.

The impact would be the same as that from the DEM uncertainty itself. Both DEMs are based on the SRTM DEM too, so the effect is likely negligible in most cases.

L. 235-236: "We also do not expect the choice of DEM to impact the final results, especially as both DEMs are based on the SRTM DEM."

• Line 243: Where can I find the classification step in Fig. 2?

This classification step is part of the evaluation and thus contained within the "Waveforms (both) and RIP (GPOD)" box. We have opted not to include it in the figure, as the figure shows the mandatory steps to obtain Level-1b and Level-2 data, which can then be evaluated.

• Line 278: how is the rejection done? Explain or give reference to explanation (Sect. 2.3.5 or 2.3.6?)

We have reformulated – the rejection is based on the degree of missing data, which is introduced later:

L. 283: "Conversely, several VS with a high percentage of missing data are located in the headwater subcatchments on smaller tributaries."

• Line 306: Where can I find the numbers (101 and 103) in Tab. 3?

These numbers refer to the across dataset total, second to last column of Table 3.

• Table 4: Is Hydroweb also using GPOD data? Or what is the reason for placing Hydroweb below GPOD? Shouldn't that be a separate column?

Indeed, this is a misplaced title and column line, as Hydroweb should be a separate column. This has been corrected.

• Legend of Fig.8: Please describe what is shown in column 1,2,3; What is the observation period in the middle column? 2016-2020?

We have updated the legend. AS there is no data prior to the OLTC update, the middle column shows the WSE from March 2019 to January 2020.

• Section titles 4.1 and 4.2: please consider to change one of them; currently "processing options" are used twice.

We have removed processing options from the title of Section 4.1.

[revised manuscript text omitted]

---

## Author Response (AR3)

Reply to the editor (original comment is in italic)

*Sorry for the delay, I wanted to give the paper a final careful read and the got buried by other tasks.*

*I appreciate your efforts and find the manuscript now suitable for publication in HESS. I only have three minor suggestions*

The authors would like to thank the editor for a thorough and fair review process.

1) *Header section 2: I suggest switching the terms, i.e. 'Study area and data' as this agrees better with the order in this section*

The terms have been switched

2) *You use a lot of abbreviations. This makes the text partly hard to follow. My suggestion would be to explain (=write out) the abbreviations at some more places, suitable places include the figures (e.g., axis text "Water surface elevation, WSE [m]" instead of only WSE, same in legends, the full terms might also be used in figure/table captions. In addition, I would recommend to not use only the abbreviations in the section headers but to write out the full words.*

We agree that the high number of abbreviations can be confusing – we have used the full terms at least once in the captions so they are easy to find for the reader, when considering the tables/figures and written them out in all section headers for OLTC, GPOD and WSE.

3) *Tempus, while I would prefer using past tense for all your work, I can accept your choice. But in this case, it is important to be consistent. the sentence "Although all 156 VS were not manually checked" (L480), for instance, should in this case also be in present tense, shouldn't it?*

We have reviewed the text with a specific focus on the tense to improve consistency.

*The lines in some of the figures are very thin, but I assume the production office will complain if needed.*

Noted. In relation to the figures, we have edited figure 9 which had some overlapping features as well as the figures in the appendix as they were incorrectly compiled and not updated in the previous version.

[revised manuscript text omitted]